# Biological markers and psychosocial factors predict chronic pain conditions

**Matt Fillingim[1,2] ✉, Christophe Tanguay-Sabourin[1,2,3], Marc Parisien [1,2,4], Azin Zare[1,4], Gianluca V. Guglietti[1,2,4], Jax Norman[1,4], Bogdan Petre [5], Andrey Bortsov[6], Mark Ware[7], Jordi Perez[7], Mathieu Roy[1,8], Luda Diatchenko [1,2,4] & Etienne Vachon-Presseau [1,2,4] ✉**

Chronic pain is a multifactorial condition presenting significant diagnostic and prognostic challenges. Biomarkers for the classification and the prediction of chronic pain are therefore critically needed. Here, in this multidataset study of over 523,000 participants, we applied machine learning to multidimensional biological data from the UK Biobank to identify biomarkers for 35 medical conditions associated with pain (for example, rheumatoid arthritis and gout) or self-reported chronic pain (for example, back pain and knee pain). Biomarkers derived from blood immunoassays, brain and bone imaging, and genetics were effective in predicting medical conditions associated with chronic pain (area under the curve (AUC) 0.62–0.87) but not self-reported pain (AUC 0.50–0.62). Notably, all biomarkers worked in synergy with psychosocial factors, accurately predicting both medical conditions (AUC 0.69–0.91) and self-reported pain (AUC 0.71–0.92). These findings underscore the necessity of adopting a holistic approach in the development of biomarkers to enhance their clinical utility.

Chronic pain is a prevalent and complex condition that is difficult to measure owing to its subjective nature and substantial variability between individuals and contexts[1]. Consequently, developing biomarkers as objective measures of chronic pain has emerged as a high priority, given their potential to improve both the prognosis and management of pain[2]. For instance, biomarkers could detect pain in vulnerable populations, stratify patients into homogeneous groups to improve treatment efficacy, or provide an objective assessment of interventions in randomized controlled trials. Biomarkers could also help to define the pathogenesis of chronic pain conditions and identify novel targets for the development of new treatments[3]. Despite their tremendous value, progress in identifying reliable biomarkers for chronic pain has been slow, and further research is needed to demonstrate their utility as diagnostic and prognostic tools.

Identifying biomarkers for chronic pain has been challenging, in part because the severity of disease or injury is not a reliable indicator of the prognosis of pain. For instance, radiographic measures of joint degeneration in osteoarthritis fail to strongly predict the pain experienced by the patient[4]. Other limiting factors hindering the identification of reliable biomarkers for pain include small sample sizes[5], lack of dataset generalizability[6], restricted availability of biological features and the investigation of a limited range of pain phenotypes[7]. In addition, the distinction between acute and chronic phases (that is, duration >3 months)[8], its aetiology (nociplastic, neuropathic or

[1]Alan Edwards Centre for Research on Pain, McGill University, Montreal, Quebec, Canada. [2]Department of Anesthesia, Faculty of Medicine and Health Sciences, McGill University, Montreal, Quebec, Canada. [3]Faculty of Medicine, Université de Montréal, Montreal, Quebec, Canada. [4]Faculty of Dental Medicine and Oral Health Sciences, McGill University, Montreal, Quebec, Canada. [5]Department of Psychological and Brain Sciences, Dartmouth College, Hanover, NH, USA. [6]Center for Translational Pain Medicine, Duke University, Durham, NC, USA. [7]Alan Edwards Pain Management Unit, McGill University Health Center, Montreal, Quebec, Canada. [8]Department of Psychology, McGill University, Montreal, Quebec, Canada. ✉e-mail: matthew.fillingim@mail.mcgill.ca; etienne.vachon-presseau@mcgill.ca

nociceptive)[9] and its spatial extent (that is, localized or widespread)[10] further complicate the identification of comprehensive signatures for chronic pain.

Adding to the challenge, efforts towards biomarker identification have occurred largely in isolation, with biological predictors examined independently of other individual factors or environmental contexts. The interplay between biomarkers from different body systems and psychosocial factors has been largely ignored. To fully leverage biomarkers in the context of personalized medicine, a holistic understanding of pain through a multifactorial approach is necessary[1]. The emergence of large prospective biobanks, such as the UK Biobank, provides a unique opportunity to concurrently evaluate numerous biomarkers and integrate them within a comprehensive framework for the classification and prediction of chronic pain.

In this study, we applied machine learning to multidimensional biological modalities including brain imaging, bone imaging, blood immunoassays and genetics to derive candidate biomarkers for chronic pain. The relevant biomarkers were then contextualized by comparing their performance with psychosocial predictors of chronic pain that substantially contribute to the development of chronic pain[10]. The synergy between biomarkers and psychosocial drivers of pain conditions was subsequently demonstrated to uncover the intricate biopsychosocial interactions that predict pain.

## Results

Our study utilized data from the UK Biobank to train and validate biomarkers for chronic-pain-related medical conditions and self-reported pain. The generalizability of the identified biomarkers was then tested using the All of Us Research Program (AoU; $n = 27,151$) and the Open Pain data repository ($n = 250$). A total of 493,211 participants were enrolled in the UK Biobank and completed the initial baseline visit. During that visit, participants were asked if they experienced pain that interfered with their usual activities in the last month at various body sites, including the head, face, neck/shoulder, stomach/abdominal, back, hip, knee and pain all over (PAO) their body. Participants who reported pain were then queried about pain lasting for more or less than 3 months (that is, chronic or acute). Participants were also asked about their medical conditions diagnosed from a physician and their use of medications. Blood samples were collected from participants to conduct whole-genome sequencing and bioassay measurements. A subsample of 19,360 participants underwent a first follow-up visit about 4 years after baseline where blood was recollected, and a subsample of 48,079 participants underwent another follow-up visit about 9 years after baseline, where brain imaging (magnetic resonance imaging, MRI) and bone imaging (dual-energy X-ray absorptiometry) were acquired. The timeline of data acquisition (T0, baseline; T1, about 4 years later;

T2, about 9 years later) and the sample sizes available are summarized in Extended Data Fig. 1. A schematic of the study aims is shown in Fig. 1a.

### Biomarkers for pain-related medical conditions

We initially aimed to identify biological signatures predictive of various medical conditions associated with chronic pain. Models underwent training through nested CV, using logistic regression to classify individuals reporting pain versus those reporting no pain (Extended Data Fig. 1). These linear classifiers were trained on measures of inflammatory/immune, metabolic and haematological assays to derive the blood biomarkers, measures from high-resolution T1-weighted imaging (grey matter), diffusion-weighted imaging (white matter) and resting-state imaging (functional connectivity) to derive the brain biomarkers, and measures from bone mineral content, shape and density to derive the bone biomarkers. The genetic biomarkers were polygenic risk scores (PRS) derived at different thresholds from whole-genome sequencing data (biological features are described in Supplementary Tables 1–6).

We selected 35 medical conditions in which the prevalence of chronic pain ranged between 48% and 80% of cases, most commonly presenting as multisite pain (Fig. 1b). By comparison, the prevalence of chronic pain in participants without any diagnosed conditions was approximately 25% and typically localized to a single site. Medical conditions with pronounced multisite pain exhibited symptoms in the neck, back, hip and knee, whereas conditions where pain was concentrated around a single epicentre were localized in the head, face and abdomen (Fig. 1b). Moreover, conditions such as fibromyalgia, chronic fatigue syndrome and polymyalgia rheumatica were characterized by pain experienced all over the body (Fig. 1b). The demographics varied between pain-associated medical conditions and are shown in Extended Data Fig. 2.

The best predictions were obtained for conditions with a well characterized pathogenesis, such as multiple sclerosis (brain, AUC 0.87), gout (blood, AUC 0.83) and polymyalgia rheumatica (blood, AUC 0.82). About half of the selected medical conditions could be predicted with good accuracy (AUCs >0.70), including fibromyalgia (brain, AUC 0.70), osteoporosis, spinal stenosis (bone, AUCs 0.72 and 0.70), peripheral neuropathy and arthritis (blood, AUCs 0.73 and 0.70). The full weighted signatures for each of these biomarkers are shown in Supplementary Figs. 1–10. Overall, conditions with well-defined pathophysiology (for example, gout and polymyalgia rheumatica) showed higher accuracy compared with those primarily defined by symptomatology (for example, non-migraine headache, irritable bowel syndrome and back pain; Fig. 1c). Importantly, although these conditions are defined by specific pathophysiology, our findings demonstrate that they can be detected and predicted using routinely collected and widely accessible biological measurements.

**Fig. 1 | Classifying pain-associated diagnoses using biological and psychosocial modalities. a**, A schematic illustrating the study workflow. **b**, Top: bar plots presenting the prevalence of chronic pain across 35 pain-associated diagnoses, categorized by the number of self-reported pain sites and ordered by overall chronic pain prevalence. Counts of diagnoses at baseline are in black and 9-year follow-up in grey. Bottom: a heat map displaying the prevalence of pain sites for each diagnosis, normalized (z score) across conditions for each specific site. The diagnosis-free control group is labelled in light grey. **c**, Bar plots displaying the mean test set ROC-AUC of models in classifying pain diagnoses, with error bars indicating the 95% CI, estimated from 1,000 bootstrap samples over five iterations of fivefold CV ($n = 25$). Overlaid points correspond to AUC scores from individual validation folds ($n = 25$ points total). The bars represent the highest ROC-AUC scores achieved, separated into biological (left) and psychosocial (right) modalities. Bubble heat maps show ROC-AUC scores for modality subcategories, where applicable; bubble colour indicates the absolute AUC score, and bubble size reflects the z score of the AUC relative to other diagnoses within a given modality or subcategory. Only diagnoses with z scores above zero, indicating performance above the group mean AUC, are shown.

For clearer visualization, diagnoses are grouped by their best biological modality performance (that is, highest AUC score) into four categories, from poor [0.60–0.65 AUC) to excellent (0.75+ AUC). **d**, A scatterplot showing the comparison of AUC scores between the best biological and psychosocial modalities for each pain diagnosis, with points coloured according to the best biological modality and labelled by diagnosis. The point size reflects the absolute AUC score difference, highlighting the discrimination discrepancy between biological and psychosocial factors for each diagnosis. Adjacent Venn diagrams depict unique and shared deviance explained ($D^2$), such as pain diagnoses (for example, RA and fibromyalgia) and self-reported chronic pain by the best biological (left circle) and psychosocial modality (right circle), with overlaps shown in the centre. Corresponding stacked bar plots display the same $D^2$ information, with each segment colour-coded by modality. Ha, headache; F, facial; N/S, neck or shoulder; S/A, stomach or abdominal; B, back; Hp, hip; K, knee; IBS, irritable bowel syndrome; Dx, diagnosis; Infl., inflammatory; COPD, chronic obstructive pulmonary disease; Psych. - Bio. AUC, area under the curve difference between psychosocial and biological predictors.

To better understand the pathogenesis of the various medical conditions, the performance of biomarkers (measured using their AUCs) within each biological category were normalized across medical conditions, after breaking down brain imaging and biochemical assays into their subcategories (Fig. 1c). Interestingly, the condition that was best predicted from PRS was ankylosing spondylitis, a condition with well-established genetic contributions[11]. Moreover, brain functional connectivity best predicted nociplastic pain conditions, such as fibromyalgia and chronic fatigue syndrome, which are hypothesized to be caused by central amplification of the nervous system[12].

The comparison between biomarker subcategories also revealed that certain medical conditions could be independently predicted from multiple biological modalities (for example, ankylosing spondylitis, fibromyalgia and polymyalgia rheumatica) while others were distinctly predicted from a single biological modality (for example, chronic fatigue syndrome, osteoporosis and ulcerative colitis). Sex differences in the performance of biomarkers across all modalities and medical conditions are detailed in Extended Data Fig. 3. Generally, differences in biomarker performance between sexes were minimal. Notable exceptions, however, included the bone-based marker for

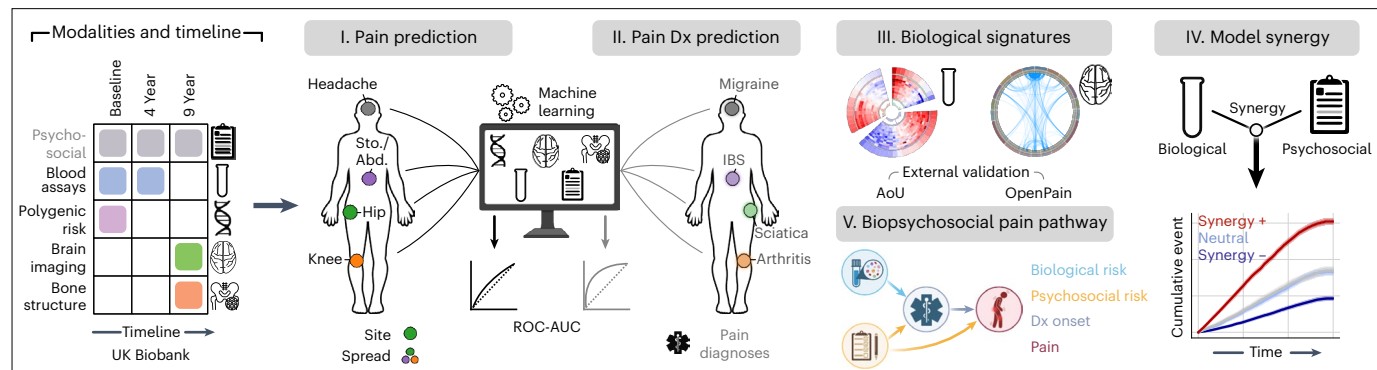

osteoporosis, which performed better in men, and the brain-based marker for peripheral neuropathy, which was more effective in women.

We next entered psychosocial features into the same machine learning analytic pipeline to classify each pain-associated medical condition. Most pain conditions were classified with fair-to-excellent accuracy, with conditions recognized as nociplastic being the most accurately classified, underscoring the significance of psychosocial factors in conditions such as fibromyalgia and chronic fatigue syndrome (Fig. 1c). A closer examination of these psychosocial contributors, which could serve as either predictors or outcomes of the conditions, revealed mood and sleep disturbances as primary features for fibromyalgia, lower substance use as a notable characteristic of chronic obstructive pulmonary disease, reduced physical activity as a marker of multiple sclerosis, and occupational factors as prominent indicators of carpal tunnel syndrome. Using support vector machines and gradient boosting decision trees as alternative machine learning algorithms showed consistent results, although decision trees artificially improved the brain and blood-based biomarkers (Extended Data Fig. 4). Altogether, our results showed that a mosaic of biological and psychosocial markers coalesce in the manifestation of various conditions.

The summary plot shown in Fig. 1d shows the simultaneous performance of models trained on either biological or psychosocial features for each condition. These findings underscore the variance in predictive capability between biological and psychosocial factors across different conditions, enabling the identification of conditions predominantly predicted by biological markers (for example, gout), those primarily predicted by psychosocial markers (for example, non-migraine headache) and those where both types of marker play a substantial role (for example, polymyalgia rheumatica, rheumatoid arthritis (RA) and stroke). The deviance explained—reflecting the reduction in discrepancy between the model's predictions and the observed data—greatly varied depending on the medical condition. For instance, RA was most strongly determined from blood assays with a strong overlap with psychosocial factors, whereas fibromyalgia was most strongly determined from psychosocial factors with little overlap with the brain biomarker (Fig. 1c). In the case of self-reported pain, the deviance explained was low and almost exclusively from psychosocial factors, emphasizing the limits of biological predictors for the subjective experience of pain.

## Validated blood-based signature for chronic pain conditions
Biomarkers developed from blood immunoassays outperformed other categories of biomarkers, successfully predicting 13 pain-associated medical conditions with good accuracy (AUC >0.70; Fig. 1c). Notably, these biomarkers demonstrated robust predictive performance on other conditions they were not originally trained to predict (Extended Data Fig. 5). We therefore trained a composite signature to concurrently predict these 13 medical conditions (Fig. 2a,b). The composite signature was composed of strong positive coefficients (elevated in disease) for C-reactive protein, neutrophils and gamma glutamyl-transferase, alongside notable negative coefficients (reduced in disease) for lymphocyte percentage, high-density lipoprotein (HDL) cholesterol and albumin (Fig. 2c). The effect of different medication categories (for example, glucocorticoids, analgesics and statins) on the composite signature was tested by reassessing the model's performance after removing individuals who were taking medications from a specific category (Extended Data Fig. 6). The performance was generally stable, with the sharpest decreased in AUC observed upon excluding users of statins, analgesics or antihypertensives (AUCs −0.05, −0.06 and −0.06, respectively; Extended Data Fig. 6).

Using longitudinal data from follow-up visits at 4 (T1) and 9 (T2) years, the composite signature measured at baseline (T0) predicted newly developed diagnoses at T1 (average AUC 0.70, range 0.62–0.79), with robust predictions for over half of the conditions (AUCs >0.70). The composite signature measured at T0 could also predict newly diagnosed medical conditions at T2 that were not yet diagnosed at T1 (for example, Crohn's disease, gout and psoriatic arthropathy AUCs >0.70; Fig. 2b). Effect sizes are shown in Fig. 2d, and longitudinal changes in the expression of the composite signature are shown in Fig. 2e. A simplified version of the composite signature (Supplementary Table 1) was validated in the AoU (cross-sectional; Fig. 2f,g), obtaining similar performances and demonstrating its generalizability across conditions in a cohort from a different country. Notably, participants in this validation cohort were younger (Extended Data Fig. 7) and therefore less likely to use statin or antihypertensive medication prescriptions.

## Validated brain-based signature for nociplastic conditions
In contrast to blood-based biomarkers, biomarkers based on functional connectivity were largely specific to distinctly nociplastic pain conditions (Fig. 3a) such as fibromyalgia, chronic fatigue syndrome and widespread pain (AUCs 0.64–0.66). The weights of the logistic regression models for these three multivariate signatures are shown in Supplementary Fig. 5. The encoding maps showing the univariate structure coefficients of nodes associated with the model prediction are shown in Fig. 3b.

A new brain-based model was developed to predict a composite phenotype that encompassed all three nociplastic conditions. The weights of the signature are shown in Supplementary Fig. 10, and the structure coefficients of edges associated with the model prediction

---

**Fig. 2 | Deriving and validating a composite blood assay signature of pain-associated conditions. a**, A schematic of the composite signature's development for 13 pain-related diagnoses, plus a timeline of UK Biobank data points used. **b**, The diagnostic and prognostic efficacy of the signature, assessed by Cohen's *d* and ROC-AUC, comparing individuals with and without diagnoses, and those who develop diagnoses versus those remaining diagnosis-free. Diagnostic accuracy is measured at baseline, prognostic accuracy at 4 and 9 years post-baseline. Bars show the mean Cohen's *d* from 1,000 bootstrap resamples, with error bars indicating the ±95% CI. Overlaid points are a random subsample (*n* = 50) of these 1,000 resamples. **c**, A circular graph depicting the signature's structure: the outer heat map displays blood marker coefficients; the middle layers show individual diagnosis marker coefficients, numbered as in **b**; and the inner layer presents the standard deviation of marker coefficients. Each segment corresponds to inflammatory/immune, metabolic or haematological assays. **d**, Pooled effect sizes from **c**, measured by Cohen's *d*, compare baseline diagnoses or diagnoses developing by 4 or 9 years with diagnosis-free individuals. A two-sided Wilcoxon rank-sum test used to compute *P* values. **e**, Temporal changes in the signature for participants with ongoing diagnoses, newly developing diagnoses at 4 or 9 years, or remaining diagnosis-free. The significance of signature change between each group and the Stay Dx-free control group was estimated using two-sided

linear mixed-effects models, with adjustments for multiple comparisons using Bonferroni correction ($P_{ongoing} = 0.07$, $P_{4-yr} < 0.001$, $P_{9-yr} = 0.001$; $P_{bonf.} < 0.05$). Data are presented as mean ± 95% CI from 1,000 bootstrap resamples. **f**, A simplified model using the top 10 assays was validated with data from the AoU. **g**, Its discriminatory power is shown via Cohen's *d* and ROC-AUC. Bars indicate the mean from 1,000 bootstraps (±95% CI), with overlaid subsampled points (*n* = 50). Inflammatory/immune markers: CRP, C-reactive protein; Neut, neutrophil; WBC, white blood cell; Mono, monocyte; Eos, eosinophil; baso, Basophil; Lymph, lymphocyte. Metabolic markers: GGT, gamma glutamyl transferase; Cys C, cystatin C; TG, triglyceride; ALP, alkaline phosphatase; UA, uric acid; HbA1c, glycated haemoglobin, Glu, glucose; ALT, alanine aminotransferase; AST, aspartate aminotransferase; Cr, creatinine; TP, total protein; Testo, testosterone; Ca, calcium; ApoB, apolipoprotein b; TBil, total bilirubin; IGF1, insulin-like growth factor; LDL-C, low-density lipoprotein cholesterol; Alb, albumin; TC, total cholesterol. Haematological markers: HLR, high light scatter reticulocyte percentage; Retic, reticulocyte; IRF, immature reticulocyte fraction; RDW, red blood cell distribution width; PCT, platelet count; PLT, platelet count; PDW, platelet distribution width; nRBC; nucleated red blood cell; SCV, sphered cell volume; MPV, mean platelet volume; Hct, haematocrit; Hgb, haemoglobin; Dx, diagnosis; UKBB, UK Biobank; nos, not otherwise specified; Dist., distribution.

are depicted in brain rendering in Fig. 3c and a circular plot in Fig. 3d. The conditions were characterized by a general pattern of dysconnectivity between brain regions, notably involving the visual cortex, the brainstem and cerebellum, the dorsal and ventral attention networks and the sensorimotor network. This brain-based signature, termed nociplastic functional signature (NFS), was validated in four distinct external datasets available on the Open Pain repository, which included individuals with chronic pain who predominately reported high levels of pain severity and impact (Extended Data Fig. 7). We then compared the performance of our brain signature with the tonic pain signature (ToPS), an existing brain-based signature trained on capsaicin-induced

pain that effectively classified clinical pain in cohorts from the Open Pain repository[6]. The signatures were thresholded across different densities (100% to 0.5%) and then applied in the Open Pain repository to classify individuals with chronic pain from pain-free individuals (Fig. 3e). While both NFS and the ToPS obtained fair to good performances when restricted to the most important functional connections, the NFS was stable across all thresholds.

## Biomarkers for self-reported pain
We have identified various biomarkers effective in classifying chronic pain associated medical conditions. Our findings, however, suggest

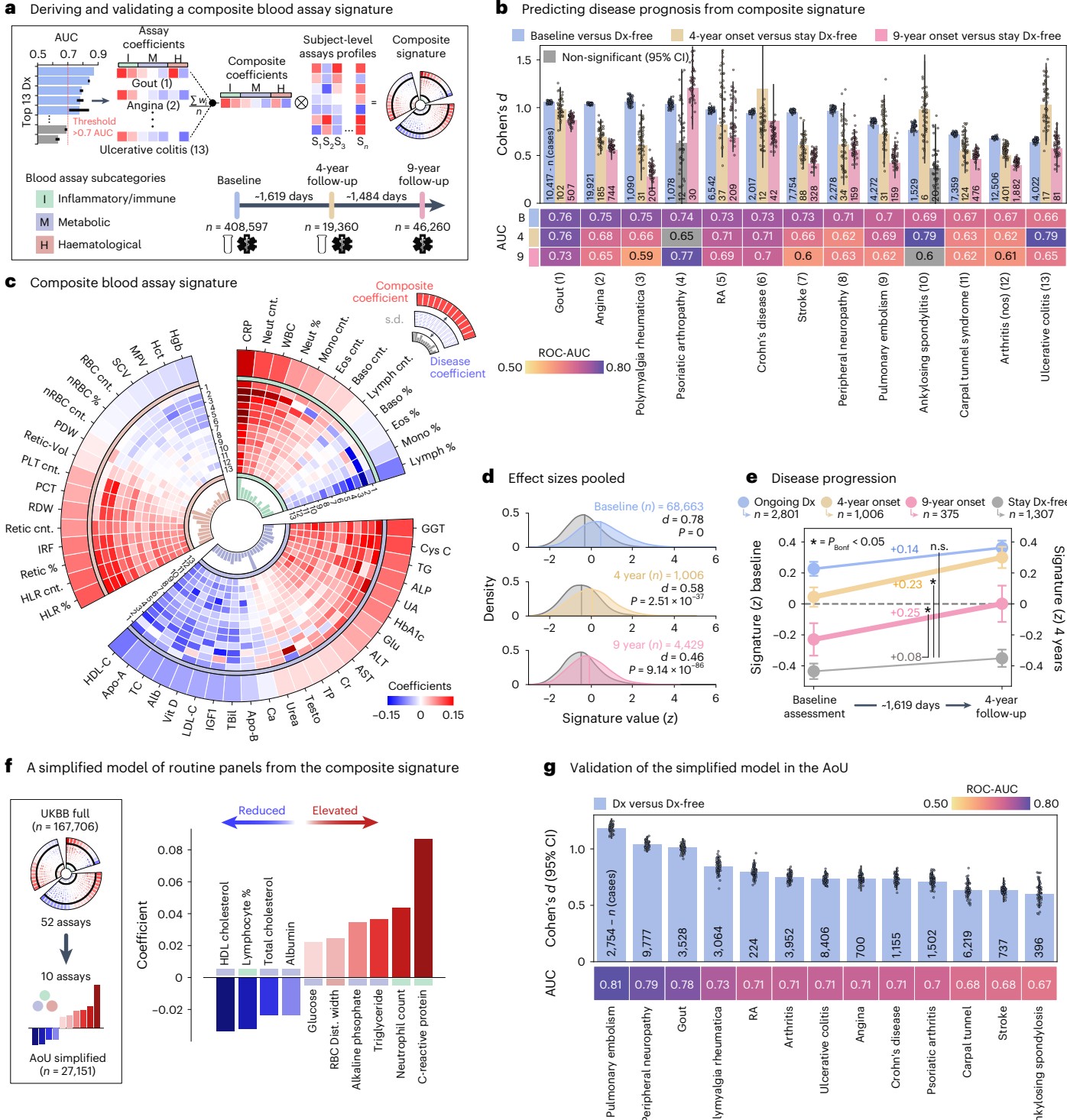

**a** Deriving and validating a composite blood assay signature

**b** Predicting disease prognosis from composite signature

**c** Composite blood assay signature

**d** Effect sizes pooled

**e** Disease progression

**f** A simplified model of routine panels from the composite signature

**g** Validation of the simplified model in the AoU

**a** Deriving a NFS of nociplastic conditions

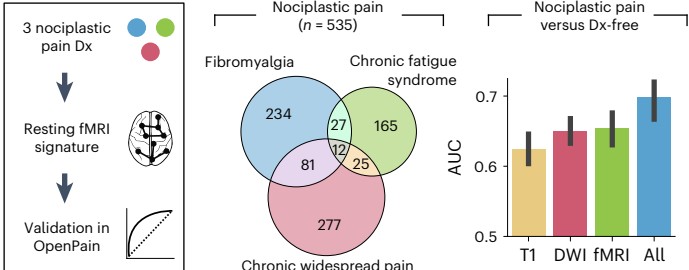

**b** Functional disconnectivity patterns in nociplastic conditions

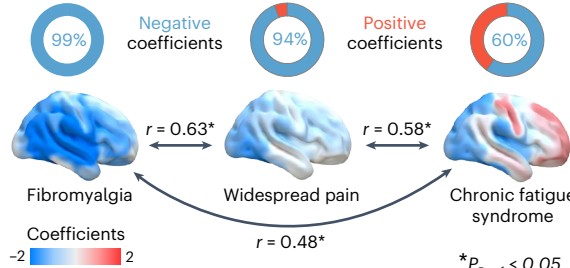

**c** NFS node-level disconnectivity patterns

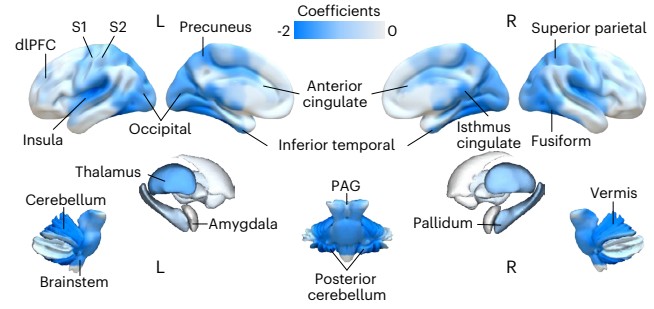

**d** NFS functional connectivity (top 10%)

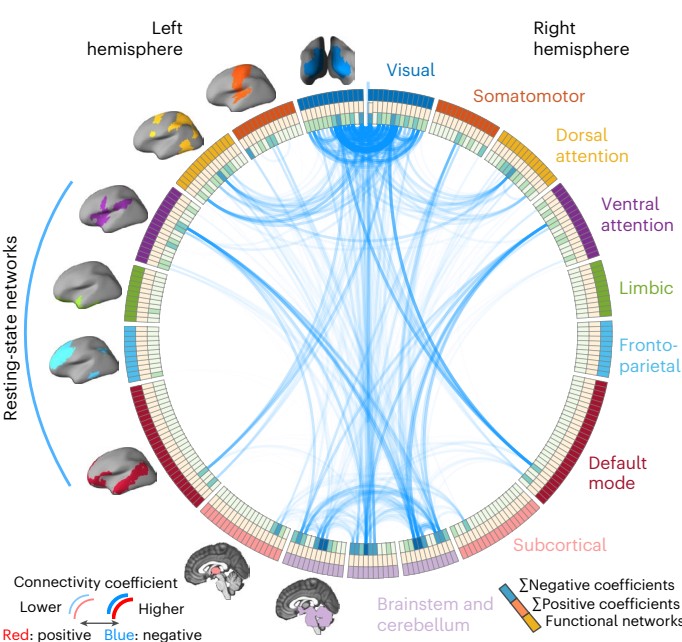

**e** Validation in OpenPain

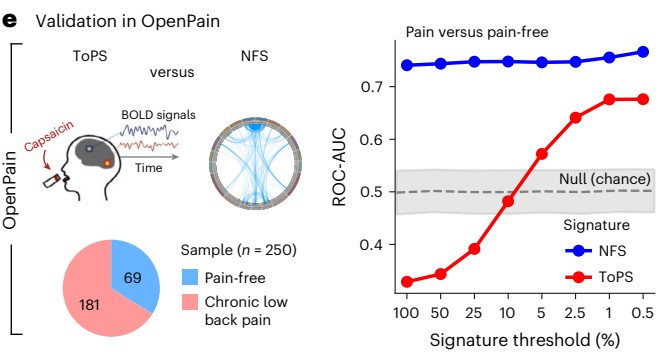

**Fig. 3 | Deriving and validating a multivariate functional connectivity signature of nociplastic pain. a**, A schematic describing the steps implemented to develop the NFS. The Venn diagram shows the sample sizes and sample overlap among the three nociplastic conditions—fibromyalgia, chronic fatigue syndrome and chronic widespread pain—used to derive a combined nociplastic pain phenotype (n = 535). Bar plots show ROC-AUC results from models trained on brain imaging modalities to classify this phenotype, with error bars denoting the 95% CIs from 1,000 bootstrap samples over five iterations of fivefold CV. **b**, Top: donut plots displaying the proportions of positive versus negative structure coefficients from models predicting the component nociplastic conditions, based on resting-state fMRI (rsfMRI) data. Bottom: cortical surface renderings visualizing connectivity, thresholded to highlight the top 25% of structure coefficients, which represent the sum of dynamic conditional correlation across brain parcels. Arrows interlinking the cortical renderings depict the association (two-sided Pearson correlation, all P < 0.001 Bonferroni corrected) between the complete unthresholded vectors of structure coefficients (parcel-to-parcel connectivity) for the respective conditions.

**c**, Visualization of nociplastic signature connectivity, thresholded to emphasize the top 25% of structure coefficients that denote the sum of dynamic conditional correlations across brain parcels. **d**, A Circos plot illustrating the top 10% of connectivity links (represented by structure coefficients) that constitute the nociplastic signature from the resting-state fMRI model, mapped across the canonical resting-state networks. **e**, Validation of the resting-state fMRI nociplastic signature (NFS) in four aggregated external cohorts from the OpenPain repository (n = 250), benchmarked against a prevalidated neural signature of capsaicin-induced sustained pain (ToPS). The performance in discriminating pain versus pain-free groups across various densities of NFS and ToPS within the OpenPain cohorts is depicted in the plot to the right along with the standard deviation band of a null classification model generated from 10,000 permutations of the NFS. Nocip, nociplastic; T1, T1 structural brain imaging; DWI, diffusion-weighted imaging; All, combination of the features from the T1, DWI and fMRI brain imaging modalities; dlPFC, dorso-lateral prefrontal cortex; S1, primary somatosensory cortex; S2, secondary somatosensory cortex; BOLD, blood-oxygen-level-dependent; PAG, periaqueductal grey.

that biomarkers' performance may be more limited for self-reported pain (Fig. 4a). We therefore trained new models to classify the subjective report of presence or absence of pain, regardless of its location on the body. Here, biomarkers from either brain, bones, blood, or genes all demonstrated weak ability to distinguish between participants with and without pain, indicating their unreliability in pain prediction (chronic pain versus pain free: all AUC between 0.55 and 0.59; acute pain versus pain free: all AUC between 0.52 and 0.54; Fig. 4b). Training separate models to predict pain located at each body site

showed minimal improvement in performance (all AUC <0.62; Fig. 4c). However, when models were trained to predict pain reported 'all over the body', all biological categories, with the exception of genetic PRS, showed increased performance (AUC 0.66–0.69). This contrasts with models trained on psychosocial factors including mood, sleep, life stressors, neuroticism, substance use, physical activity and socio-economic factors (psychosocial features are described in Supplementary Table 6), which performed well across all body sites (AUC >0.70; Fig. 4c) and exhibited excellent accuracy for pain reported all over

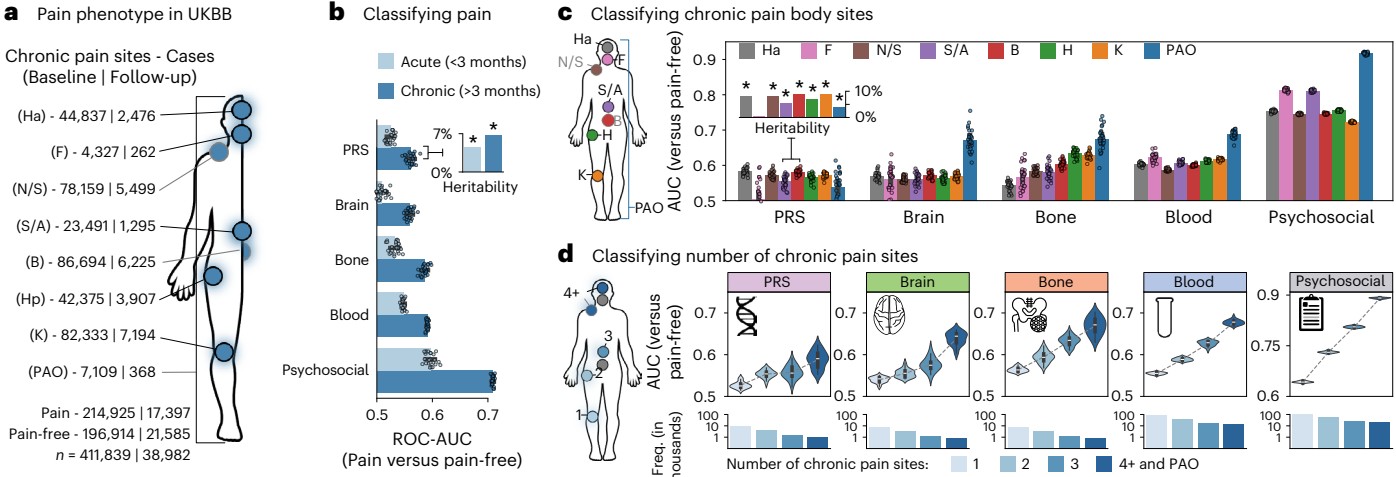

**Fig. 4 | Classifying chronic pain phenotypes using biological and psychosocial modalities. a**, An anatomical body map of chronic pain sites and their counts for the full (baseline) sample and for individuals with a follow-up visit 9 years later. **b**, The performance of machine learning models in classifying participants reporting acute (light blue) and chronic (dark blue) pain from pain-free ($n_{chronic}$ = 2,679–42,985; $n_{acute}$ 1,012–16,259). Bars show test set ROC-AUC scores, with error bars indicating the 95% CI, estimated from 1,000 bootstrap samples over five iterations of fivefold CV ($n$ = 25). Overlaid points correspond to AUC scores from individual validation folds ($n$ = 25 points total). Also included are heritability estimates from GWAS for both acute and chronic pain types. These heritability estimates are significant according to a two-sided, FDR-corrected Wald Test ($P_{FDR}$ < 0.001 for both). **c**, Left: body map of chronic pain sites. Right: bar plots grouped by modality, showing machine learning model performance (ROC-AUC) in distinguishing participants with specific pain sites from pain-free controls, as described in **b**. Bars represent mean test set ROC-AUC scores, with error bars indicating the 95% CI, estimated from 1,000 bootstrap samples over five iterations of fivefold CV ($n$ = 25). Heritability estimates are shown for each pain site and are significant using a two-sided, FDR-corrected Wald test (all $P_{FDR}$ < 0.001), except for facial pain ($P_{FDR}$ = 0.093). **d**, Left: body map showing aggregation of chronic pain sites to depict a phenotype based on the number of self-reported pain sites (that is, anatomical pain spread), ranging from 1 to 4 or more distinct sites. Right: violin plots illustrate the distribution of test set ROC-AUC scores across five iterations of fivefold CV (25 total AUC scores) for models classifying chronic pain spread versus pain-free individuals. Below, bar plots indicate the frequency of each pain spread phenotype in modality-specific models. Ha, headache; F, facial; N/S, neck or shoulder; S/A, stomach or abdominal; B, back; Hp, hip; K, knee. The asterisk denotes a Bonferroni-corrected $P$ value of less than 0.05.

the body (AUC 0.92). Using alternative machine learning algorithms did not improve the prediction of self-reported pain (Extended Data Fig. 4). Our findings show that candidate biological markers did not predict chronic pain effectively unless pain was spread across body sites. Conversely, psychosocial factors offered a more reliable prediction of self-reported pain.

We then trained separate biomarkers to predict the number of pain sites reported by the participant, varying the extent of pain spreading as the target instead of the anatomical location (Fig. 4d). Here, each biomarker showed a consistent increase in its performance as the number of pain sites was increased (average improvement in AUCs between single site and 4+ sites was 0.10; Fig. 4d). Feature encoding maps of biomarkers trained on varying number of pain sites were generally consistent, suggesting that pain spreading did not modify the biological signature but instead amplified its expression (Extended Data Fig. 8). Our results suggest that biomarkers showed increased sensitivity to the spreading of pain rather than to its specific location, although the performance remained relatively poor compared with psychosocial factors.

**Synergy between biological and psychosocial markers for pain**
Lastly, we aimed to generate a holistic biopsychosocial framework by examining the interactions between biomarkers and psychosocial factors in relation to the onset and progression of chronic pain (Fig. 5a). To investigate this interaction, participants were grouped into five quintiles representing the expression of their pooled biomarker and their psychosocial risk. We extracted log probabilities from our top-performing models as individual risk scores, reflecting the likelihood of subjects having certain conditions. These scores were then averaged to generate a pooled risk score for each subject, effectively summarizing their overall biomarker and psychosocial risk profile. Figure 5b shows the expression of the pooled risk scores and the odds

ratio (OR) of diagnosis for the 13 medical conditions predicted from that risk score, as initially shown in Fig. 2b. Being in the highest quintile for either biomarkers (H) or psychosocial factors (H) elevated the risk of having the medical condition, as indicated by the OR range between 6 and 18. However, being in the highest quintile for both (H–H) dramatically amplified the risk, with the range of ORs escalating between 18 and 42, as shown using a log scale in Fig. 5b,c. This effect was also observed in cross-sectional data when varying the biomarker modality and using the medical conditions best predicted from those biomarkers: using the brain-based risk common to nociplastic conditions for fibromyalgia and chronic fatigue syndrome (Fig. 5d) or bone-based risk for osteoporosis, spinal stenosis, carpal tunnel syndrome and gout (Fig. 5e). Overall, the synergy between biological and psychosocial markers was consistently observed across different biomarker modalities and pain-associated medical conditions, substantially enhancing the prediction of various medical conditions.

Concordant results were obtained using the blood risk score in the longitudinal data (Fig. 5f), as new diagnoses over a period of 15 years occurred more rapidly in the H–H group (hazard ratio (HR) 2.26 (2.20–2.32) compared with participants in either H–L (HR 1.06 (1.01–1.11)) or L–H (HR 1.05 (0.99–1.11)) or L–L groups (HR 0.54 (0.51–0.56); Fig. 5g). Thus, participants with high biomarker and psychosocial risk (H–H quintiles at baseline) exhibited a significantly increased incidence of a new medical condition compared with subjects with high biomarker but low psychosocial risk scores (H–L quintiles at baseline). These findings imply that the same biomarker performs differently depending on the psychosocial context in which it is being tested.

One of the important limitations of these biomarkers was their inability to explain the subjective experience of pain associated with the medical condition (Fig. 5h). For each medical condition, the variance in the number of pain sites was better explained from the psychosocial risk score ($r^2$ ranging from 0.07 to 0.28) than the expression of the

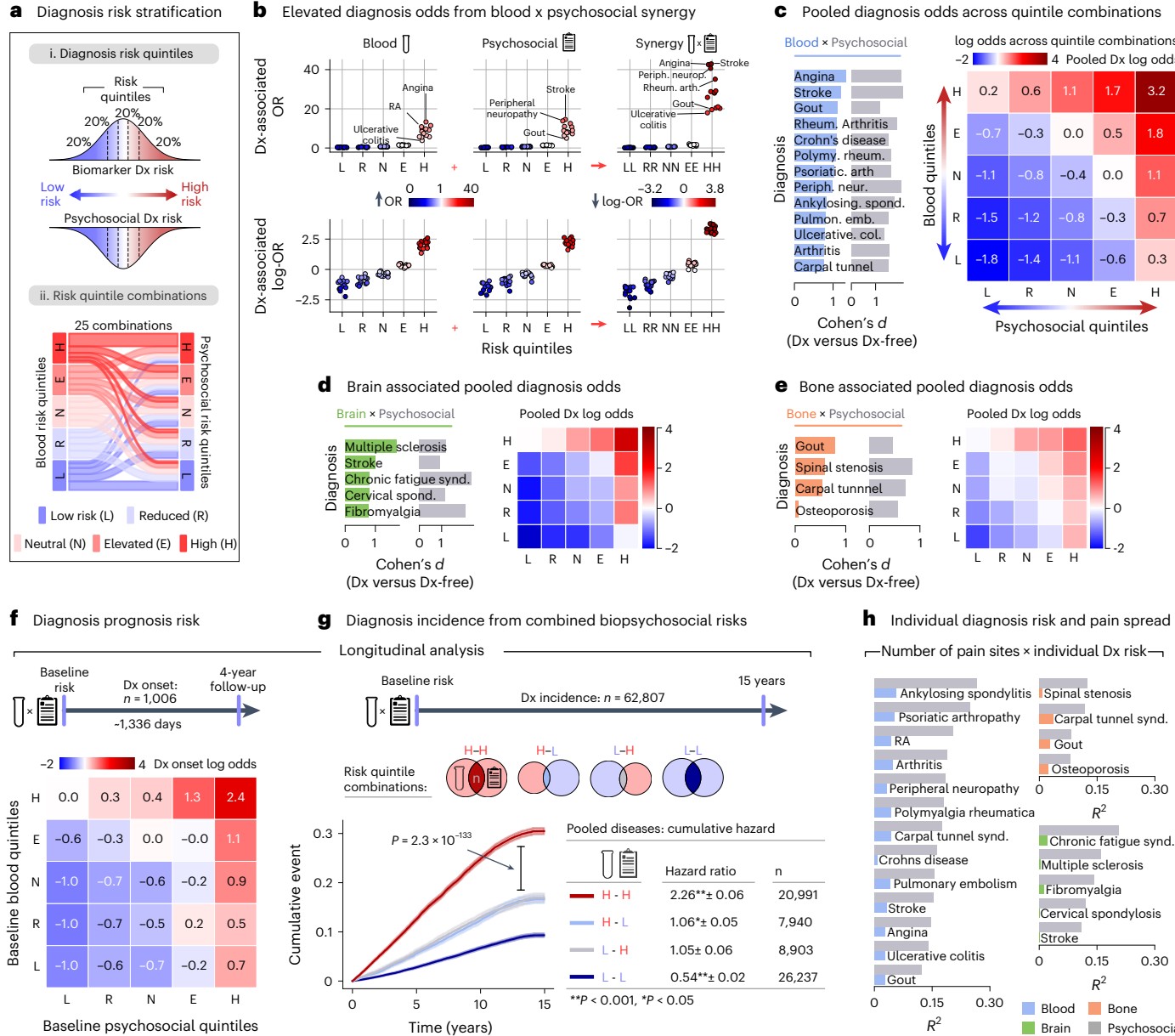

**Fig. 5 | Assessing biopsychosocial synergy in the prognosis of pain-associated medical conditions. a**, An overview of the development of diagnosis risk stratification models based on pooled probabilities from biological (blood, brain and bone) and psychosocial modalities. (i) Biological and psychosocial risk scores were segmented into quantiles for stratification. (ii) A Sankey diagram visualizes the possible combinations blood and psychosocial risk quintiles; this operation is similarly conducted for bone and brain risk models. **b**, Top: diagnosis-associated ORs for each diagnosis are computed for participants within each risk quantile of blood assay risk scores, psychosocial risk scores and combined risk scores, in comparison with all other participants. Bottom: the ORs are transformed to log-odds to elucidate the protective effect associated with lower risk quantiles. **c**, The performance of the pooled risk scores for each diagnosis was measured against diagnosis-free participants using Cohen's $d$ effect size. The heat map displays the log-ORs of having a diagnosis across all risk quintile combinations, highlighting the synergistic impact of high combined

biological and psychosocial risks and the protective effects conferred by lower combined risks. **d**,**e**, The analysis performed in **c** was replicated for diagnoses most accurately classified by the brain (**d**) and bone (**e**) modalities. **f**, log-ORs depict the synergy between blood assay biomarkers and psychosocial factors in disease prognosis 4 years later. **g**, Kaplan–Meier curves show the cumulative incidence of receiving a diagnosis up to 15 years after baseline, segregated into four groups according to combinations of blood and psychosocial risk quantiles (high–high, high–low, low–high and low–low). HRs are calculated using Cox-proportional hazard models, while the $P$ value of the differences between groups is calculated using a two-sided log-rank test ($P_{HH} < 0.001$, $P_{HL} = 0.047$, $P_{LH} = 0.055$, $P_{LL} < 0.001$). **h**, Bar plots illustrate the association between biological and psychosocial risk scores for blood (left), bone (top right) and brain (bottom right) risks with categories of pain site spread (ranging from 1 to 4+ sites) The $R^2$ values indicate the strength of these associations, as determined by Spearman's rank correlation. *$P < 0.001$.

biomarker (best modality: $r^2$ ranging from 0.00 to 0.06). These findings were consistent across all categories of biomarkers and medical conditions evaluated. We conclude that biomarkers and psychosocial factors do not act in isolation but synergize to influence the development and prognosis of pain-related medical conditions. However,

solely psychosocial factors reliably determine the presence, bodily distribution and impact of chronic pain.

The blueprint of this holistic framework for the prediction of chronic pain is illustrated by applying structural equation modelling to longitudinal data available in the UK Biobank (Fig. 6). Here, the

A prognostic biopsychosocial pathway of chronic pain

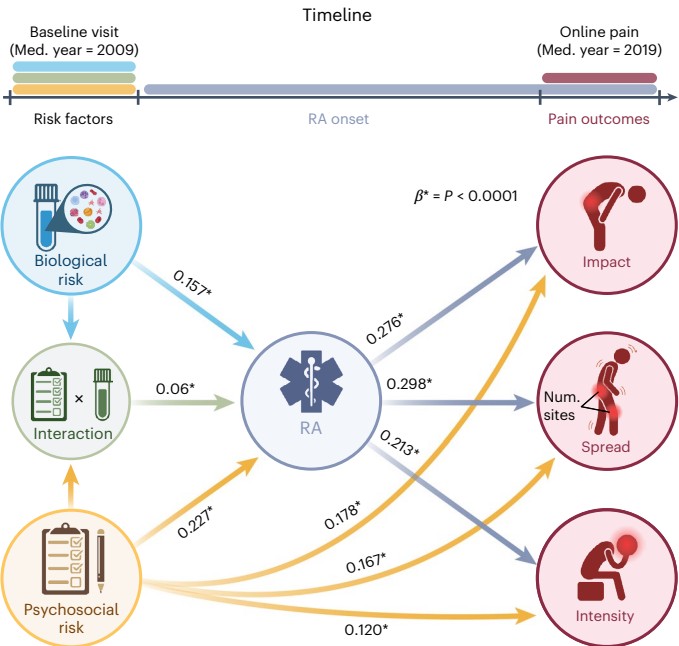

**Fig. 6 | A holistic biopsychosocial framework for the development of chronic pain.** A framework for the development of chronic pain over time is depicted through a data-driven structural equation model utilizing longitudinal data. The baseline biological risk is calculated using a blood risk score for RA, derived from 52 blood markers encompassing inflammatory and immune, metabolic and haematological assays. Baseline psychosocial risk is quantified using a psychosocial risk score for RA, derived from 90 pain-agnostic features that include mental, physical and sociodemographic factors. Pain diagnosis denotes the onset of healthcare diagnosed RA between the initial risk assessment and the online follow-up pain questionnaire roughly 10 years later. Pain outcomes encompass the interference of chronic pain across several dimensions (pain impact), the count of self-reported chronic pain sites (pain spread) and the rating of the worst pain in the last 24 h (pain intensity) in the online pain questionnaire. Arrows are labelled with regression coefficients derived from the structural equation modelling. Arrows are drawn only for significant associations. *$P < 0.0001$; bootstrap (1,000 iterations) tests were used for significance testing of regression coefficients; Med., median; Num., number.

blood-based risk score, the psychosocial risk score and their interaction measured at baseline uniquely contributed to the development of RA over a period of 10 years. Pain outcomes were then extracted from the online pain questionnaire collected in 2019. The model shows that biomarkers, psychosocial risk factors and their combined effects explained unique variance in the development of RA. However, only the RA diagnosis and the baseline-measured psychosocial risk factors were directly linked to pain intensity, spread and disability.

## Discussion

This study aims to identify and compare various biomarkers for chronic pain and its associated conditions using biological measures from different body systems. Overall, we found that biomarkers for self-reported pain performed poorly, in contrast to those targeting the medical conditions underlying the pain that demonstrated strong diagnostic and prognostic potential. Importantly, biological markers and psychosocial factors synergize to significantly influence the prevalence of chronic pain conditions, evident in both cross-sectional and longitudinal assessments. This highlights the importance of adopting a holistic framework to identify biomarkers for chronic pain and contributes to refining the biopsychosocial model for the prediction of future chronic pain conditions.

## Limitations of biomarkers in predicting self-reported pain

One of the main results of our study is that candidate biomarkers for self-reported pain performed poorly. This outcome seems contradictory to the extensive literature associating clinical pain with biological anomalies[13–15], but several factors can explain this discrepancy. First, our findings do not imply the absence of such associations; they instead show that these factors alone are inadequate for predicting chronic pain in new participants. Second, outside the realm of genetics, most previous studies comparing patients with chronic pain with pain-free controls made no distinction between the medical condition causing the pain and the subjective pain report. Although our results show that pain symptoms were most strongly predicted from psychosocial factors, they also show that the underlying condition often could be predicted from biomarkers. Finally, the generalizability of several studies that report the identification of biomarkers for chronic pain may be limited due to their small sample sizes, which can lead to overfitting and inflated effect sizes[16]. This is consistent with recent findings in the UK Biobank, where minor effect sizes (Cohen's $d < 0.15$) were observed in grey matter reduction when large groups of chronic pain patients were compared with those without pain[17]. For these reasons, biological abnormalities documented in the pain literature remain in the realm of explanation rather than prediction[18]. The novelty of our study lies in testing the predictive capacities of various biomarkers in left-out participants. If successful, the identified biomarkers were then further evaluated in a new group of participants outside of the UK Biobank. This is especially important given that accurate predictions of clinical outcomes within trials in which models were developed often showed no better than chance level when tested in out-of-sample patients[19].

The poor performance of biomarkers for self-reported pain is likely to be explained by the heterogeneity of the conditions (for example, ankylosing spondylitis, spine arthritis or sciatica) causing pain at a common body site (for example, back). Yet, pain and its spatial spread were still best predicted from psychosocial factors after selecting a single medical condition for which an effective biomarker was identified (for example, RA, ankylosing spondylitis and Crohn's disease). This highlights the challenges of identifying biomarkers for a subjective measure characterized by nonlinearity between its biological manifestation and its subjective experience. Our results suggest that chronic pain cannot be fully accounted for from the pathogenesis of the medical condition, which aligns with abundant previous research and clinical experience[20,21]. This phenomenon was consistent regardless of the body system used to train the biomarkers or the aetiology of the predicted medical conditions. These findings emphasize that overlooking the role of psychosocial factors would severely constrain our capacity to identify future biomarkers that would generalize across individuals. The limited ability of biological markers, such as brain imaging, for clinical predictions is not exclusive to the field of pain research. The field of psychiatry has encountered a comparable problem with depression[22] for which no clinically useful brain-based biomarkers have yet been identified[23].

Our results do not undermine the importance of subjective pain reports or their clinical relevance; patient-reported pain and psychosocial factors, such as mental and physical health, remain central targets for treatment. However, we found that even effective biomarkers for specific pain diagnoses, such as gout and RA, do not correlate directly with these subjective pain reports (Figs. 5 and 6). Instead, psychosocial factors play a key role in understanding reports of pain severity, impact and spread (Fig. 6), highlighting the often-overlooked importance of psychosocial contexts in which biomarkers are identified.

## Refining biomarker research by targeting pain-related conditions

Nonetheless, efforts in chronic pain research have historically aimed to identify biomarkers specific to clinical pain (that is, presence of chronic pain)[2,14,23]. Our findings suggest we should focus our efforts on

identifying biomarkers generalizable to the medical conditions associated with clinical pain. Chronic pain originates from various mechanisms—nociplastic, neuropathic, inflammatory and nociceptive—each characterized by unique pathological features embedded in different bodily systems[9,24–26]. Mixed pain states further complicate the landscape of chronic pain, as they involve combinations of these mechanisms (for example, nociceptive and neuropathic pain in conditions such as low back pain). Addressing these mixed states requires biomarkers that can accurately capture the interplay of these mechanisms. To enhance prediction accuracy for mixed pain conditions, composite biomarkers that integrate various mechanisms can be utilized. This approach enables the development of biomarkers that address not only specific conditions but also the multifaceted nature of mixed pain states. For instance, we have demonstrated the effectiveness of this approach with a composite blood signature that predicts several inflammatory, neuropathic and nociceptive conditions (Fig. 2), as well as a functional connectivity signature for nociplastic pain (Fig. 3). These examples highlight that biomarkers reflecting shared underlying pathologies, rather than focusing on one specific condition, can provide robust predictive capabilities.

Importantly, this approach does not ignore the subjective experience of pain but rather acknowledges that understanding the biological underpinnings of specific conditions can lead to more precise management and treatment strategies of the pain itself. While our biomarkers are designed to target specific diagnoses, they also enhance our understanding of chronic pain by revealing the pathologies that underlie common pain symptoms. In this way, our findings support a model where biological pathology informs the risk of pain, while psychosocial factors shape its expression. Importantly, our findings demonstrate that these elements do not operate in isolation but interact synergistically, amplifying both the risk of developing pain pathology (as shown in Fig. 5g) and its actual expression (as shown in Fig. 6). This holistic approach allows us to address both the biological and experiential dimensions of pain, offering a more comprehensive understanding of its mechanisms.

Reliable biomarkers for pain-causing pathology were unavailable for most forms of pain conditions[2]. Here, the most consistent predictions were obtained using blood immunoassays, where notably, higher C-reactive protein, neutrophils and cystatin-C and lower lymphocyte percentage, HDL cholesterol and albumin predicted various inflammatory, musculoskeletal and neuropathic pain conditions—a conclusion supported by clinical and research evidence[27–30]. The signature is therefore likely to reflect a systemic state of inflammation or general health deterioration that is unspecific to any individual pain condition but rather characterizes a broader state of pathology. There is precedent for such multidisease models, as indicated by Buergel et al. (2022), who applied deep learning to metabolomics to assess the risk of various common diseases[31]. Here, our study used routine immunoassays—such as C-reactive protein for inflammation, uric acid for metabolic assessments and neutrophil counts for immune response—targeting mechanisms central to pain-associated conditions. We then applied interpretable linear models and encoding maps to specifically trace their impacts on the conditions. Importantly, the composite signature trained to simultaneously predict multiple pain conditions performed well cross-sectionally and longitudinally and generalized in an external large dataset, the AoU. Our results also uncover brain-based markers that were more specific for pain conditions characterized by widespread pain, such as fibromyalgia, chronic fatigue syndrome or pain experienced all over the body. These findings align with the literature suggesting that widespread pain observed in nociplastic conditions arises from altered brain function[12,32]. The structure coefficients of the brain connectivity associated with the prediction of the model revealed lower functional connectivity between a distributed set of brain regions, a phenomenon that has previously been reported in patients with fibromyalgia[33]. The NFS was successfully validated in four external datasets available in the Open Pain repository.

## A holistic approach to chronic pain research

The comprehensive set of biomarkers derived in this study to classify various medical conditions and pain reports highlights the importance of a holistic framework for the prediction of chronic pain. Our results show that biological and psychosocial risk factors additively influence the likelihood of receiving a medical diagnosis, indicating that changes in environmental, behavioural and psychosocial factors either preceding or concurrent with disease onset are crucial for accurately classifying chronic pain conditions. By combining a blood draw measuring the composite signature with a brief questionnaire predicting pain spread such as the Risk of Pain Spreading (ROPS)[10], the holistic approach outlined in this study could be leveraged to substantially improve the prognosis of pain-associated conditions and self-reported pain symptoms such as spread, severity and interference.

### Limitations

Our study has important limitations. First, the UK Biobank lacks diversity, being a predominantly white population of middle-aged and older individuals. This may restrict the applicability of our models, as studies have demonstrated that algorithms trained on mostly white participants can be mischaracterized in non-white participants[34]. This is particularly relevant for our PRS analyses, where individuals of non-European ancestry were excluded from genome-wide association study (GWAS) estimation, a decision informed by the existing literature and genetic makeup of the UK Biobank cohort, in which more than 80% of individuals are of European descent. However, other models in our study, such as those involving blood biochemistry and brain imaging, were validated using external datasets such as All of Us (Fig. 2g) and OpenPain (Fig. 3e), which include more ethnically diverse populations. These validations suggest that the findings from our biochemical and brain-based models are applicable and generalizable across different ethnic backgrounds. In addition, the UK Biobank may have a 'healthy volunteer' selection bias given the low participation of 5.45% (ref. 34). We, however, validated our blood and brain-based biomarkers in independent cohorts with different demographic and general health characteristics (Extended Data Fig. 7d), indicating that our biomarkers were not the result of selection bias or other potential unidentified biases.

## Conclusion

Altogether, our results imply that painful medical conditions can be accurately predicted from the synergy between biological and psychosocial factors, but that pain report was predicted solely from psychosocial factors. Thus, our findings challenge the established idea of a reliable biological marker that could, on its own, detect or predict the subjective experience of chronic pain reported by patients. Instead, our research suggests that future biomarker discovery efforts in chronic pain should incorporate relevant psychosocial factors—such as occupational influences in carpal tunnel syndrome or lifestyle factors in Crohn's disease—into their development protocols. This holistic approach aims to enhance the diagnostic and prognostic utility of biomarkers and support personalized treatment strategies. By integrating biological and psychosocial insights, this comprehensive strategy promises not only to refine diagnostic accuracy but also to identify treatments that address both the mechanistic underpinnings and the psychosocial dimensions of pain, ultimately reducing patient suffering.

## Methods

### Overview of the UK Biobank population

The UK Biobank project is a large-scale, prospective and ongoing study initially established to allow extensive investigation on genetic factors and lifestyle determinants of a diverse range of common diseases of middle-aged and older adults[8]. To recruit the intended sample size of approximately 500,000 participants, over 9 million invitations were sent to individuals registered in the UK National Health Service with an

inclusion age range of 40–69 years old and based on their living within a reasonable distance from an assessment centre. Baseline recruitment and data collection from 503,317 participants who consented to join the study took place between 2006 and 2010 in 22 assessment centres throughout Scotland, England and Wales[35]. All participants gave written, informed consent, and the study was approved by the Research Ethics Committee (REC number 11/NW/0382). Further information on the consent procedure can be found elsewhere. Subsets of baseline participants were invited later for follow-up visits and/or were asked to provide data on certain online questionnaires at certain timepoints. The following datasets from different timepoints are used to address different aims of our study.

### Biological modalities

Broad biological modalities indexing four diverse domains of physiological health were selected for analyses. Biological measures included blood immunoassays, PRS, bone structure estimates derived from dual-energy X-ray absorptiometry scans, and brain imaging phenotypes. For certain analyses, we segmented blood and brain modalities into subcategories. Specifically, blood was divided into three distinct assay types: those assessing inflammatory or immune functions, metabolic functions and haematological functions. Brain imaging was divided into T1-weighted MRI for anatomical insights, diffusion MRI for white matter architecture and resting-state functional MRI (fMRI) for brain connectivity[36]. The availability across study visits, derivation and processing of each modality are detailed in the Supplementary Information.

### Psychosocial phenotypes

We used a truncated set of psychosocial features originally defined in our recent paper (see ref. 10 for details). As the goal of the present study is to compare biological predictors to psychosocial predictors, we excluded biological features included in the original model to generate a strictly psychosocial modality, including body mass index, age, sex, ethnicity, grip strength and blood pressure. Thus, a total of 81 features collected at the baseline and follow-up visits were selected a priori on the basis of their relevance to chronic pain. The selection was based on the Prognosis Research Strategy (PROGRESS) group who recently provided a framework for the development of a prognostic model to determine risk profile[37]. Variables were organized through an iterative approach along a hierarchical framework from 81 features into 8 categories forming three distinct domains (that is, mental health, physical health and sociodemographic). The three domains are as follows.

**Mental health.** The mental health domain includes three categories: (1) neuroticism (all individual items and their total sum score) based on 12 neurotic behaviours closely linked to negative affect, (2) traumas (illness, injury, bereavement or stress in the past 2 years) including six events, and (3) mood (reported frequency of certain moods in the past 2 weeks and visits to a general practitioner or psychiatrist for nerves, anxiety, tension or depression).

**Physical health.** The physical health domain includes three categories: (1) physical activity (metabolic equivalent task scores computed using the International Physical Activity Questionnaire guidelines[38]), (2) sleep (questions regarding duration, napping, snoring and sleeplessness) and (iii) substance use (smoking and alcohol use).

**Sociodemographic.** The sociodemographic domain includes two categories: (1) socioeconomic status (education completed, income, employment and so on), (2) occupational measures (individuals present within household, social entourage and job type (for example, manual or physical job)).

For a detailed view of the biological and psychosocial features included in this study, see Supplementary Tables 1–6.

### Pain phenotypes in the UK Biobank

**One-month pain.** At each study session, participants reported if they experienced pain impacting their usual activities at any major anatomical body sites (head, face, neck or shoulder, back, stomach or abdominal, hip, knee or PAO) in the past month. Participants answering PAO were not allowed to report additional pain locations. This category consists of both chronic and acute pain.

**Acute and chronic pain sites.** Participants reporting pain at a given site in the last month were then asked if the pain at that site had persisted for more than 3 months. This question was used to distinguish between a chronic pain site, one present for more than 3 months, and an acute pain site, one present for 3 months or less, according to the classification from the International Association for the Study of Pain[8].

**Deriving 14 target pain phenotypes.** From these data, we derived 14 pain phenotypes, including a general chronic pain phenotype representing pain at any of the body sites, and a general acute pain phenotype. In addition, we identified seven chronic pain site phenotypes, each representing pain experienced at one of the specified body sites, and four phenotypes that quantified the number of chronic pain sites reported, categorizing the extent of chronic pain spread ranging from 1 to 4 or more distinct sites. Importantly, pain-phenotypic predictive models were trained using data from the same sessions in which pain phenotypes were reported (Fig. 4). For instance, models using blood immunoassays were designed to predict pain reports at the baseline session based on immunoassays collected at baseline. Similarly, models based on brain imaging aimed to predict pain reports from the 9-year follow-up using imaging data collected at that same follow-up. Thus, these models aim to predict the concurrent, or cross-sectional, presence of pain phenotypes.

### Pain-associated diagnoses

Participants' diagnoses were sourced from both self-reported interviews conducted at UK Biobank assessment centres (field IDs 20001 and 20002) and healthcare records provided by the UK National Health Service. In the interview process, trained nurses validated and, if necessary, refined the medical conditions that participants initially reported through a touchscreen questionnaire. In cases of uncertainty, participants described their condition to a nurse, who then assigned a suitable code or logged it as a free-text description. This free text was later matched to a specific entry by a physician.

For health outcomes resulting in hospital admissions, hospital inpatient records utilizing the International Classification of Diseases and Related Health Problems (ICD-10) coded primary or secondary diagnoses (field IDs 41270 and 41271) were available for over 87% of study cohort ($n$ = 446,829). Meanwhile, primary care data, available for 45% of the study cohort ($n$ ≈ 230,000), were obtained from Read Codes v2, as coded by general practitioners (field ID 42040). The UK Biobank additionally provided curated data fields indicating the first occurrence of a set of diagnostic codes (category 1712) for a wide range of health outcomes across self-report, primary care, hospital inpatient data and death data, mapped to a three-digit ICD code. For broader diagnoses that were effectively captured with a three-digit ICD code (for example, RA; ICD codes M05 and M06), we extracted information from this first occurrences database. By contrast, for conditions defined by more granular ICD coding (for example, ankylosing spondylitis; ICD code M081), data were manually curated from self-report and hospital inpatient records.

For our analysis, we selected 35 pain-associated diagnoses based on their high pain prevalence and ample sample size. Criteria included having over 45% pain prevalence (acute or chronic) and an occurrence exceeding 100 participants at the 9-year follow-up. These diagnoses were then aligned with the first occurrences database, ICD-10 codes and primary care data, collating all participants with a record for each

specific diagnosis. For instance, the sciatica group amalgamated both self-reported and healthcare recorded (ICD codes M543 and M544) instances of sciatica. A detailed list of codes for the 35 diagnoses is available in Supplementary Table 7. To ascertain that an illness's onset or diagnosis predated a given study visit, we contrasted the earliest recorded illness date with the participant's respective visit date at each visit (for example baseline, 4-year and 9-year follow-up). This method allowed us to conduct both cross-sectional and longitudinal analyses, examining the presence or onset of diagnoses either concurrent with or following the measurement of predictor variables, as in Fig. 2b. Health records also facilitated time-to-event analyses by providing precise diagnosis dates (Fig. 5g).

To enable comparisons across diagnoses, we defined the healthy control group as participants with no self-reported diagnoses and no self-reported or healthcare recorded instances of the 35 pain-associated diagnoses. This resulted in a healthy control sample size of 103,034 (20.5% of full sample size) participants at the baseline visit and 5,237 (11.6% of full sample size) participants at the 9-year follow-up. As such, the control population may not be representative of the full study population.

Demographic information including sex, ethnicity and age distributions for each diagnosis within the UK Biobank is detailed in Extended Data Figs. 2 and 3.

### Online UK Biobank pain questionnaire
From 2019 to 2020, the UK Biobank conducted follow-up assessments through online questionnaires in a subset of participants originally recruited at baseline. These 'Experience of Pain' questionnaires were administered approximately 8–13 years after the baseline visit (median 10 years) to improve the phenotyping of individuals with chronic pain. Out of the 332,587 participants who received invitations, 167,255 completed the questionnaire. Subsections were used to examine the associations between biological and psychosocial variables recorded at baseline and longitudinal pain outcomes assessed at the online pain questionnaire. These pain outcomes measured several dimensions of chronic pain, such as its interference with daily activities measured via the Brief Pain Inventory, the number of self-reported chronic pain sites to assess pain spread, and the severity of the worst pain experienced in the last 24 h to gauge pain intensity. Detailed information about the questionnaire is available in the UK Biobank documentation (https://biobank.ndph.ox.ac.uk/showcase/ukb/docs/pain_questionnaire.pdf).

### Validation cohorts
To validate the biomarkers identified in this study, data were sourced from two distinct validation cohorts: the AoU and four datasets from the OpenPain repository (Extended Data Fig. 7).

### All of Us Research Program
AoU aims to enroll over one million US participants, prioritizing underrepresented groups in research. AoU collaborates with healthcare organizations to collect and share electronic health records (EHRs), including clinical diagnoses and blood assays. Data are standardized to the Observational Medical Outcomes Partnership Common Data Model, ensuring consistency across different EHR systems[39].

### OpenPain repository datasets
Four resting-state fMRI datasets from OpenPain were combined into a validation cohort of 181 patients with chronic back pain and 69 controls from the UK, Japan and the USA. Preprocessing included motion correction (MCFLIRT), susceptibility distortion correction, T1 registration (FLIRT), alignment to 3-mm Montreal Neurological Institute (MNI) space and removal of physiological and motion noise via signal.clean in Nilearn. Additional steps encompassed connectivity despiking (3DDespike, AFNI), 6-mm smoothing (Nilearn) and resampling to 3-mm resolution. Dynamic functional connectivity was computed using dynamic conditional correlation (DCC) on 279

Brainnetome atlas parcels. Finally, NeuroCombat harmonization was applied to mitigate site-specific effects.

### Statistical analysis
**Machine learning models.** Predictive machine learning models were constructed to classify the 14 pain phenotypes and 35 pain-associated diagnoses from pain-free or diagnosis-free controls, respectively. Machine learning models implemented nested cross-validated logistic regression (SnapML) with a randomized hyperparameter search (scikit-learn) to optimize the ridge regression regularization hyperparameter ('l2') for each individual model. To address cases of class imbalance, the 'class_weight' parameter in the models was set to 'balanced'. This adjustment modifies the loss function penalty to ensure that the majority class does not disproportionately influence the model. Both inner and outer layers of the nested cross-validation (CV) utilized a five-fold strategy (see Extended Data Fig. 1 for a schematic of the modelling pipeline). To prevent data leakage and minimize model bias, feature preprocessing steps including standardization and residualization were fit to the training folds and then applied to the validation fold within each nested CV. Alternative algorithms such as support vector machines and gradient boosting trees were evaluated but were not chosen as the primary machine learning methodology based on their performance metrics (Extended Data Fig. 4).

We trained separate models on 5 distinct modalities—blood, genetics, brain, bone and psychosocial—resulting in 70 pain-phenotype models (14 phenotypes × 5 modalities) and 175 pain-diagnosis models (35 diagnoses × 5 modalities). Additional models were trained on modality subcategories (for example, inflammatory/immune, metabolic, haematological, T1 imaging, diffusion and resting-state connectivity). To account for variability, each model was run five times with unique random states (that is, five times repeated fivefold nested CV), and confidence intervals (CIs) for performance metrics were calculated on the basis of these iterations.

**Mitigating confounding variables.** The influence of confounding variables on machine learning predictions, especially in the context of biological data, is well documented[40–42]. To mitigate potential biases introduced by confounders, we integrated regression-based deconfounding (that is, residualization) within each of our modelling pipelines.

For each CV fold:

(1) Fit a linear regression model on training data (for each modality) to predict confounders.
(2) Apply this model to both training and validation data to remove linear effects of confounders. Then use the resulting residuals as features for model training and evaluation.

For a comprehensive list of the confounders addressed and their handling within each modality, refer to the detailed modality pipeline descriptions in the Supplementary Information.

### Classification of pain phenotypes
The area under the receiver operating characteristic curve (ROC-AUC) scores were calculated from models trained to differentiate between pain phenotypes and pain-free controls (Fig. 4b–d). ROC-AUC scores were obtained from the testing folds of 5 iterations of machine learning models, each using a 5-fold CV strategy, thereby yielding a total of 25 testing metrics per pain endpoint. The reported ROC-AUC scores estimate the effectiveness of the models in distinguishing between participants with a given pain phenotype (that is general chronic pain, general acute pain, chronic pain site or number of chronic pain sites) and those without. CIs at the 95% level were calculated using 1,000 bootstrap resamples of the 25 AUC metrics. In addition, heritability estimates were derived from GWAS of the general chronic pain, acute pain and chronic pain site models.

## Phenotyping and classification of pain-associated diagnoses

We phenotyped pain-associated diagnoses by self-reported chronic pain prevalence, segmenting them by the number of reported pain sites (1 to 4+) and determining pain localization from prevalence at seven body sites. These prevalence rates were then z-scored across conditions for each site, yielding standardized measures of pain distribution (Fig. 1b).

Classification performance for differentiating each diagnosis from diagnosis-free participants followed the same approach used for pain phenotypes. We identified the best-performing biological modality for each diagnosis by highest average AUC, then calculated 95% CIs via 1,000 bootstrap resamples (Fig. 1c). Psychosocial classification AUCs are presented for comparison, and extended results are shown in Extended Data Fig. 3. We additionally trained models on subcategories (for example, inflammatory/immune, metabolic and haematological for blood; T1, diffusion and resting state for brain; mood, neuroticism, life stressors and so on for psychosocial). Bone structure and genetics remained unanalysed at the subcategory level. Within each diagnosis, AUCs were z-scored across subcategories (or entire modalities if not subdivided) to determine whether certain diagnoses reflect domain-specific alterations or broader multidomain changes.

We quantified deviance explained ($D^2$) by biological, psychosocial and combined predictors for two conditions—RA (blood modality) and fibromyalgia (brain modality)—then compared these values with models of self-reported chronic pain. Following a method adapted from Dinga et al.[41], we calculated predicted probabilities from each data source (biological alone, psychosocial alone and combined), fit logistic regressions to each outcome and derived deviance. We then computed the fraction of deviance explained ($D^2$) and partitioned unique ($\Delta D^2_{\mathrm{B}}$, $\Delta D^2_{\mathrm{P}}$) versus shared ($\Delta D^2_{\mathrm{B,P}}$) contributions of biological and psychosocial factors (Fig. 1d). For a detailed explanation of deviance explained calculation, see the Supplementary Information.

## Nociplastic functional signature

We developed a resting-state functional connectivity signature to classify nociplastic pain conditions (fibromyalgia, chronic fatigue syndrome and widespread chronic pain) within the UK Biobank. These conditions share features of central sensitization—including diffuse pain and fatigue[9,25,43–45]—and were combined into a single nociplastic class (n = 535). Logistic regression models trained on resting-state connectivity data separated nociplastic cases from diagnosis-free individuals. From these models, we derived structure coefficients—model encoding maps—indicating how strongly each DCC feature was linked to the predicted nociplastic status. See the Supplementary Information for details on structure coefficient calculation.

**Measuring functional dysconnectivity in nociplastic conditions.** Structure coefficient maps were generated for each model iteration and averaged. To evaluate condition-specific connectivity patterns, we calculated separate maps for fibromyalgia, chronic widespread pain and chronic fatigue syndrome, then correlated them using two-sided Pearson's coefficients with Bonferroni correction. We highlighted the top 25% of node-averaged coefficients on cortical surface renderings (Fig. 3b). Subcortical and brainstem regions were included in the correlation analyses but not displayed on the cortical maps. The same approach was taken to visualize the NFS in Fig. 3c.

**Validation of the NFS.** The NFS was validated in four external datasets available in the OpenPain repository. These datasets were merged to form a validation cohort of 250 participants, allowing an evaluation of the generalizability of NFS in classifying chronic pain. This performance was then benchmarked against the ToPS, a previously validated marker for sustained experimental pain[46]. ToPS weights were applied to the participant-level DCC data (Brainnetome atlas parcellation) in the OpenPain dataset across various thresholds (100%, 50%, 25%, 10%,

5%, 1% and 0.5%) to test for generalizability. The same thresholding procedure was used with the NFS to establish its performance within the OpenPain dataset. For each threshold, we calculated the AUC to differentiate between participants with chronic pain and pain-free individuals. A null AUC distribution for each threshold was generated by randomizing the outcome variable across 1,000 permutations and recalculating the AUC (Fig. 3e).

## Composite blood assay signature

A composite blood assay signature capturing shared alterations across 13 well-predicted diseases (AUC ≥0.70 from their respective blood-based predictive models) was generated by averaging the structure coefficients from each disease model. This yielded a 52-feature map reflecting the direction and magnitude of blood assay features consistently altered in these conditions (Fig. 2c). Subject-level risk scores were derived by taking the dot product of the signature coefficients with each subject's standardized, deconfounded blood assay profile.

**Assessment of the signature.** We evaluated the signature's ability to predict both cross-sectional disease prevalence (at baseline) and future disease incidence (at ~4 and ~9 years) within the UK Biobank. For each of the 13 diseases, participants were grouped as either cases—disease at baseline, disease onset at 4 years or disease onset at 9 years—or timepoint-matched diagnoses-free controls. We then computed Cohen's d (95% CIs derived from 1,000 bootstrap iterations) and AUC for each case–control pairing (Fig. 2b). In parallel, we examined a pooled disease phenotype by aggregating all 13 diagnoses into a single group at each timepoint (Fig. 2d).

**Disease progression analysis.** We further investigated changes in the signature using repeated blood assay measures from the 4-year follow-up (n = 19,360). Signature scores were recalculated using the same coefficients, and a linear mixed-effects model tested the effects of time, disease group and their interaction, with participant as a random factor. Bonferroni-corrected P values were used to assess significant differences between each disease group and controls (Fig. 2e).

**Validation of the signature in the AoU.** A simplified ten-assay version of the composite signature (details in Supplementary Table 1) was applied to the AoU dataset, where participants were classified into pain-diagnosis or healthy control groups based on ICD-10-CM codes. After adjusting each assay for age and sex, we computed subject-level scores and compared them between disease groups and controls via Cohen's d and AUC (95% CIs via 1,000 bootstrap iterations; Fig. 2f,g).

## Biological and psychosocial risk scores

We identified diagnoses with sufficient classification accuracy (AUC ≥0.70) in the blood (13 diagnoses), brain (5 diagnoses) and bone (4 diagnoses) modalities; the PRS modality was excluded due to insufficient performance. Within each modality, biological and psychosocial risk scores were generated by averaging the log-transformed predicted probabilities from the respective models (for example, brain-based models for fibromyalgia, stroke and so on). Participants were then assigned to risk quintiles (low, reduced, neutral, elevated and high) based on these scores (Fig. 5a).

**Biopsychosocial synergy in disease.** In the blood modality analysis, we computed ORs for each of the 13 diagnoses independently against diagnosis-free controls. We determined the OR using the unconditional maximum likelihood estimate, comparing the odds for participants within a specific quintile (for example, 'low') with those of all other participants in the cohort. This approach aimed to quantify the likelihood of having a specific condition (for example, gout) based on a participant's placement in each of the biological and psychosocial risk quintiles. Next, we calculated ORs for each diagnosis based on

combinations of biological and psychosocial quintiles, assessing the likelihood of a diagnosis for participants categorized within both 'low' biological and psychosocial quintiles (LL), 'reduced' (RR) and so forth, up to the 'High' (HH) combination (Fig. 5a). Logarithmic ORs were computed for pooled (combined) diagnoses within each modality, covering all 25 possible combinations of biological and psychosocial risk quintiles. For each modality, Cohen's *d* effect sizes were also calculated, comparing common risk scores for each diagnosis against scores from diagnosis-free individuals (Fig. 5c–e).

**Longitudinal analysis of biopsychosocial synergy.** Within the blood modality, new onset diagnoses (*n* = 1,006) at 4-year follow-up were compared with those remaining diagnosis-free (*n* = 1,282), using baseline biological and psychosocial risk quintiles to compute log-ORs (Fig. 5f). Kaplan–Meier curves illustrated 15-year diagnoses by four baseline risk groups (low–low, high–low, low–high and high–high). Cox-proportional hazards and log-rank tests identified significant differences, focusing on the high–high versus low–high comparison (Fig. 5g).

## Biopsychosocial chronic pain pathway

A structural equation model examined how biological (blood-based) and psychosocial risks, plus their interaction, relate to RA development and subsequent pain outcomes. Pain was characterized using three dimensions from UK Biobank's online questionnaire:

(1) Pain impact: Assessed with the Brief Pain Inventory (BPI-39), the functional impact of pain was evaluated across seven areas (general activity, mood, walking ability, work, interpersonal relations, sleep and life enjoyment), each on a scale from 0 (no interference) to 10 (complete interference)[47].

(2) Pain spread: The extent of pain was quantified by asking participants if they experienced pain or discomfort persistently or intermittently over more than 3 months, followed by specifying the body sites affected in the last three months. A summative phenotype representing the spread of pain was created based on the number of reported pain sites.

(3) Pain intensity: Chronic pain sufferers were prompted to rate their most bothersome pain at its worst in the past 24 hours on a scale from 0 (no pain) to 10 (pain as severe as imaginable).

RA development was modelled as a function of blood risk, psychosocial risk, and their interaction; pain outcomes were modelled as functions of RA, blood risk, and psychosocial risk. Model fitting involved estimating parameters that best reflected the covariances among the observed variables. The fit of the model was assessed using standard indices, including the Comparative Fit Index (CFI), Root Mean Square Error of Approximation (RMSEA), and the Standardized Root Mean Square Residual (SRMR) (Fig. 6).

## Extended data

**Biological amplification in pain spreading.** We derived biological signatures from models predicting the number of self-reported pain sites, ranging from 1 to 4 or more, utilizing structure coefficients from predictive models. In blood modality analyses, we derived structure coefficients for each model to reveal blood assay signature similarity as pain increases in spread. For brain modality studies, we node-averaged the structure coefficients from resting-state connectivity data and visualized the top 25% of absolute values on cortical surfaces. In the bone modality, we averaged coefficients related to bone density, mineral content, and area across different skeletal regions—including the spine, femur, head, legs, pelvis, arms and ribs. In genetic analyses, we identified the top 5% false discovery rate (FDR)-corrected Neuro-Immune Gene Ontology pathways from each PRS model specific to the number of pain sites and categorized these pathways into biological processes using REVIGO (Extended Data Fig. 8)[48].

**Cross-prediction models.** Cross-prediction models were used wherein models initially trained to classify specific diagnoses were tested on alternative diagnoses they were not originally trained to identify. We extracted ROC-AUC scores, sensitivity and specificity from these cross-prediction tasks. Sensitivity and specificity scores were averaged for each original diagnosis across predicted alternative diagnoses within both biological and psychosocial modalities (Extended Data Fig. 5).

**Impact of medication use on the composite blood assay signature.** To assess medication impact on the blood assay signature, we considered 11 medication families linked to the 13 diagnoses comprising the signature (Extended Data Fig. 6). A network model mapped chi-squared associations between diagnoses and medication families, with edge betweenness clustering identifying diagnosis–medication clusters (Extended Data Fig. 6b).

We then recalculated effect sizes and discrimination metrics for the signature, excluding patients taking individual medication families. For instance, statistics were reevaluated after excluding patients on antiepileptic drugs (Anatomical Therapeutic Chemical code N03A). This exclusion was systematically applied to each medication class to assess the signature's predictive accuracy devoid of medication influences (Extended Data Fig. 6c).

## Statistical analysis

Data preprocessing and statistical analyses were performed using Python v.3.8 (including Numpy (v.1.22.0), Pandas (v.1.4.3), Scipy (v.1.10.1), Sklearn (v.1.3.2), Nilearn (v.0.10.0), Lifelines (v.0.26.4), Semopy (v.2.3.9) and SnapML (v.1.9.1)). For cross-sectional analyses involving established diagnoses or pain phenotypes before predictor measurements, we applied L2 (ridge) logistic regression. Longitudinal analyses, where pain diagnoses or phenotypes developed subsequent to predictor measurements, used Cox-proportional hazards models. Nested fivefold CV was used to obtain unbiased model performance results. Permutation tests (with 1,000 iterations) were used to test whether the associations by Pearson's *r* correlation were significantly higher than a null association. We used bootstrap resampling with 1,000 iterations to indicate the estimated error in the Cohen's *d* and ROC-AUC effect sizes. In all analyses, significance was based on $P < 0.05$ for single testing and Bonferonni-corrected *P* value <0.05 for multiple testing. Further details of the statistical methods are specified in each relevant section above.

## Ethical approval

The UK Biobank was approved by the National Information Governance Board for Health and Social Care and the National Health Service North West Multicenter Research Ethics Committee (ref. no. 06/MRE08/65). All participants gave written, informed consent, and the study was approved by the Research Ethics Committee (no. 11/NW/0382). Further information on the consent procedure can be found at https://biobank.ctsu.ox.ac.uk/crystal/field.cgi?id=200. Informed consent for participants in the AoU was obtained either in person or via an eConsent platform, encompassing primary consent along with Health Insurance Portability and Accountability Act (HIPAA) authorization for the research use of EHRs and additional external health data. For each dataset acquired from the OpenPain repository, participants provided written informed consent that authorized the collection of brain imaging, behavioural data and questionnaire responses. Protocols, consent forms and study procedures were approved by McGill Institutional Review Board and/or Douglas Mental Health University Institute Research Ethics Board. This study received ethics approval under institutional review board application number A03-M20-21B (21-03-079).

## Reporting summary

Further information on research design is available in the Nature Portfolio Reporting Summary linked to this article.

## Data availability

All data provided from the UK Biobank are available to other investigators online upon permission granted by https://www.ukbiobank.ac.uk/. Restrictions apply to the availability of these data, which were used under licence for the current study (project ID 20802). The All of Us Research Program data are accessible for individuals at approved institutions. Details can be found at https://www.researchallofus.org/register/. OpenPain data can be accessed openly at https://www.openpain.org/.

## Code availability

Detailed and annotated code is available via GitHub at https://github.com/EVPlab. The medication classification performed by Wu et al.[49] is available in the supplementary data from the original article at https://doi.org/10.1038/s41467-019-09572-5. Code to extract the ToPS by Lee et al.[23] is available at https://cocoanlab.github.io/tops/.

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

## Acknowledgements

This work was supported by the Canadian Institutes of Health Research (RN441786–453096), the Fonds de recherche du Québec en Santé (283687), the Réseau québécois de recherche sur la douleur, and the Louise and Alan Edwards Grants in Pain Research (to E.V.-P.); by the Louise and Alan Edwards Foundation (to M.F.); and by Vanier Scholarship from the Canadian Institutes of Health Research (to C.T.-S.). Additional support was provided by the Pfizer Canada Professorship in Pain Research, the Canadian Excellence Research Chairs grant (CERC09) and the National Institutes of Health (U54 DA049110, to L.D.). This study utilized data from the UK Biobank (project ID 20802), and we gratefully acknowledge the UK Biobank participants and team for making this resource available. Data from the All of Us Research Program were also used; we thank its participants, their families and the researchers and staff who contributed to these valuable cohorts. The funders had no role in the study design, data collection and analysis, decision to publish or preparation of the manuscript.

## Author contributions

M.F. and E.V.-P. conceived the project, designed the study and interpreted the results. M.F. conducted and generated all main analyses and figures, with E.V.-P. supervising the project. M.F., C.T.-S., G.V.G. and A.Z. contributed to data curation, preprocessing and result analysis. M.F., E.V.-P., C.T.-S., M.R., M.O.M. and A.Z. drafted the manuscript, while M.F., E.V.-P., A.Z., C.T.-S. and J.N. reviewed and edited it. M.F. and C.T.-S. organized and analysed the brain imaging data, and M.P. and L.D. organized and analysed the genetic data. L.D. and E.V.-P. secured access to the UK Biobank and obtained the necessary funding. M.F. and A.B. facilitated access to and curated the All of Us dataset. M.W. and J.P. provided continuous external feedback and expertise as pain specialists. All authors contributed to reviewing and editing the manuscript and approved the final version.

## Competing interests

The authors declare no competing interests.

## Additional information

**Extended data** is available for this paper at https://doi.org/10.1038/s41562-025-02156-y.

**Correspondence and requests for materials** should be addressed to Matt Fillingim or Etienne Vachon-Presseau.

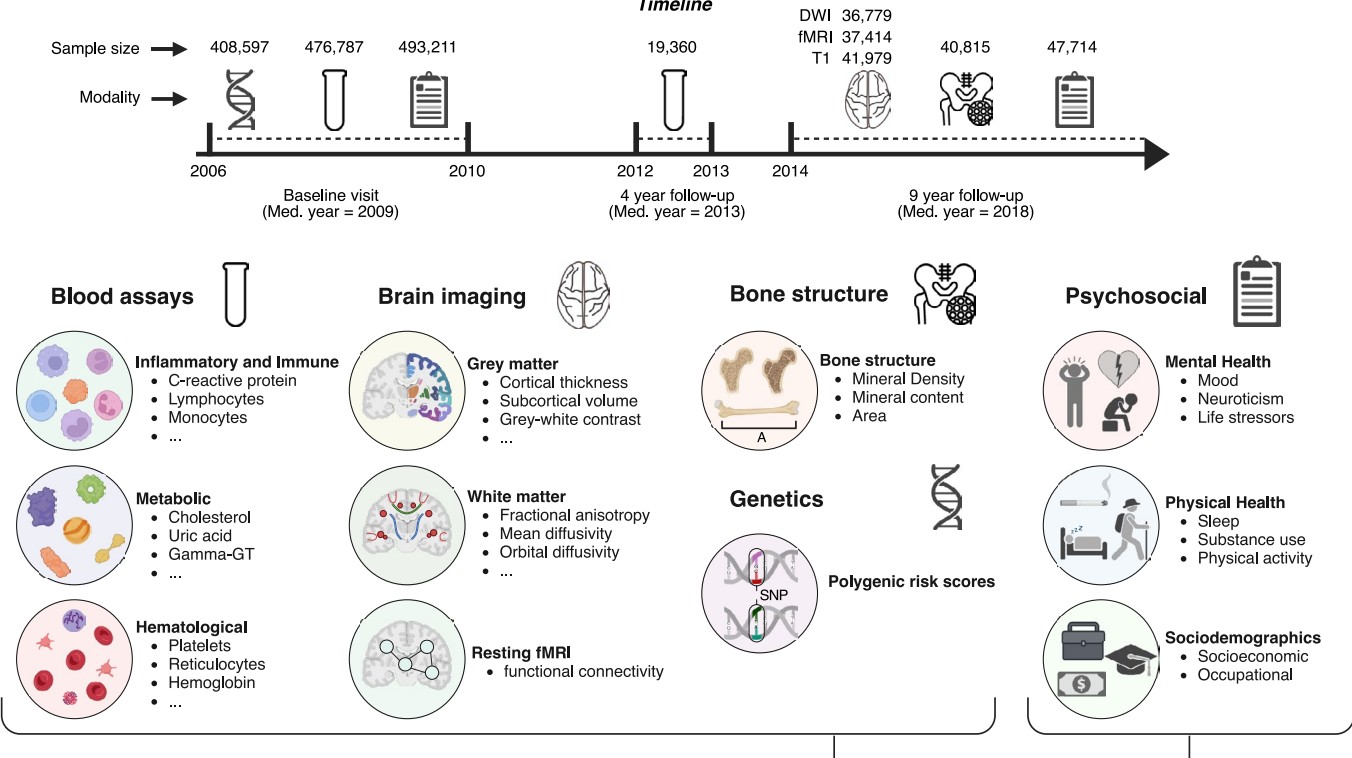

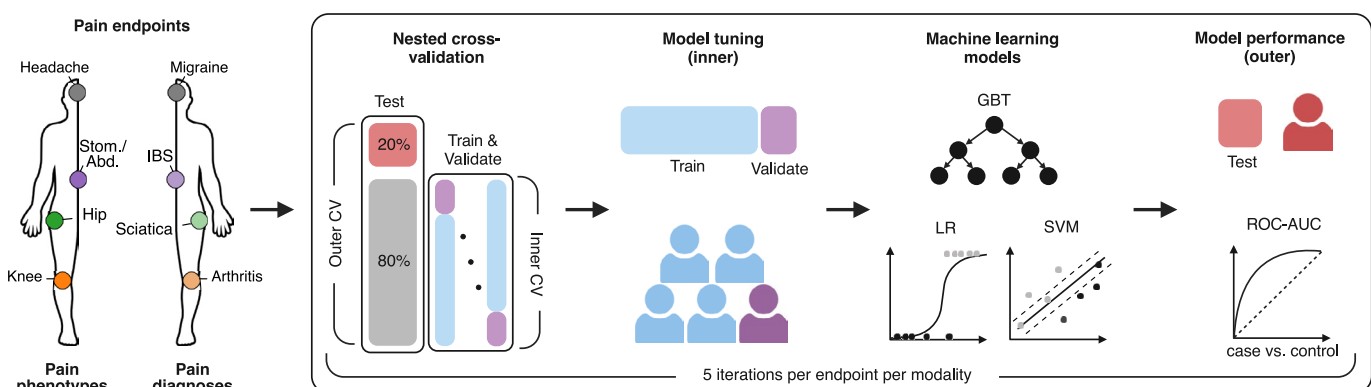

**Extended Data Fig. 1 | Candidate modalities and machine learning pipeline.**
**a**, Top: A timeline depicts the collection of biological and psychosocial modalities across 3 different UK Biobank study sessions, with the number of samples available for analysis after data cleaning indicated above each modality icon. Below: Each modality is broken down into subcategories where applicable, with a description of example measures within these subcategories. Modalities are highlighted in large, bold font; subcategories in smaller bold font; and example measures are listed in regular font with bullets. **b**, A schematic outlines the machine learning pipeline employed to evaluate the ability of candidate modalities to classify pain endpoints. A nested Cross-Validation (CV) approach with 5-fold inner and 5-fold outer CV was utilized to optimize model performance

without data leakage. The inner loop optimizes performance by training a model on each training fold and tuning hyperparameters on the validation fold to maximize the score. In the outer loop, the model's generalizability is gauged by averaging the scores across left-out test sets. Three machine learning algorithms we're assessed: gradient boosting trees, logistic regression, and linear support vector machines. This process was iterated five times for each modality, with participant order randomized in each iteration to prevent model performance bias based on train/test participant arrangement. GBT, Gradient boosting trees; LR, Logistic regression; SVM, Support vector machine; ROC-AUC, Receiver operating characteristic area under the curve.for age distribution, both for the full cohort and stratified by each diagnosis.

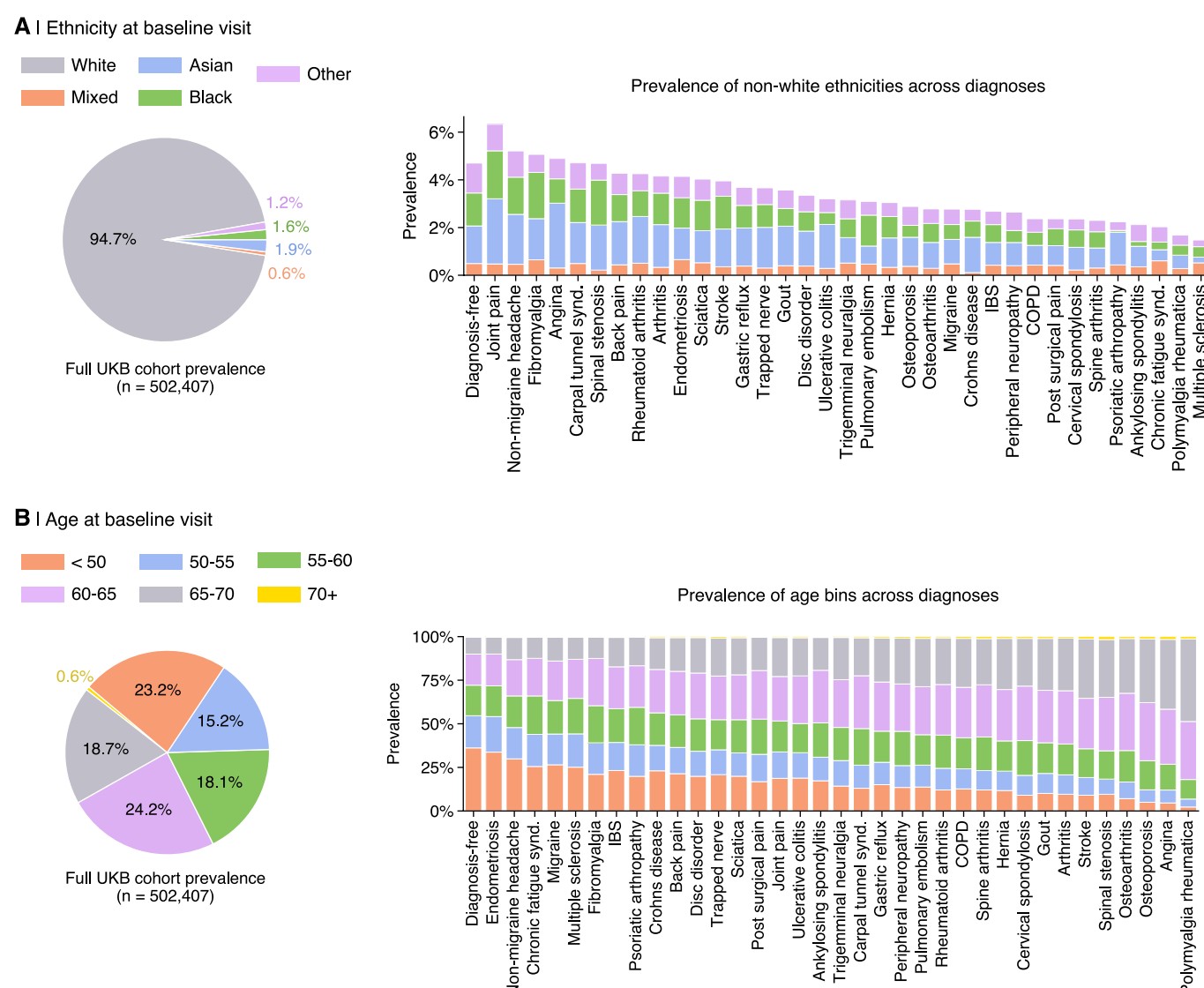

**Extended Data Fig. 2 | Ethnicity and age in the UK Biobank. a**, Ethnicity prevalence for the entire UK Biobank cohort (n = 502,407) is shown in the pie chart, with a breakdown of on non-white ethnicity prevalence across each pain-associated diagnosis shown using stacked barplots. In part **b**, data are visualized for age distribution, both for the full cohort and stratified by each diagnosis.

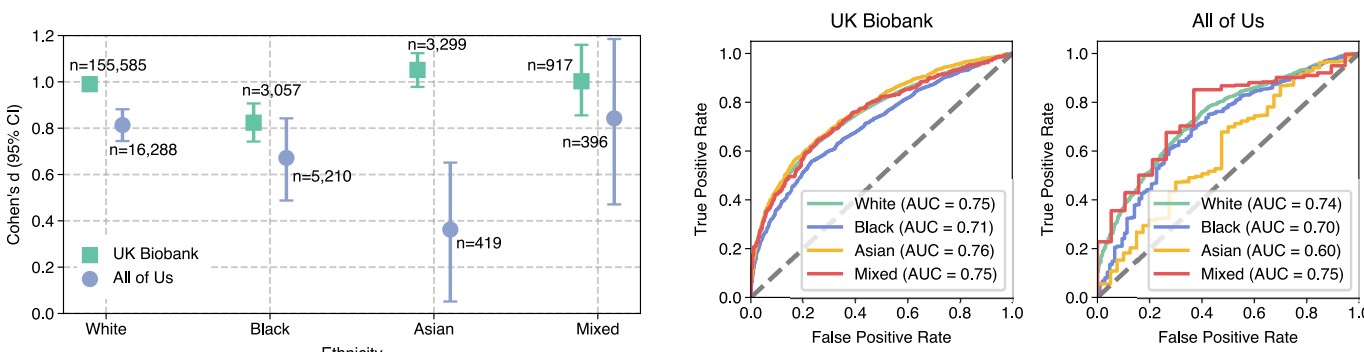

**A | Sex stratified prevalence and model performance in UKB**

**B | Composite blood signature performance by ethnicity**

**Extended Data Fig. 3 | See next page for caption.**

**Extended Data Fig. 3 | Performance of Pain-Diagnosis Classification Models, Stratified by Sex and Ethnicity. a**, The stacked barplot shows the sex prevalence for each pain-associated diagnosis assessed, with the overall UK Biobank cohort sex prevalence indicated by a dotted line. Models trained on both males and females were evaluated separately in males and females, using ROC-AUC scores from cross-validation testing folds. These scores depict the 95% confidence interval (1,000 bootstrap resamples) from 5 iterations of 5-fold CV for each model. Two-sided Wilcoxon signed-rank tests assessed significant performance differences between sexes, with results Bonferroni corrected for multiple comparisons. In the visualization, orange dots represent classification performance in females, blue dots denote performance in males, and green dots reflect the performance on both sexes (full model). **b**, Evaluation of the composite blood assay signature for 13 pooled pain diagnoses was conducted separately across racial groups (white, black, Asian, and mixed) within both the UK Biobank and the All of Us cohort, using Cohen's d (left) and ROC curves (right). The forest plot displays mean Cohen's d values, with error bars showing the 95% confidence interval, derived from 1,000 bootstrap resamples. The specific diagnoses included: Gout, Polymyalgia rheumatica, Stroke, Crohn's disease, Angina, Rheumatoid arthritis, Psoriatic arthropathy, Peripheral neuropathy, Ankylosing spondylitis, Carpal tunnel syndrome, Pulmonary embolism, Ulcerative colitis, and Arthritis.

**A** | Comparative performance of algorithms in pain diagnosis classification

**B** | Influence of confounders in brain-based classification of diagnoses

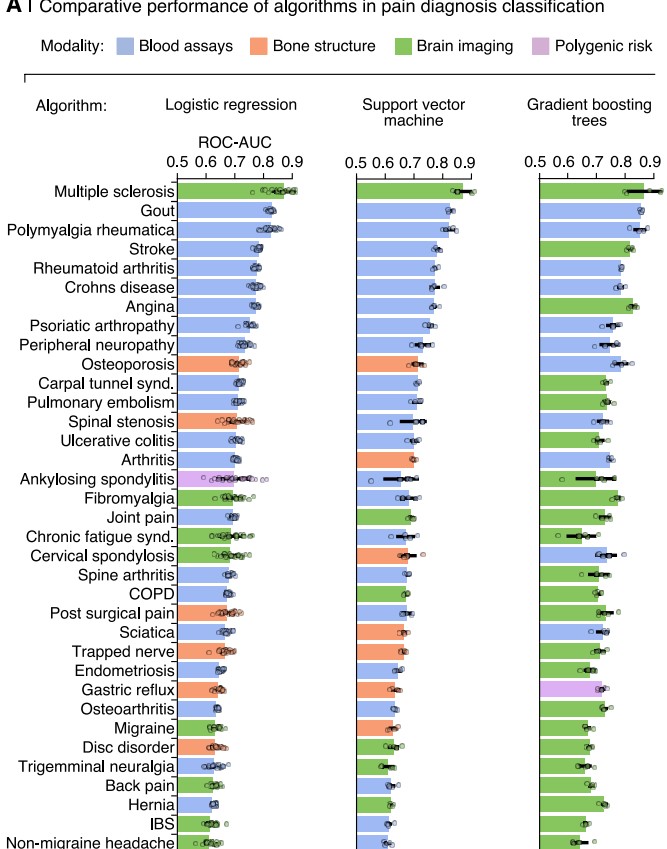

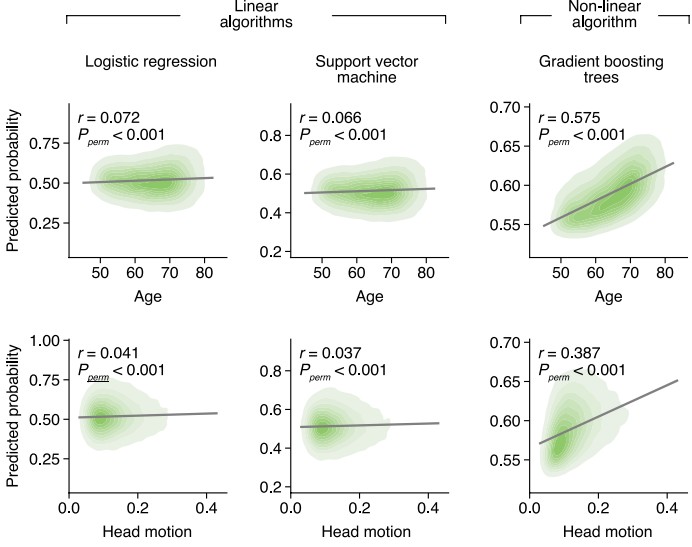

**C** | Influence of confounders in blood-based classification of diagnoses

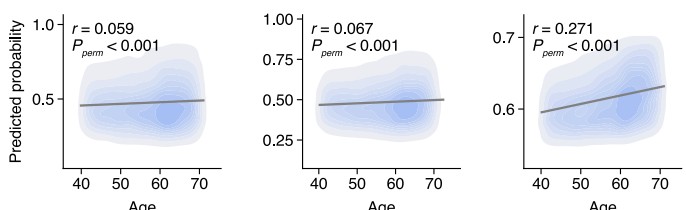

**D** | Comparative performance of algorithms in chronic pain classification

**E** | Influence of confounders in brain-based classification of chronic pain

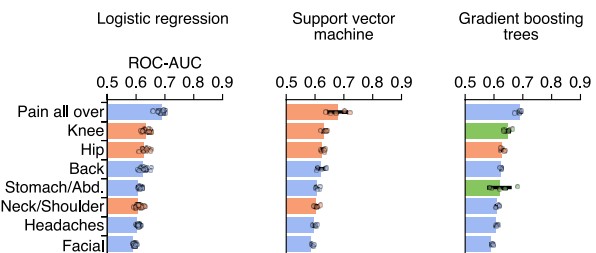

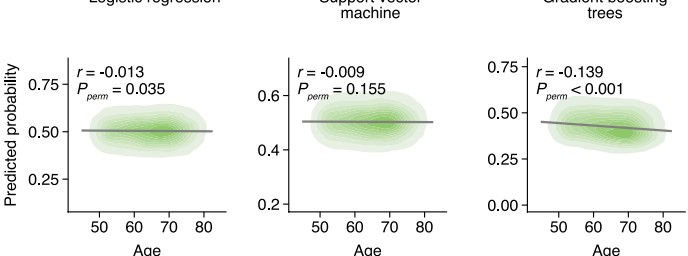

**Extended Data Fig. 4 | Evaluating alternative classification algorithms. a,** Bar plots depict mean test set Receiver Operating Characteristic Area Under the Curve (ROC-AUC) scores for three candidate algorithms—logistic regression, support vector machine (SVM), and gradient boosting trees—in classifying pain-associated diagnoses. Error bars represent the standard deviation across 5-fold cross-validation. Overlaid points indicate AUC scores from individual validation folds (n = 25 for logistic regression; n = 5 for SVM and gradient boosting trees). ROC-AUC scores are color-coded according to the modality that most accurately predicted each outcome, with results arranged based on the performance of the logistic regression model. **b, c, e,** Regression density plots illustrate the association between the predicted probability of a diagnosis (**b, c**) or chronic pain site (**e**) (averaged across all models within a given modality) and confounding factors such as age and/or head motion. These associations are quantified using two-sided Pearson's correlations, with significance assessed through 1,000 permutation tests. **d,** Performance (ROC-AUC) for self-reported chronic pain body sites is shown, with error bars calculated as in **a**.

**A** | Cross-prediction models

**B** | Cross-prediction ROC-AUC scores

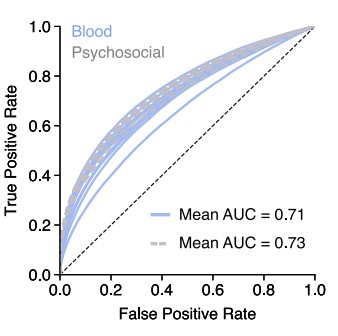

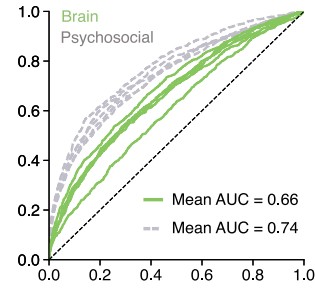

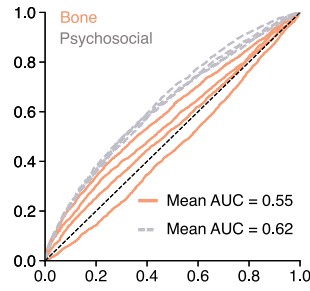

**C** | Cross-prediction sensitivity x specificity analysis

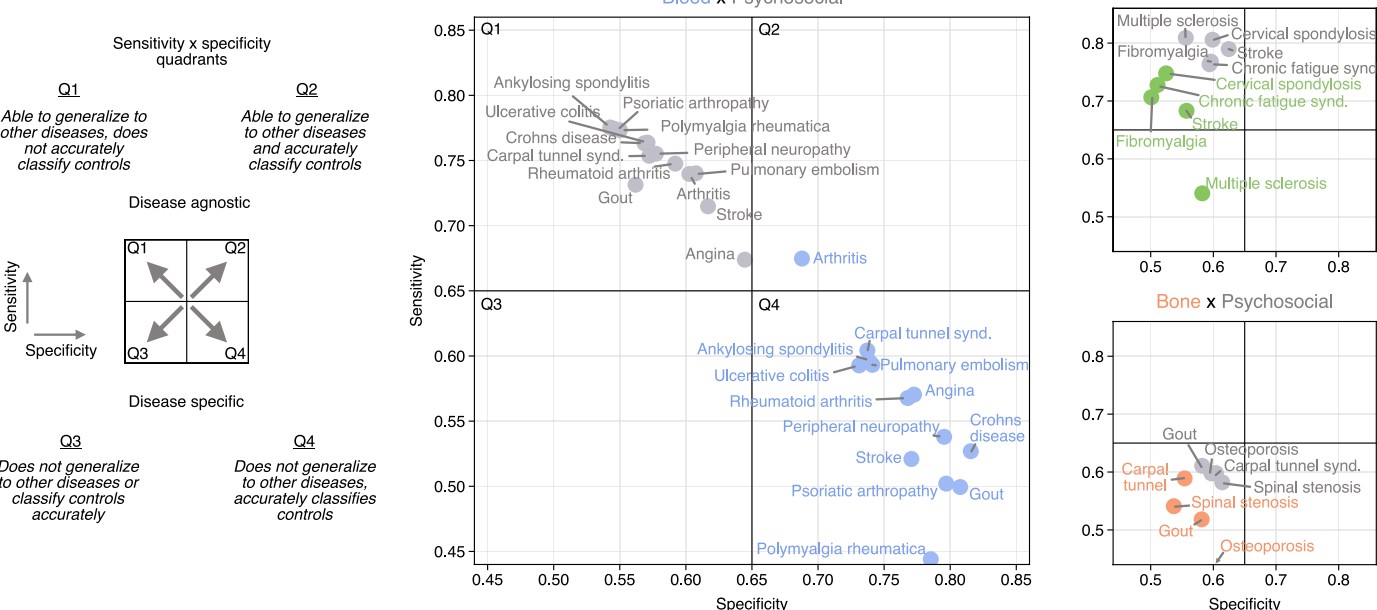

**Extended Data Fig. 5 | Biological and psychosocial cross-prediction models for pain diagnoses. a**, Models trained on specific diagnoses were evaluated for their ability to predict other diagnoses that they weren't trained on based on averaged performance metrics (ROC-AUC, sensitivity, specificity) across untrained diagnoses. This cross-prediction analysis was conducted for the diagnoses that were most accurately classified using blood, brain, and bone modalities alongside psychosocial models. **b**, Average cross-prediction ROC-AUC curves are displayed for both biological and psychosocial models within each biological modality. **c**, Quadrant plots show the average sensitivity and specificity of cross-prediction for each diagnosis. Points within the plots are color-coded by modality and labeled by the diagnosis on which the model was trained. Here, sensitivity measures the model's accuracy in detecting untrained diagnoses, while specificity gauges its precision in identifying diagnosis-free controls.

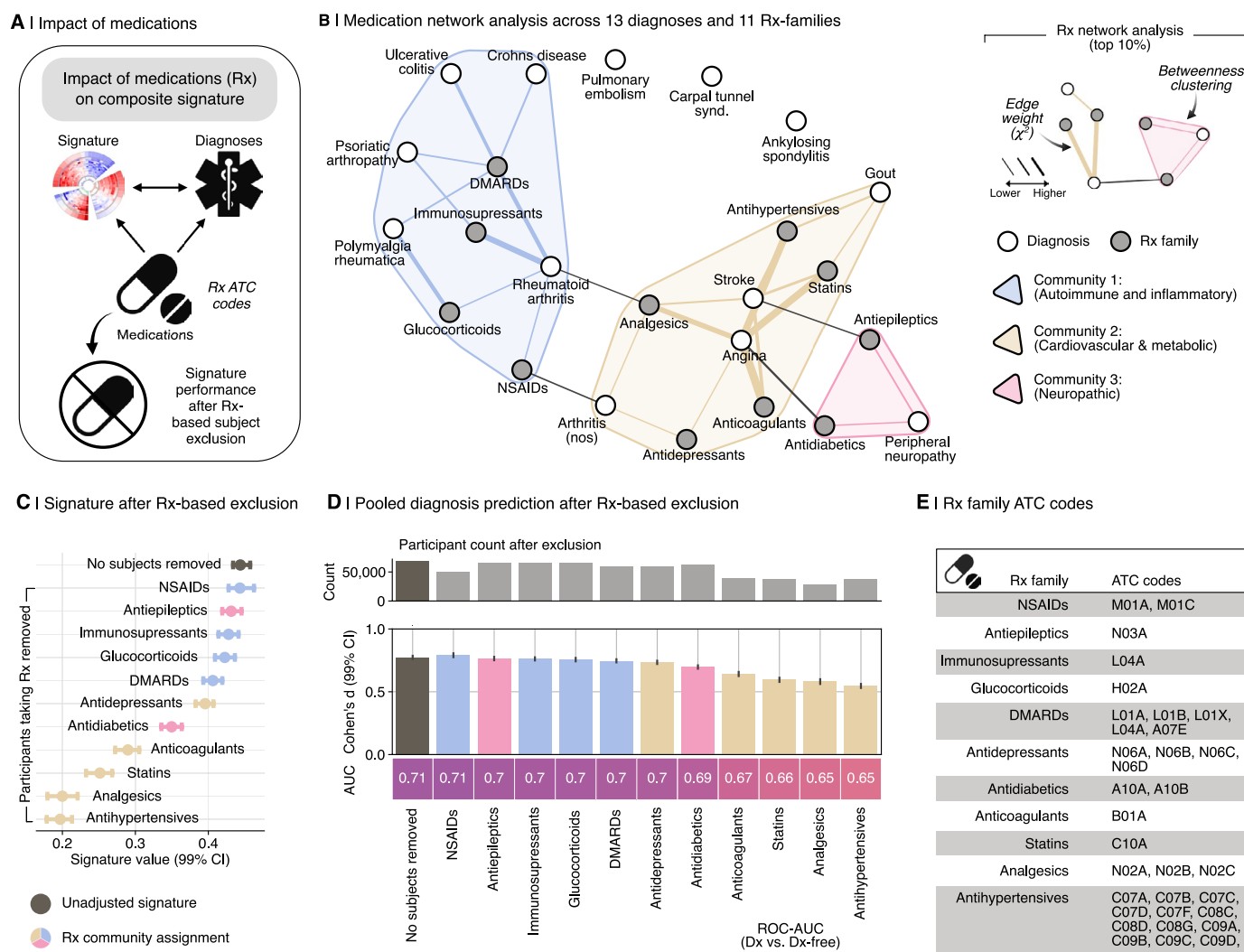

**A** | Impact of medications

**B** | Medication network analysis across 13 diagnoses and 11 Rx-families

**C** | Signature after Rx-based exclusion

**D** | Pooled diagnosis prediction after Rx-based exclusion

**E** | Rx family ATC codes

| Rx family | ATC codes |
|---|---|
| NSAIDs | M01A, M01C |
| Antiepileptics | N03A |
| Immunosupressants | L04A |
| Glucocorticoids | H02A |
| DMARDs | L01A, L01B, L01X, L04A, A07E |
| Antidepressants | N06A, N06B, N06C, N06D |
| Antidiabetics | A10A, A10B |
| Anticoagulants | B01A |
| Statins | C10A |
| Analgesics | N02A, N02B, N02C |
| Antihypertensives | C07A, C07B, C07C, C07D, C07F, C08C, C08D, C08G, C09A, C09B, C09C, C09D, |

**Extended Data Fig. 6 | Impact of medications (Rx) on composite blood signature performance. a**, Schematic of the medication-based exclusion approach to assess medication impact on the composite signature. **b**, Network analysis using edge-betweenness clustering on chi-squared values shows medication-diagnosis communities across 13 diagnoses and 11 medication families. **c**, Adjusted mean composite signature values for diagnosed participants, excluding those taking specific medications, are shown with error bars representing 99% confidence intervals (CI) estimated from 1,000 bootstrap

samples. **d**, The composite signature's diagnostic performance after medication-based exclusion is evaluated for classifying 13 diagnoses. Performance metrics include Cohen's *d* and ROC-AUC, comparing diagnosed individuals to diagnosis-free controls. Bars represent the mean Cohen's *d*, with error bars depicting 99% CI, estimated from 1,000 bootstrap samples. **e**, Medication (Rx) families are organized by their corresponding Anatomical Therapeutic Chemical (ATC) classification codes and displayed in a table format.

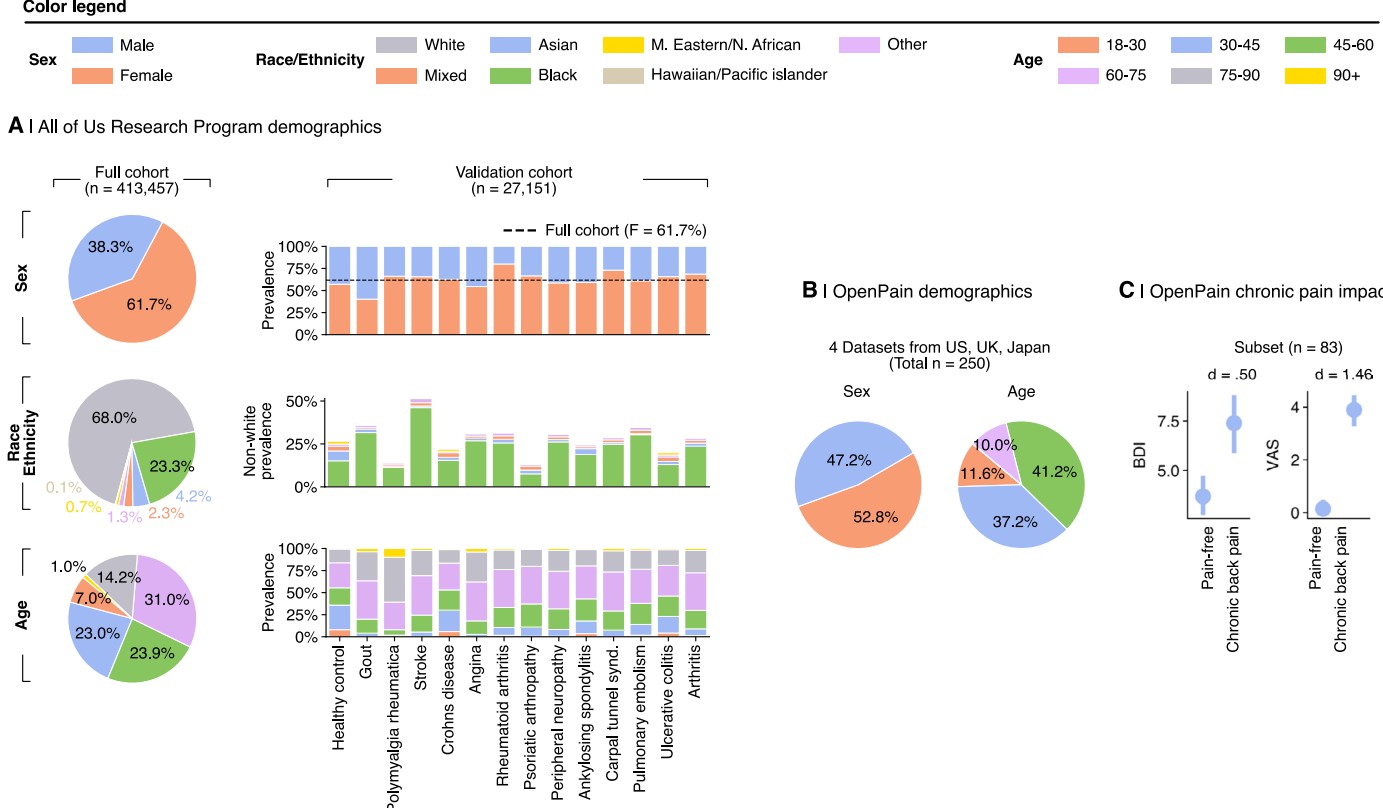

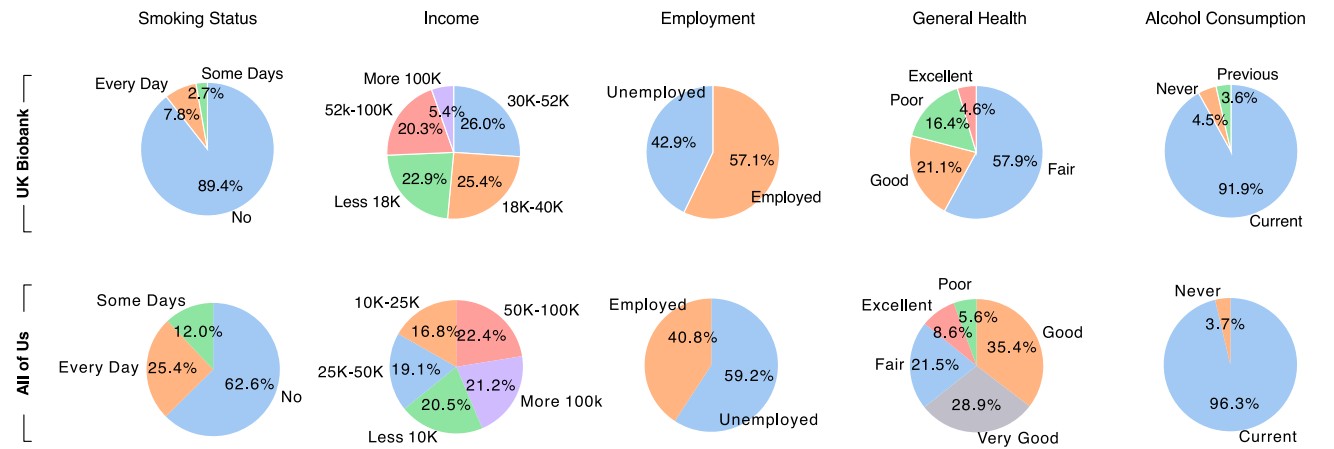

**Extended Data Fig. 7 | Validation datasets demographics.** Top: Color legend indicates the categories for the three demographic dimensions analyzed in each validation dataset (sex, race/ethnicity, and age). **a**, Demographics for the All of Us Research Program (AoU) are shown with pie charts for the entire cohort and stacked barplots for the subset of participants used for validation of the composite blood signature, categorized by each diagnosis and the healthy control group. Demographics are similarly depicted for the OpenPain datasets **b**, Ethnicity/race data were not available for the OpenPain datasets. **c**, Cohen's

d analysis shows the Cohen's d effect size of pain impact comparing patient and control groups within a subset of OpenPain, measured by the Brief Depression Inventory (BDI) and Visual Analogue Scale (VAS) for pain intensity in the last 24 hours. Error bars indicate the 95% confidence interval, estimated from 1,000 bootstrap samples. Statistical significance is determined use a two-sided Wilcoxon rank-sum test (all P < 0.001). **d**, Demographic and general health characteristics within the UK Biobank (Top row) and All of Us Research Program (Bottom row) are shown with pie charts for the entire cohorts.

**A** | Examing biological signatures across pain spread

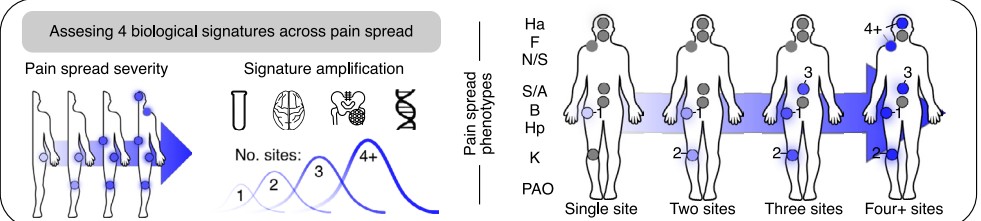

**B** | Amplification of blood biomarkers

**C** | Amplification of functional disconnectivity

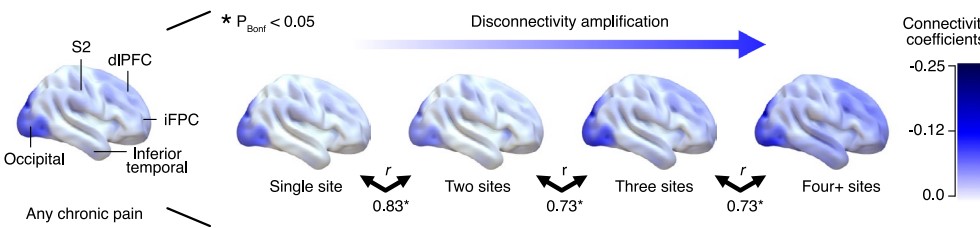

**D** | Reduction in bone density, content, and area in multisite pain

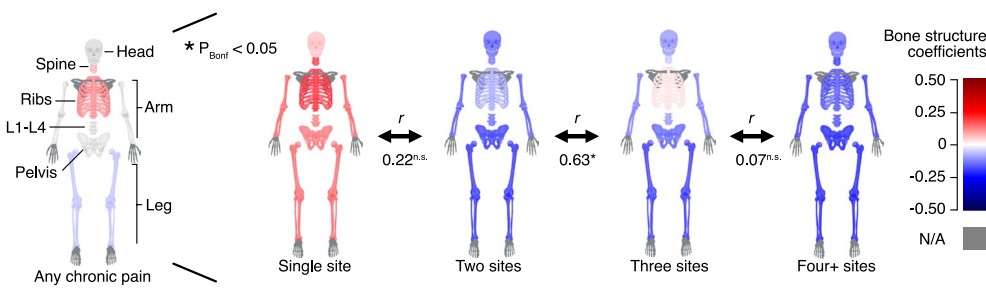

**E** | Amplification of genetic pathways

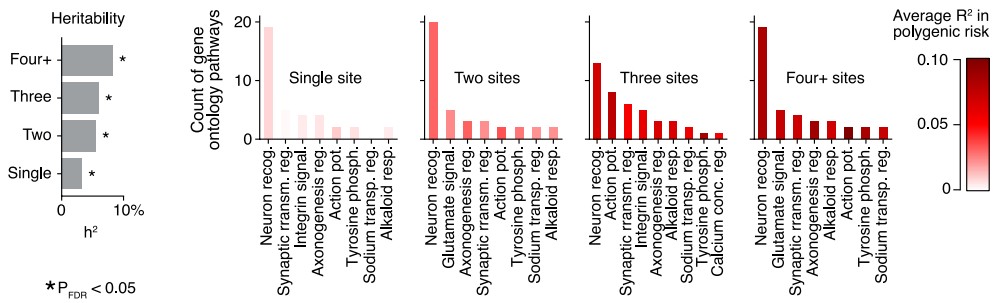

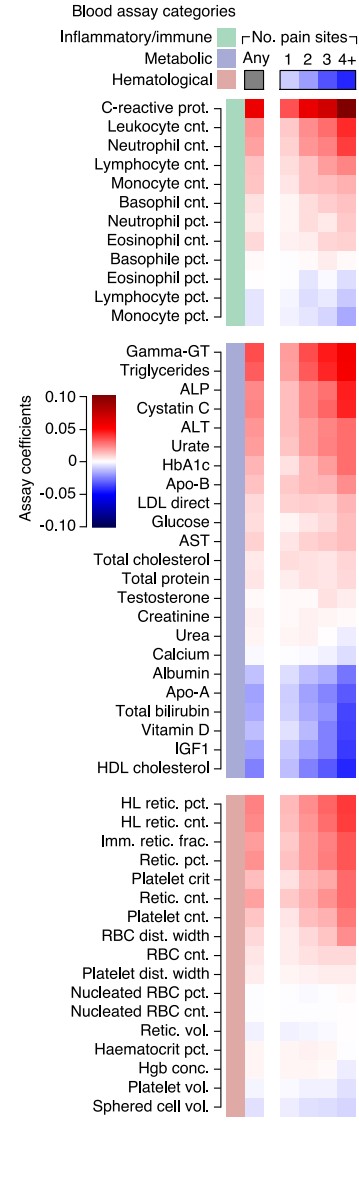

**Extended Data Fig. 8 | Biological pattern amplification in chronic pain spread.**
**a**, Schematic of chronic pain spreading, quantified by the total number of chronic pain sites, alongside the investigated biological modalities: blood, brain, bone, and genetics. Biological patterns, represented by structure coefficients, associated with pain spread severity were derived from models distinguishing between levels of pain spread severity, ranging from low (one chronic pain site versus pain-free) to high (four or more chronic pain sites versus pain-free). **b**, Structure coefficients from models are shown for each level of pain spread, ordered by severity and segmented by subcategory. Additionally, coefficients for a generalized chronic pain model, encompassing any number of pain sites, are also shown. **c**, Cortical surface renderings visualize resting functional connectivity, thresholded to highlight the top 25% of structure coefficients, which represent the sum of dynamic conditional correlation across brain parcels.

Arrows interlinking the cortical renderings depict the association (two-sided Pearson correlation, all $P < 0.001$ bonferonni corrected) between the complete unthresholded vectors of structure coefficients (parcel to parcel connectivity) for adjacent levels of pain spread. **d**, Skeletal body maps show bone segments colored based on structure coefficients for each assessed bone system, using DXA-derived bone density, mineral content, and area estimates. For each spreading level, averages of these three estimates for each bone system (e.g., head, spine, leg) are depicted on the maps. **e**, Heritability estimates derived from polygenic risk scores (PRS) are shown. These heritability estimates are significant according to a two-sided, FDR-corrected Wald Test (all $P < 0.001$). The top 5% of FDR-corrected gene ontology pathways for each PRS are tallied and organized by biological process. The corresponding bars are shaded based on the average explained variance ($R^2$) across pathways within each biological process.

# Reporting Summary

## Statistics

For all statistical analyses, confirm that the following items are present in the figure legend, table legend, main text, or Methods section.

| n/a | Confirmed | |
|---|---|---|
| ☐ | ☒ | The exact sample size (*n*) for each experimental group/condition, given as a discrete number and unit of measurement |
| ☐ | ☒ | A statement on whether measurements were taken from distinct samples or whether the same sample was measured repeatedly |
| ☐ | ☒ | The statistical test(s) used AND whether they are one- or two-sided<br>*Only common tests should be described solely by name; describe more complex techniques in the Methods section.* |
| ☐ | ☒ | A description of all covariates tested |
| ☐ | ☒ | A description of any assumptions or corrections, such as tests of normality and adjustment for multiple comparisons |
| ☐ | ☒ | A full description of the statistical parameters including central tendency (e.g. means) or other basic estimates (e.g. regression coefficient) AND variation (e.g. standard deviation) or associated estimates of uncertainty (e.g. confidence intervals) |
| ☐ | ☒ | For null hypothesis testing, the test statistic (e.g. *F*, *t*, *r*) with confidence intervals, effect sizes, degrees of freedom and *P* value noted<br>*Give P values as exact values whenever suitable.* |
| ☒ | ☐ | For Bayesian analysis, information on the choice of priors and Markov chain Monte Carlo settings |
| ☒ | ☐ | For hierarchical and complex designs, identification of the appropriate level for tests and full reporting of outcomes |
| ☐ | ☒ | Estimates of effect sizes (e.g. Cohen's *d*, Pearson's *r*), indicating how they were calculated |

*Our web collection on statistics for biologists contains articles on many of the points above.*

## Software and code

Policy information about availability of computer code

| Data collection | No software was used for data collection in this study. Data was sourced from three publicly available datasets: the UK Biobank, the All of Us Research Program (AoURP), and OpenPain repository. Details regarding data acquisition and collection methods for these datasets have been documented in previous work. |
|---|---|
| Data analysis | Data pre-processing and statistical analyses were performed using Python v.3.8 (including Numpy (v.1.22.0), Pandas (v.1.4.3), Scipy (v.1.10.1), Sklearn (v.1.3.2), Nilearn (v.0.10.0), Lifelines (v.0.26.4), Semopy (v.2.3.9), and SnapML (v.1.9.1)). Manuscript analysis code of the main figures will be available upon publication (https://github.com/EVPlab). |

For manuscripts utilizing custom algorithms or software that are central to the research but not yet described in published literature, software must be made available to editors and reviewers. We strongly encourage code deposition in a community repository (e.g. GitHub). See the Nature Portfolio guidelines for submitting code & software for further information.

## Data

Policy information about availability of data

All manuscripts must include a data availability statement. This statement should provide the following information, where applicable:
- Accession codes, unique identifiers, or web links for publicly available datasets
- A description of any restrictions on data availability
- For clinical datasets or third party data, please ensure that the statement adheres to our policy

All source data is available (upon data access application) from the UK Biobank. See https://www.ukbiobank.ac.uk/enable-your-research/apply-for-access for an application. The AoURP cohort can be obtained by registration: www.researchallofus.org. The OpenPain data is accessible through https://www.openpain.org/

## Research involving human participants, their data, or biological material

Policy information about studies with human participants or human data. See also policy information about sex, gender (identity/presentation), and sexual orientation and race, ethnicity and racism.

| | |
|---|---|
| Reporting on sex and gender | The sex of participants, as reported in the UK Biobank, was analyzed in relation to various biomarkers and psychosocial variables. We tested the generalizability of our models separately for males and females. Our analysis identified sex-specific effects in certain painful medical conditions associated with specific biomarkers. These findings are reported alongside other demographic variables, which are also presented separately. It is important to note that no information regarding gender identity was collected. |
| Reporting on race, ethnicity, or other socially relevant groupings | The study cohort consists of the general population from the United Kingdom, aged 40-70 years at the baseline visit, with 51-55% of participants being female and predominantly of white ethnicity (94-96%). After stratifying by painful medical conditions, the sample sizes were too small to estimate reliable effects to determine if models perform differently across ethnicities. For the replication cohorts (All of Us Research Program [AoURP] and OpenPain), demographics are more representative, and the models achieved comparable effect sizes to those in the UK Biobank. We report the age, ethnicity, and sex characteristics for each visit from each cohort. |
| Population characteristics | The study cohort consists of the general population from the United Kingdom, aged 40-70 years at the baseline visit, with 51-55% of participants being female and predominantly of white ethnicity (94-96%). Participants provided electronic health records that detail current and past clinical diagnoses or provided self-reported medical history. In all machine learning models, age and sex were controlled to reliably estimate the biological effects stemming from the disease process. |
| Recruitment | *Describe how participants were recruited. Outline any potential self-selection bias or other biases that may be present and how these are likely to impact results.* |
| Ethics oversight | All participants provided written, informed consent, and the study was approved by the Research Ethics Committee (REC number 11/NW/0382). Further information on the consent procedure can be found elsewhere (https://biobank.ndph.ox.ac.uk/ukb/field.cgi?id=200). |

Note that full information on the approval of the study protocol must also be provided in the manuscript.

# Field-specific reporting

Please select the one below that is the best fit for your research. If you are not sure, read the appropriate sections before making your selection.

☒ Life sciences ☐ Behavioural & social sciences ☐ Ecological, evolutionary & environmental sciences

For a reference copy of the document with all sections, see nature.com/documents/nr-reporting-summary-flat.pdf

# Life sciences study design

All studies must disclose on these points even when the disclosure is negative.

| | |
|---|---|
| Sample size | No sample-size calculation was done. A total of 493,211 participants were included in this study, using the UK Biobank. For the replication cohorts, the AoURP had a total sample size of 27,151 and OpenPain had a total sample of 250 participants. |
| Data exclusions | In the UK Biobank, participants with more than 50% of missing features within each biological and psychosocial modality were excluded. No other exclusion criteria were applied to ensure that the study findings would be as generalizable as possible to the wider population. For the All of Us Research Program (AoURP), participants were excluded if they had missing assays from the routine composite blood panel or if they lacked diagnoses of any of the 13 conditions included in the composite blood assay signature. A healthy control group was created by selecting participants with no clinical records of inflammatory, musculoskeletal, or cardiovascular diagnoses. In the OpenPain project, participants were excluded if they had corrupt imaging data or if they did not complete the pain questionnaire. |
| Replication | All models were derived and validated within the UK Biobank cohort using nested cross-validation to ensure robustness. For select biomarkers, exact replications were conducted in additional cohorts: the All of Us Research Program (AoURP) for the blood-based biomarker (n=27,151) and OpenPain for the brain imaging biomarker (n=250). |

| Randomization | The study design did not include randomization of participants. |
|---|---|
| Blinding | Blinding was not necessary for this study, as the determination of feature importance was conducted algorithmically through the machine learning models. For the routine panel replication in the All of Us Research Program, we selected the top 10 assays based on their feature importance as assigned by the model developed using the UK Biobank data. |

# Reporting for specific materials, systems and methods

We require information from authors about some types of materials, experimental systems and methods used in many studies. Here, indicate whether each material, system or method listed is relevant to your study. If you are not sure if a list item applies to your research, read the appropriate section before selecting a response.

## Materials & experimental systems

| n/a | Involved in the study |
|---|---|
| ☒ | ☐ Antibodies |
| ☒ | ☐ Eukaryotic cell lines |
| ☒ | ☐ Palaeontology and archaeology |
| ☒ | ☐ Animals and other organisms |
| ☒ | ☐ Clinical data |
| ☒ | ☐ Dual use research of concern |
| ☒ | ☐ Plants |

## Methods

| n/a | Involved in the study |
|---|---|
| ☒ | ☐ ChIP-seq |
| ☒ | ☐ Flow cytometry |
| ☐ | ☒ MRI-based neuroimaging |

## Plants

| Seed stocks | *Report on the source of all seed stocks or other plant material used. If applicable, state the seed stock centre and catalogue number. If plant specimens were collected from the field, describe the collection location, date and sampling procedures.* |
|---|---|
| Novel plant genotypes | *Describe the methods by which all novel plant genotypes were produced. This includes those generated by transgenic approaches, gene editing, chemical/radiation-based mutagenesis and hybridization. For transgenic lines, describe the transformation method, the number of independent lines analyzed and the generation upon which experiments were performed. For gene-edited lines, describe the editor used, the endogenous sequence targeted for editing, the targeting guide RNA sequence (if applicable) and how the editor was applied.* |
| Authentication | *Describe any authentication procedures for each seed stock used or novel genotype generated. Describe any experiments used to assess the effect of a mutation and, where applicable, how potential secondary effects (e.g. second site T-DNA insertions, mosiacism, off-target gene editing) were examined.* |

## Magnetic resonance imaging

### Experimental design

| Design type | UK Biobank brain imaging resting-state functional MRI scans |
|---|---|
| Design specifications | Single 6-minutes resting-state run, eyes open. T1 susceptibility-weighted structural imaging. Diffusion weighted imaging. |
| Behavioral performance measures | The number of self-reported pain sites, specific pain body sites, and 35 distinct pain-associated medical conditions. |

### Acquisition

| Imaging type(s) | UK Biobank brain imaging data: structural (T1 susceptibility-weighted), diffusion weighted, and resting-state functional scans - See Methods for details. |
|---|---|
| Field strength | 3T |
| Sequence & imaging parameters | Please see Miller et al., Nature Neuroscience 2016 for a full list of the imaging parameters. |
| Area of acquisition | Whole brain |
| Diffusion MRI | ☒ Used    ☐ Not used |
| Parameters | 100 distinct diffusion-encoding directions,  50x b=1000 s/mm2, 50x b=2000 s/mm2, multi-shell, no cardiac gating |

# Preprocessing

| | |
|---|---|
| Preprocessing software | fsl, nilearn, AFNI |

**Normalization**

Minimal processing was done according to Miller et al., Nature Neuroscience 2016. Additional processing was conducted including despiking (AFNI from Nipype), 6-mm kernel smoothing (Nilearn), and resampling to 3-mm (for storage purposes) to resemble an a-priori brain-based signature for sustained pain (ToPS; see Lee et al., 2021 Nature Medicine).

The preprocessing of diffusion imaging data from the UK Biobank involved using Eddy and BEDPOSTx outputs that remained in the space and resolution of the native diffusion data space after gradient distortion correction (GDC). A nonlinear transformation, as estimated by Tract-Based Spatial Statistics (TBSS), was applied to align this data into the 1mm MNI standard space for generating tractography results in this standardized space.

**Normalization template**

Data were normalize to MNI152 template space.

**Noise and artifact removal**

Minimal processing was done according to Miller et al., Nature Neuroscience 2016. MRI-based covariates included head motion (linear, squared, and cubed), imaging site, position in the scanner, and coil position (Z, Y, Z respectively). Two deconfounding framework were used - see Method for details.

The preprocessing of the diffusion imaging data begins with correction for eddy currents and head motion using the Eddy tool, which also addresses outlier slices in the 4D data set. Following this, gradient distortion correction (GDC) is applied, culminating in the production of the 4D output file. This process is detailed in the documentation found at FSL's Eddy tool website and is based on methodologies described by Andersson and Sotiropoulos in their 2015 and 2016 publications.

**Volume censoring**

N/A

# Statistical modeling & inference

**Model type and settings**

Logistic regression machine learning models were trained using a nested 5-fold cross-validation framework to classify participants either reporting chronic pain or diagnosed with pain-associated medical conditions from their pain-free or diagnosis-free counterparts. These models utilized imaging features derived from three types of scans: resting-state fMRI, diffusion-weighted tractography, and T1 susceptibility-weighted anatomical imaging. The performance of the models was evaluated on the left-out subjects from the testing folds, and quantified using the receiver operating characteristic area under the curve (ROC-AUC) to measure the accuracy of the classification.

**Effect(s) tested**

Effects of chronic pain phenotypes and pain-associated diagnoses on various brain-based imaging derived phenotypes - see Methods for details.

**Specify type of analysis:** ☒ Whole brain ☐ ROI-based ☐ Both

**Statistic type for inference**

(See Eklund et al. 2016)

This study used functional connectivity, anatomical integrity (e.g., cortical thickness, gray matter volume), and white matter tractography (e.g., fractional anisotropy, mean diffusivity).

**Correction**

Significance of group comparisons was determined using false discovery rate (q = 0.05).

# Models & analysis

| n/a | Involved in the study |
|---|---|
| ☐ | ☒ Functional and/or effective connectivity |
| ☒ | ☐ Graph analysis |
| ☐ | ☒ Multivariate modeling or predictive analysis |

**Functional and/or effective connectivity**

DCC was used for Dynamic Connectivity following the same signature extraction from the Tonic Pain Signature (see Lee et al., 2021 Nature Medicine).

**Multivariate modeling and predictive analysis**

independent variables include all imaging derive phenotypes. No feature extraction or dimension reduction was used. The logistic regression ridge (l2 penalty) was tuned using randomized hyperameter search to optimize feature weights. ROC-AUC scores were used to optimize and evaluate models.

