## [Peer Review File · Nature Human Behaviour]

Biological Markers and Psychosocial Factors Predict Chronic Pain Conditions

Corresponding Author: Dr Etienne Vachon-Preseu

Version 0:

Decision Letter:

24th July 2024

Dear Dr Vachon-Preseu,

Thank you once again for your manuscript, entitled "A Biomarker-Based Framework for the Prediction of Future Chronic Pain", and for your patience during the peer review process.

Your Article has now been evaluated by 4 referees. You will see from their comments copied below that, although they find your work of considerable potential interest, they have raised quite substantial concerns. In light of these comments, we cannot accept the manuscript for publication in the current form, but would be interested in considering a revised version if you are willing and able to fully address reviewer and editorial concerns.

We hope you will find the referees' comments useful as you decide how to proceed. If you wish to submit a substantially revised manuscript, please bear in mind that we will be reluctant to approach the referees again in the absence of major revisions. We are committed to providing a fair and constructive peer-review process. Do not hesitate to contact us if there are specific requests from the reviewers that you believe are technically impossible or unlikely to yield a meaningful outcome.

In particular, it will be important to address the technical concerns of Reviewer #4 with appropriate re-analyses.

If you wish to submit a suitably revised manuscript, we would hope to receive it within 4 months. I would be grateful if you could contact us as soon as possible if you foresee difficulties with meeting this target resubmission date.

- Include a "Response to the editors and reviewers" document detailing, point-by-point, how you addressed each editor and referee comment. If no action was taken to address a point, you must provide a compelling argument. When formatting this document, please respond to each reviewer comment individually, including the full text of the reviewer comment verbatim followed by your response to the individual point. This response will be used by the editors to evaluate your revision and sent back to the reviewers along with the revised manuscript.
- Highlight all changes made to your manuscript or provide us with a version that tracks changes.

Link Redacted

Thank you for the opportunity to review your work. Please do not hesitate to contact me if you have any questions or would like to discuss the required revisions further.

Sincerely,

REVIEWER COMMENTS:

Reviewer #1:

Remarks to the Author:

In this article, the authors used machine learning on data from over 523,000 participants from the UK biobank to assess the predictive value of biomarkers for chronic pain. They found that blood biomarkers together with psychosocial risk factors increased the risk for chronic pain. This synergy, although long suspected and discussed, has never been empirically shown in such detail, and this is a great merit of the present work. Furthermore, in spite of the wealth of data and complicated analyses, the paper is well-illustrated and readable.

In the introduction, the authors give some very good examples of non-reliable biomarkers currently used, like radiographic measures of joint degeneration that fail to predict pain, but lead to a lot of uncertainty and worries in patients, and may lead to not-indicated interventions.

In 25% of patients there was no medical diagnosis related to the pain. Were these patients labeled as "chronic primary pain" or "chronic idiopathic pain" by their physicians, or were psychological factors assumed? It is unusual for patients in the health system to go without any label, if there are complaints.

It is interesting that even conditions without known pathophysiology (fibromyalgia, chronic low back pain) could be predicted with reasonable accuracy. When the authors talk about predictive accuracy, would that also be helpful in the differential diagnosis, i.e. for distinguishing RnA and fibromyalgia or CTS from peripheral neuropathy? That would be clinically helpful.

Of course, to avoid circular reasoning, the question is to which extent the biomarkers were already included in diagnosis making in the first place. For example, a genetic marker as in ankle spind may have been used to make the diagnosis and define the disease, such that it is not surprising that this marker did indeed identify the disease in the present analysis. What precautions did the authors take to avoid mistakes of this type?

How certain were the diagnoses, i.e. what diagnostic criteria were underlying the codes?

Reviewer #2:

Remarks to the Author:

major comments:

This is a well-written manuscript describing an interesting set of analyses very relevant in the field of chronic pain. The analyses themselves are hard to find fault with. However I don't think the title of the manuscript & framing this work as 'biomarkers for chronic pain' makes sense, given these biomarkers predict chronic pain associated conditions, and not pain itself. The authors have derived biomarkers that predict a range of conditions, and state themselves that prediction is highest in conditions where pathogenesis is already well-characterized, and that biomarkers (blood, imaging derived etc.) do not predict self-report pain well at all, and psychosocial factors over time are most important in experience of pain. I think this is excellent work and would be better framed as an exploration of chronic pain condition-associated biomarkers and their relationship (or lack of) with self report pain.

I would also disagree with the arguments in line 480 onward - how does 'a more targeted approach' (biomarkers for specific painful conditions) allow for prediction of chronic pain in a generalizable way similar to that referenced in the opening sentence, when the findings in this manuscript say exactly the opposite? Unless the targeted approach meant here is just biomarkers for specific conditions, predicting these specific conditions. In line 498 on, again I don't see how this can be a paper framed as chronic pain biomarkers, when there is an argument to 'sidestep' subjective pain altogether and focus only on specific pain diagnoses - while this is obviously also valuable, how can chronic pain in general be understood or treated in this case?

I appreciate that these are vague points to consider compared to specific reviewer comments, but the discussion especially feels a little confusing to read, with the narrative swapping between 'psychosocial factors are important in predicting pain, a holistic approach is important' to 'better prediction of specific conditions is important, we should sidestep subjective pain altogether'. Editing to bring these two viewpoints together, and/or use of subheadings if allowed would assist the reader.

I want to emphasize again that this work is excellent, and that these criticisms are relatively minor and conceptual.

minor comments/ questions:

~line 483 would benefit from discussion of mixed pain states - even focusing on singular pain related conditions does not guarantee a singular mechanistic pain descriptor / singular distinct pain mechanism

Are the All of Us participants included in validation matched to UKB in terms of genetic ancestry, other characteristics?

For PRS calculation, it is not clear if population structure was assessed in training vs testing sets, which may be present even when comparing same-ancestry UKB participant subgroups, or if genetic PCs were included in PRS analyses

Reviewer #3:

Remarks to the Author:

Nature Human Behaviour 29227

A Biomarker-Based Framework for the Prediction of Future Chronic Pain

Key results:

The authors investigated medical conditions (n=35) and self-reported pain in 523,000 participants (49,001 participants with MRI of the brain including T1-weighted MRI, diffusion MRI and resting-state functional MRI from the UK- Biobank to detect biomarkers predicting chronic pain development.

Participants were investigated at baseline (T0), four years later (T1) and nine years later (T2). The authors used machine learning strategies to investigate multi-dimensional biological modalities including brain imaging, bone imaging, blood immunoassays, and genetics to derive candidate biomarkers for chronic pain. Composite blood-based signature about 9 years in advance predicted medical conditions. Psychosocial factors accurately predicted both medical conditions and self-report pain. Both factors had twice the cumulative incidence of diagnoses for pain-associated medical condition in comparison to the blood-based signature alone which highlights the importance of psychological factors for chronic pain development. Best predictors were found for well characterized pathogenesis such as MS, gout and rheumatoid polymyalgia. Genetic factors (polygenic risk scores) best expressed outcome of pain in ankylosing spondylitis. Brain functional connectivity best predicted nociplastic pain conditions, such as fibromyalgia and chronic fatigue syndrome. Sex differences did not much modulate differences in biomarker performance besides of a brain-based marker for peripheral neuropathy, which was more effective in women and a bone-based marker for osteoporosis, which performed better in men. For psychological factors mood and sleep disturbances were primary features for fibromyalgia. Over all, distinguishing between participants with and without pain, indicated their unreliability in pain prediction. Psychosocial factors offered a more reliable prediction of self-reported pain.

The psychological scores from the topics of mental health, physical health and sociodemography are comprehensive and 81 features have been differentiated. Pain has been obtained by one-month pain and acute pain and 14 pain phenotypes had been differentiated.

Overall, the manuscript is well written and although the data analyses has been quite complex, it is structured well. Due to the complexity of biomarkers investigated the reviewer can only evaluate parts of the methods applied. The procedures performed seem to be well elaborated.

Originality and significance: The major claims are well expressed and represent a significant advance in the field. Data material is very interesting and methods applied are straightforward.

As far as I could figure out the analysis plain had not been preregistered. Statistics have to be evaluated by a specialized reviewer.

Conclusions seem to be robust and valid. References are well selected and Figures are complex but very good. The Abstract is clear and all parts of the manuscript appropriate.

Although the manuscript is long (especially the methods) I do not have a strategy to write it shorter. Overall, I applaud the authors for evaluating this huge amount of data and for identifying the major results from this enormous study.

Reviewer #4:

Remarks to the Author:

Thank you for inviting me to review this manuscript by Fillingim and colleagues, which sought to identify blood, imaging and genetics-based markers that associate with clinical and self-reported pain outcomes.

Their overall conclusions challenge the concept of achieving an 'objective' marker for multiple pain states, which is important and highlights the potential pitfalls of seeking to establish such markers for complex, incompletely-understood medical conditions.

I have the following comments that I hope are helpful for the authors to consider.

- 1) Generalisability - this encompasses two factors. First, UK Biobank cohort is subject to selection bias, which may affect the applicability of findings therefrom to other contexts. Did the authors consider the use of weighting to mitigate this participation/'healthy volunteer' bias? (E.g. <https://www.nature.com/articles/s41562-023-01579-9>) Second, the use of polygenic risk information was only obtained/evaluated in those with European ancestry. This significantly limits the implications of their findings to those from ethnic minority backgrounds. Did the authors consider using other approaches to maximise the involvement of people of non-European ancestry in the analyses in general? It is not clear from the Methods/main paper how this was sought. Do each of the analyses comprise different sub-cohorts?
- 2) Missing data handling - median imputation was used to handle missing values in biochemical data. Why was this approach used rather than a multiple imputation, or a more robust form of deterministic imputation that integrates relationships between covariates? Median imputation will artificially manipulate/restrict the distribution of predictors.
- 3) Clarity on definition of outcome - the authors discuss the pain phenotypes in the 'Online Methods' section. The timing of ascertaining prediction 'target' is not clear - I assume that these were based on baseline session at the assessment centre? It is

useful to be completely clear about the temporality and other facets of defining the 'endpoint'. The title refers to 'future', but pain-associated diagnoses appear to be able to be present on baseline assessment, or on NHS records (during follow-up?). Were incident cases and prevalent cases used, or only one type? The relationship between the timing of the predictor measurements and the outcomes they seek to predict is not clear but impacts the interpretation of results and steers the appropriateness of analyses.

4) Use of logistic regression - an example of why definition of the endpoint is crucial to understand. If there is a time-to-event component, i.e. individuals may not have had pain at baseline, but developed pain/received a pain state diagnosis years after covariate collection, then logistic regression is not an appropriate modelling choice (far better choices could be a flexible parametric survival model, or penalised Cox models that handle the time-to-event nature of these data).

5) Usefulness of AUC as a comparison between markers/models - there has been some criticism of comparing models using a single discrimination metric. Did the authors consider using a broader set of techniques to formally compare the predictive capabilities of models using different markers/different sets? (<https://www.fharrell.com/post/addvalue/>)

Version 1:

Decision Letter:

Our ref: NATHUMBEHAV-24051690A

6th December 2024

Dear Dr. Vachon-Preseau,

Thank you for submitting your revised manuscript "A Holistic Framework for Biomarkers of Pain Conditions and Self-Reported Pain" (NATHUMBEHAV-24051690A). It has now been seen by the original referees and their comments are below. As you can see, the reviewers find that the paper has improved in revision. We will therefore be happy in principle to publish it in Nature Human Behaviour, pending minor revisions to comply with our editorial and formatting guidelines.

We are now performing detailed checks on your paper and will send you a checklist detailing our editorial and formatting requirements within two weeks. Please do not upload the final materials and make any revisions until you receive this additional information from us.

Sincerely,

██████████

Dr ██████████ ██████████

██████████
Nature Human Behaviour

Reviewer #1 (Remarks to the Author):

Thank you for answering all the questions and for including additional analyses.

Reviewer #2 (Remarks to the Author):

I thank the authors for this really great rebuttal document. The authors have fully addressed all of my points, I have nothing further to add.

Reviewer #3 (Remarks to the Author):

The authors addressed all my concerns and I think the manuscript is now ready for acceptance.

Reviewer #4 (Remarks to the Author):

Thank you very much for the robust and comprehensive discussion around the comments. The authors have clearly expended great effort responding to all reviewers, have outlined in clear detail their rationale, and provided excellent clarifications to the main body and the supplementary information.
I have no further concerns from my perspective.

Version 2:

Decision Letter:

Dear Dr Vachon-Preseau,

We are pleased to inform you that your Article "Biological Markers and Psychosocial Factors Predict Chronic Pain Conditions", has now been accepted for publication in Nature Human Behaviour.

With best regards,

██████

Dr ██████████

██████████
Nature Human Behaviour

P.S. Click on the following link if you would like to recommend Nature Human Behaviour to your librarian
<http://www.nature.com/subscriptions/recommend.html#forms>

** Visit the Springer Nature Editorial and Publishing website at http://editorial-jobs.springernature.com?utm_source=ejp_NHumB_email&utm_medium=ejp_NHumB_email&utm_campaign=ejp_NHumB for more information about our career opportunities. If you have any questions please click [here](mailto:editorial.publishing.jobs@springernature.com).

Reviewer Comments and Author Rebuttals

1. Reviewer Comments:

Manuscript Title: A Biomarker-Based Framework for the Prediction of Future Chronic Pain

Editor's Remarks to the Author:

Thank you once again for your manuscript, entitled "A Biomarker-Based Framework for the Prediction of Future Chronic Pain", and for your patience during the peer review process.

Your Article has now been evaluated by 4 referees. You will see from their comments copied below that, although they find your work of considerable potential interest, they have raised quite substantial concerns. In light of these comments, we cannot accept the manuscript for publication in the current form, but would be interested in considering a revised version if you are willing and able to fully address reviewer and editorial concerns.

We hope you will find the referees' comments useful as you decide how to proceed. If you wish to submit a substantially revised manuscript, please bear in mind that we will be reluctant to approach the referees again in the absence of major revisions. We are committed to providing a fair and constructive peer-review process. Do not hesitate to contact us if there are specific requests from the reviewers that you believe are technically impossible or unlikely to yield a meaningful outcome.

In particular, it will be important to address the technical concerns of Reviewer #4 with appropriate re-analyses.

If you wish to submit a suitably revised manuscript, we would hope to receive it within 4 months. I would be grateful if you could contact us as soon as possible if you foresee difficulties with meeting this target resubmission date.

- Include a "Response to the editors and reviewers" document detailing, point-by-point, how you addressed each editor and referee comment. If no action was taken to address a point, you must provide a compelling argument. When formatting this document, please respond to each reviewer comment individually, including the full text of the reviewer comment verbatim followed by your response to the individual point. This response will be used by the editors to evaluate your revision and sent back to the reviewers along with the revised manuscript.

- Highlight all changes made to your manuscript or provide us with a version that tracks changes.

Reviewer #1:

Remarks to the Author:

In this article, the authors used machine learning on data from over 523,000 participants from the UK biobank to assess the predictive value of biomarkers for chronic pain. They found that blood biomarkers together with psychosocial risk factors increased the risk for chronic pain. This synergy, although long suspected and discussed, has never been empirically shown in such detail, and this is a great merit of the present work. Furthermore, in spite of the wealth of data and complicated analyses, the paper is well-illustrated and readable.

In the introduction, the authors give some very good examples of non-reliable biomarkers currently used, like radiographic measures of joint degeneration that fail to predict pain, but lead to a lot of uncertainty and worries in patients, and may lead to not-indicated interventions.

In 25% of patients there was no medical diagnosis related to the pain. Were these patients labeled as “chronic primary pain” or “chronic idiopathic pain” by their physicians, or were psychological factors assumed? It is unusual for patients in the health system to go without any label, if there are complaints.

It is interesting that even conditions without known pathophysiology (fibromyalgia, chronic low back pain) could be predicted with reasonable accuracy. When the authors talk about predictive accuracy, would that also be helpful in the differential diagnosis, i.e. for distinguishing R/A and fibromyalgia or CTS from peripheral neuropathy? That would be clinically helpful.

Of course, to avoid circular reasoning, the question is to which extent the biomarkers were already included in diagnosis making in the first place. For example, a genetic marker as in ankle spind may have been used to make the diagnosis and define the disease, such that it is not surprising that this marker did indeed identify the disease in the present analysis. What precautions did the authors take to avoid mistakes of this type?

How certain were the diagnoses, i.e. what diagnostic criteria were underlying the codes?

Reviewer #2:

Remarks to the Author:

major comments:

This is a well-written manuscript describing an interesting set of analyses very relevant in the field of chronic pain. The analyses themselves are hard to find fault with. However I don't think the title of the manuscript & framing this work as 'biomarkers for chronic pain' makes sense, given these

biomarkers predict chronic pain associated conditions, and not pain itself. The authors have derived biomarkers that predict a range of conditions, and state themselves that prediction is highest in conditions where pathogenesis is already well-characterized, and that biomarkers (blood, imaging derived etc.) do not predict self-report pain well at all, and psychosocial factors over time are most important in experience of pain. I think this is excellent work and would be better framed as an exploration of chronic pain condition-associated biomarkers and their relationship (or lack of) with self report pain.

I would also disagree with the arguments in line 480 onward - how does 'a more targeted approach' (biomarkers for specific painful conditions) allow for prediction of chronic pain in a generalizable way similar to that referenced in the opening sentence, when the findings in this manuscript say exactly the opposite? Unless the targeted approach meant here is just biomarkers for specific conditions, predicting these specific conditions. In line 498 on, again I don't see how this can be a paper framed as chronic pain biomarkers, when there is an argument to 'sidestep' subjective pain altogether and focus only on specific pain diagnoses - while this is obviously also valuable, how can chronic pain in general be understood or treated in this case?

I appreciate that these are vague points to consider compared to specific reviewer comments, but the discussion especially feels a little confusing to read, with the narrative swapping between 'psychosocial factors are important in predicting pain, a holistic approach is important' to 'better prediction of specific conditions is important, we should sidestep subjective pain altogether'. Editing to bring these two viewpoints together, and/or use of subheadings if allowed would assist the reader.

I want to emphasize again that this work is excellent, and that these criticisms are relatively minor and conceptual.

minor comments/ questions:

~line 483 would benefit from discussion of mixed pain states - even focusing on singular pain related conditions does not guarantee a singular mechanistic pain descriptor / singular distinct pain mechanism

Are the All of Us participants included in validation matched to UKB in terms of genetic ancestry, other characteristics?

For PRS calculation, it is not clear if population structure was assessed in training vs testing sets, which may be present even when comparing same-ancestry UKB participant subgroups, or if genetic PCs were included in PRS analyses

Reviewer #3:

Remarks to the Author:

Nature Human Behaviour 29227

A Biomarker-Based Framework for the Prediction of Future Chronic Pain

Key results:

The authors investigated medical conditions (n=35) and self-reported pain in 523,000 participants (49,001 participants with MRI of the brain including T1-weighted MRI, diffusion MRI and resting-state functional MRI from the UK- Biobank to detect biomarkers predicting chronic pain development.

Participants were investigated at baseline (T0), four years later (T1) and nine years later (T2). The authors used machine learning strategies to investigate multi-dimensional biological modalities including brain imaging, bone imaging, blood immunoassays, and genetics to derive candidate biomarkers for chronic pain. Composite blood-based signature about 9 years in advance predicted medical conditions. Psychosocial factors accurately predicted both medical conditions and self-report pain. Both factors had twice the cumulative incidence of diagnoses for pain-associated medical condition in comparison to the blood-based signature alone which highlights the importance of psychological factors for chronic pain development. Best predictors were found for well characterized pathogenesis such as MS, gout and rheumatoid polymyalgia. Genetic factors (polygenic risk scores) best expressed outcome of pain in ankylosing spondylitis. Brain functional connectivity best predicted nociplastic pain conditions, such as fibromyalgia and chronic fatigue syndrome. Sex differences did not much modulate differences in biomarker performance besides of a brain-based marker for peripheral neuropathy, which was more effective in women and a bone-based marker for osteoporosis, which performed better in men. For psychological factors mood and sleep disturbances were primary features for fibromyalgia. Over all, distinguishing between participants with and without pain, indicated their unreliability in pain prediction. Psychosocial factors offered a more reliable prediction of self-reported pain.

The psychological scores from the topics of mental health, physical health and sociodemography are comprehensive and 81 features have been differentiated. Pain has been obtained by one-month pain and acute pain and 14 pain phenotypes had been differentiated.

Overall, the manuscript is well written and although the data analyses has been quite complex, it is structured well. Due to the complexity of biomarkers investigated the reviewer can only evaluate parts of the methods applied. The procedures performed seem to be well elaborated.

Originality and significance: The major claims are well expressed and represent a significant advance in the field. Data material is very interesting and methods applied are straightforward.

As far as I could figure out the analysis plain had not been preregistered. Statistics have to be evaluated by a specialized reviewer.

Conclusions seem to be robust and valid. References are well selected and Figures are complex but very good. The Abstract is clear and all parts of the manuscript appropriate.

Although the manuscript is long (especially the methods) I do not have a strategy to write it shorter. Overall, I applaud the authors for evaluating this huge amount of data and for identifying the major results from this enormous study.

Reviewer #4:

Remarks to the Author:

Thank you for inviting me to review this manuscript by Fillingim and colleagues, which sought to identify blood, imaging and genetics-based markers that associate with clinical and self-reported pain outcomes.

Their overall conclusions challenge the concept of achieving an 'objective' marker for multiple pain states, which is important and highlights the potential pitfalls of seeking to establish such markers for complex, incompletely-understood medical conditions.

I have the following comments that I hope are helpful for the authors to consider.

1) Generalisability - this encompasses two factors. First, UK Biobank cohort is subject to selection bias, which may affect the applicability of findings therefrom to other contexts. Did the authors consider the use of weighting to mitigate this participation/'healthy volunteer' bias? (E.g. <https://www.nature.com/articles/s41562-023-01579-9>)

Second, the use of polygenic risk information was only obtained/evaluated in those with European ancestry. This significantly limits the implications of their findings to those from ethnic minority backgrounds. Did the authors consider using other approaches to maximise the involvement of people of non-European ancestry in the analyses in general? It is not clear from the Methods/main paper how this was sought. Do each of the analyses comprise different sub-cohorts?

2) Missing data handling - median imputation was used to handle missing values in biochemical data. Why was this approach used rather than a multiple imputation, or a more robust form of deterministic imputation that integrates relationships between covariates? Median imputation will artificially manipulate/restrict the distribution of predictors.

3) Clarity on definition of outcome - the authors discuss the pain phenotypes in the 'Online Methods' section. The timing of ascertaining prediction 'target' is not clear - I assume that these were based on baseline session at the assessment centre? It is useful to be completely clear about the temporality and other facets of defining the 'endpoint'. The title refers to 'future', but pain-associated diagnoses appear to be able to be present on baseline assessment, or on NHS records (during follow-up?). Were incident cases and prevalent cases used, or only one type? The relationship between the timing of the predictor measurements and the outcomes they seek to predict is not clear but impacts the interpretation of results and steers the appropriateness of analyses.

4) Use of logistic regression - an example of why definition of the endpoint is crucial to understand. If there is a time-to-event component, i.e. individuals may not have had pain at baseline, but developed pain/received a pain state diagnosis years after covariate collection, then logistic regression is not an appropriate modelling choice (far better choices could be a flexible parametric survival model, or penalised Cox models that handle the time-to-event nature of these data).

5) Usefulness of AUC as a comparison between markers/models - there has been some criticism of comparing models using a single discrimination metric. Did the authors consider using a broader set of techniques to formally compare the predictive capabilities of models using different markers/different sets? (<https://www.fharrell.com/post/addvalue/>)

2. Authors Rebuttals to Initial Comments:

[BLACK BOLD] - ORIGINAL COMMENT

[BLUE] - RESPONSE TO COMMENT

[HIGHLIGHTED] - TEXT AND FIGURE AMENDMENTS

Response to Reviewer #1:

Ref 1.1:

In this article, the authors used machine learning on data from over 523,000 participants from the UK biobank to assess the predictive value of biomarkers for chronic pain. They found that blood biomarkers together with psychosocial risk factors increased the risk for chronic pain. This synergy, although long suspected and discussed, has never been empirically shown in such detail, and this is a great merit of the present work. Furthermore, in spite of the wealth of data and complicated analyses, the paper is well-illustrated and readable.

Response: We are thankful for the kind feedback.

Ref 1.2:

In 25% of patients there was no medical diagnosis related to the pain. Were these patients labeled as “chronic primary pain” or “chronic idiopathic pain” by their physicians, or were psychological factors assumed? It is unusual for patients in the health system to go without any label, if there are complaints.

Response: Among the 25% of patients who reported pain without a corresponding medical diagnosis, several factors may be at play. One possibility is that some individuals did not seek medical advice, thereby missing a formal diagnosis. It is crucial to note that these individuals reported their symptoms through questionnaires, which does not guarantee that their concerns were evaluated by the healthcare system. New analyses conducted in the UKB data show that patients who reported pain without a medical diagnosis exhibited lower pain intensity (worst pain

in the last 24 hours), impact (BPI), and symptom severity compared to those who had a diagnosis.

Pain Impact by Diagnosis Status

Moreover, the majority of these patients reported pain at only one site, compared to higher rates of multi-site pain in patients with a medical diagnosis, as shown in Fig. 1B. Thus, our findings indicate that the intensity of pain, its impact, and severity are correlated with the number of pain sites (figure below). Notably, these measures of pain interference are weakest among those reporting single-site pain, which suggests that the pain experienced by individuals in this 25% may not significantly interfere with their daily life, which might lead to a lower likelihood of seeking medical attention. This observation may help to contextualize the lack of formal medical labels among this group of patients, despite experiencing chronic pain.

Pain Impact by Number of Chronic Pain Sites

Alternatively, some patients might be reporting recent pain for which they have not yet visited a healthcare provider at the time of the UK Biobank session. The UK Biobank's pain questionnaire specifically asks if participants have experienced pain affecting their daily activities at major body sites for more than three months. This query could capture recently developed pain that has not been clinically evaluated yet.

Additionally, a small fraction of UK Biobank participants (about 12%) lacks accessible National Health Service (NHS) records, and we must rely on self-reported diagnoses for these individuals. It is possible that some of these individuals did not report existing conditions, possibly due to forgetfulness or survey fatigue. This is now emphasized in the methods second of the revised manuscript as follows:

Page 38: For health outcomes resulting in hospital admissions, hospital inpatient records utilizing the International Classification of Diseases and Related Health Problems (ICD-10) coded primary or secondary diagnoses (Field IDs 41270 and 41271) were available for over 87% of study cohort (n = 446,829). Meanwhile, primary care data, available for 45% of the study cohort (n = ~230,000), were obtained from Read Codes v2, as coded by general practitioners (Field ID 42040). The UK Biobank additionally provided curated data fields indicating the first occurrence of a set of diagnostic codes (Category: 1712) for a wide range of health outcomes across self-report, primary care, hospital inpatient data and death data, mapped to a 3-digit ICD code. For broader diagnoses that were effectively captured with a three-digit ICD code (e.g., Rheumatoid arthritis; ICD code: M05 & M06), we extracted information from this first occurrences database. In contrast, for conditions defined by more granular ICD coding (e.g., ankylosing spondylitis; ICD code: M081), data were manually curated from self-report and hospital inpatient records.

In summary, while the majority of patients reported pain associated with a clinical diagnosis, some individuals experienced pain without an identifiable medical condition. Notably, these patients reported experiencing lower pain intensity that had less impact on their daily lives, which may explain pain without a clinical diagnosis.

Ref 1.3:

It is interesting that even conditions without known pathophysiology (fibromyalgia, chronic low back pain) could be predicted with reasonable accuracy. When the authors talk about predictive accuracy, would that also be helpful in the differential diagnosis, i.e. for distinguishing RhA and fibromyalgia or CTS from peripheral neuropathy? That would be clinically helpful.

Response: *Thank you for raising this point. We indeed examined the predictive accuracy in differential diagnosis, as detailed in Extended Fig. 5, where we used a cross-prediction paradigm. This approach is based on training models on specific diagnoses (e.g., gout) and testing their performance to classify other conditions (e.g., Rheumatoid Arthritis, Crohn's disease, Carpal Tunnel Syndrome, etc.) for which they were not initially trained to predict.*

For instance, our analyses showed that blood-based models showed relatively low sensitivity (.45-.67) for classifying diagnoses on which they were not trained on. They nonetheless exhibited medium to high specificity (0.68-0.83), indicating they can effectively identify healthy controls. On the other hand, brain-based models showed mixed performances based on the training diagnosis. Cross-prediction models for nociplastic pain conditions like fibromyalgia and chronic fatigue syndrome were accurate in predicting alternative conditions that share symptomatic profiles (sensitivity 0.70-0.75). However, for conditions with distinct pathophysiology such as multiple sclerosis, the models showed lower sensitivity (0.53) when applied to other diagnoses (Extended Fig. 5C).

A | Cross-prediction models

B | Cross-prediction ROC-AUC scores

C | Cross-prediction sensitivity x specificity analysis

Extended Data Fig. 5 | Biological and psychosocial cross-prediction models for pain diagnoses. **A**, Models trained on specific diagnoses were evaluated for their ability to predict other diagnoses, with average performance metrics (ROC-AUC, sensitivity, specificity) across these diagnoses. This cross-prediction analysis was conducted for the diagnoses that were most accurately classified using blood, brain, and bone modalities alongside psychosocial models. **B**, Average cross-prediction ROC-AUC curves are displayed for both biological and psychosocial models within each biological modality. **C**, Quadrant plots show the average sensitivity and specificity of cross-prediction for each diagnosis. Points within the plots are color-coded by modality and labeled by the diagnosis on which the model was trained. Here, sensitivity measures the model's accuracy in detecting untrained diagnoses, while specificity measures its precision in identifying diagnosis-free controls.

To fully address the reviewer's query, we evaluated the performance of models specifically trained to differentiate common diagnoses using blood assays and brain imaging for conditions they best predicted. The results showed a range of performances, with AUC values ranging between 0.40 and 0.70. Notably, the model distinguishing RA from fibromyalgia based on blood assays performed the best, achieving an AUC of 0.70. However, for most other differential diagnoses, the predictive accuracy was less robust, with AUC of 0.65 or lower. This includes the comparisons of Gout versus Rheumatoid Arthritis (AUC = 0.65) and Multiple Sclerosis versus Stroke (AUC = 0.56), suggesting that these models were less effective at distinguishing between these conditions than comparing them with healthy controls (AUCs > 0.78). Overall, while biomarkers may be useful in detecting deviations from a normative healthy state, their effectiveness in distinguishing between various clinical conditions remains less clear, despite some examples achieving fair-good accuracy in differential diagnoses.

Differential diagnoses using blood immunoassays

Differential diagnoses using brain imaging

In summary, our findings indicate that while there is potential for using some of our biomarkers in differential diagnosis, the overall accuracy of these models varied across various conditions, suggesting that further refinement is needed before they can be reliably implemented in clinical practice. These results underscore the need for ongoing research to enhance the precision and applicability of biomarker-based models in complex diagnostic scenarios.

Ref 1.4:

Of course, to avoid circular reasoning, the question is to which extent the biomarkers were already included in diagnosis making in the first place. For example, a genetic marker as in ank spond may have been used to make the diagnosis and define the disease, such that it is not surprising that this marker did indeed identify the disease in the present analysis. What precautions did the authors take to avoid mistakes of this type?

Response: *The reviewer is once again raising an important point. We would like to highlight the following points to mitigate their concerns:*

- Broad Categorization of Conditions:** *We grouped conditions broadly to avoid biases in our target definitions. For instance, both seropositive and seronegative forms were included under the Rheumatoid Arthritis category, and idiopathic cases under Gout, ensuring inclusivity. This helped mitigate circular reasoning for some of these conditions.*
- The use of Routinely collected Biological Predictors:** *We demonstrated that pain conditions can be predicted using non-specific biological measures collected in both research settings and routine clinical testing. For example, Multiple Sclerosis is typically diagnosed with gadolinium-based contrast agents to detect demyelinating lesions, a test not available in the UK Biobank. Instead, we showed that basic routine sequences, such as*

diffusion-weighted imaging, T1-weighted imaging, or even resting-state functional connectivity, could achieve good-excellent predictive performance. Similarly, Crohn's disease, which is characterized by bowel inflammation usually diagnosed through endoscopy or other imaging techniques, can be predicted with an AUC of 0.78 using a blood test. Although these conditions are defined by specific pathophysiology, our findings demonstrate that they can be detected and predicted using widely accessible biological measurements. This is an important take away from our manuscript, which has been emphasized in this revision.

3. **Predictive Validity in Longitudinal Samples:** Several of our models have successfully predicted individuals who would later develop certain diseases (e.g., **Fig. 2B**, **Fig. 5G**), indicating their predictive power beyond existing diagnostic criteria.
4. **Conditions with Less Defined Pathophysiology:** For multiple conditions we studied, such as fibromyalgia and peripheral neuropathy, the pathophysiology isn't well-defined, and reliable biomarkers for diagnosis are lacking. This was also the case for our models trained on pain symptoms (e.g., knee pain), therefore reducing the risk of circular reasoning for these conditions.

Despite these mitigation strategies, the concern of circular reasoning is relevant for conditions with well-defined pathophysiology's, such as Gout, Multiple Sclerosis, and Rheumatoid Arthritis, where specific diagnostic tests are often integral to the diagnosis. Yet, even in those cases, the capacity of our models applied in left out individuals (external to our training sample) did not obtain perfect accuracy, revealing that our biomarkers misdiagnosed several participants. We argue that this help contextualize the ranges of effect sizes of less well understood conditions, such as Fibromyalgia that appears to be more predictable than previously recognized. This approach allows us to establish a spectrum of "biomarker potential" across various pain diagnoses, ranging from those with well-characterized pathophysiological markers but also for various conditions for which those biomarkers are less defined. Thus, the key strength of our approach lies in the simultaneous consideration of multiple conditions and biological predictors. For instance, in the case of ankylosing spondylitis highlighted by the reviewer, the condition showed good, though not perfect, predictive accuracy using polygenic risk scores (PRS). Notably, it can also be predicted using inflammatory and immune blood markers, which may be crucial for a deeper understanding of the condition's determinants and predictors. Finally, the main finding of our manuscript is that biomarkers should be identified and leveraged by integrating biology within its biopsychosocial contexts. This holistic approach shows how biological and psychosocial factors together influence disease processes (refer to **Fig. 5-6**). We demonstrate that these factors interact synergistically, where their combined effect increases the likelihood of a future diagnosis more significantly than when they operate individually. This synergy holds true even in cases where a clinical diagnosis may have been informed by a patient's biological markers. This comprehensive perspective is now reflected in the new title of the paper: **A Holistic Framework for Biomarkers of Pain Conditions and Self-Reported Pain**.

Ref 1.5:

How certain were the diagnoses, i.e. what diagnostic criteria were underlying the codes?

Response: Diagnoses were obtained from two primary sources: participant self-reports during study sessions and coded health records from the NHS, which include data from general

practitioners and primary care. These were curated by UKB into 'first occurrence' fields, which define each health outcome by the 3-character codes within ICD10's diagnostic chapters, excluding cancer. More details can be found here: <https://biobank.ndph.ox.ac.uk/ukb/label.cgi?id=1712>.

To clarify the specific diagnostic codes associated with each diagnosis, we have created a reference table (Supplementary table 7). This table enumerates all the ICD-10 formatted diagnostic codes pertaining to each diagnosis.

Unfortunately, there is no way for us to determine the specific diagnostic criteria used by physicians to deliver a diagnosis to an individual participant. However, these codes follow standardized criteria for diagnosis, ensuring a consistent and reliable framework for defining each condition. This diagnostic coding scheme has been repeatedly used in the UKBiobank by previous studies and should ideally be kept consistent to harmonize findings and discoveries in this large biobank^{1,2,8}.

Response to Reviewer #2:

Ref 2.1:

This is a well-written manuscript describing an interesting set of analyses very relevant in the field of chronic pain. The analyses themselves are hard to find fault with. However I don't think the title of the manuscript & framing this work as 'biomarkers for chronic pain' makes sense, given these biomarkers predict chronic pain associated conditions, and not pain itself. The authors have derived biomarkers that predict a range of conditions, and state themselves that prediction is highest in conditions where pathogenesis is already well-characterized, and that biomarkers (blood, imaging derived etc.) do not predict self-report pain well at all, and psychosocial factors over time are most important in experience of pain. I think this is excellent work and would be better framed as an exploration of chronic pain condition-associated biomarkers and their relationship (or lack of) with self report pain.

***Response:** We are thankful to the reviewer for their positive feedback. They raise an important point about the framing of our manuscript, particularly the use of the term "biomarkers for chronic pain." We agree that the manuscript would benefit from a clearer distinction between chronic pain and chronic pain-associated conditions. To address this, we have revised the title to “**A Holistic Framework for Biomarkers of Pain Conditions and Self-Reported Pain**” and have made corresponding adjustments in the discussion to clarify that while biomarkers can predict certain chronic pain conditions, they do not directly predict self-reported pain experiences, which are more strongly influenced by psychosocial factors over time. We outline the changes made to the manuscript in your other comments below.*

These changes aim to better reflect the scope of our findings and emphasize the relationship between biomarkers, chronic pain conditions, and the subjective experience of pain. We appreciate the reviewer's insights, which have helped refine our manuscript.

Ref 2.2:

I would also disagree with the arguments in line 480 onward - how does 'a more targeted approach' (biomarkers for specific painful conditions) allow for prediction of chronic pain in a generalizable

way similar to that referenced in the opening sentence, when the findings in this manuscript say exactly the opposite? Unless the targeted approach meant here is just biomarkers for specific conditions, predicting these specific conditions. In line 498 on, again I don't see how this can be a paper framed as chronic pain biomarkers, when there is an argument to 'sidestep' subjective pain altogether and focus only on specific pain diagnoses - while this is obviously also valuable, how can chronic pain in general be understood or treated in this case?

***Response:** These are excellent points. Our discussion from line 480 onward did refer to a targeted approach based on the chronic pain condition. Throughout the discussion, we aimed to distinguish between the utility of biomarkers for specific conditions and those for self-reported chronic pain. This distinction arises from the observation that similar pain symptoms (e.g. self-reported chronic pain at a given body location) can emerge from diverse underlying conditions, such as neuropathic pain from pinched nerves, neuralgias, or neurodegenerative diseases. We argue that biomarkers are more effective in predicting these specific conditions rather than the broader experience of pain, which is influenced heavily by psychosocial factors. This approach does not ignore the subjective experience of pain but rather acknowledges that understanding the biological underpinnings of specific conditions can lead to more precise management and treatment strategies of the pain itself. Thus, while our biomarkers target specific diagnoses, they indirectly contribute to our understanding of chronic pain by elucidating the pathologies that underlie common pain symptoms.*

*This line of reasoning was previously proposed in a review paper about biomarkers for chronic pain from Van Der Miesen, Lindquist, and Wager, who argued that: "Pain, however, has few biomarkers that are widely used in clinical practice. Some biomarkers are intended to track pain intensity and complement self-reports as a way of assessing the incidence or intensity of pain. Others are intended to reveal underlying pathobiological conditions that cause pain. As we argue below, this latter type is what is most badly needed."²⁰ The reviewer can refer to van der Miesen et al. (2019) for the full argumentation; but in summary, it is proposed that biomarkers for chronic pain conditions hold value where pathology informs the risk of pain (gleaned from biological markers of specific conditions). As shown in our manuscript, biomarkers could classify pain conditions whereas the psychosocial factors shaped the self-reported expression of pain. Importantly, our findings demonstrate that these elements do not operate in isolation but interact synergistically, amplifying both the risk of developing pain pathology (as shown in **Fig. 5G**) and its actual self-reported expression, which is predictable from non-biological risk factors (as shown in **Fig. 6**). This holistic approach allows us to address both the biological and experiential aspects of pain, providing a more comprehensive understanding of its mechanisms. We believe this synergy has been mostly ignored by the field and we propose that future research aiming at identifying biomarkers for pain include these non-biological contextual factors, as initially posited by the biopsychosocial model for chronic pain.*

We appreciate the critique and agree that the manuscript should clearly differentiate these perspectives. We have adjusted our discussion to better articulate how targeting specific conditions with biomarkers can contribute to the broader understanding and treatment of chronic pain. We have made the following changes to the manuscript:

The paragraph in discussion page 23 has been modified to discuss the benefits of focusing on deriving biomarkers of specific pain conditions and their mechanisms.

Nonetheless, efforts in chronic pain research have historically aimed to identify biomarkers generalizable to clinical pain (i.e., presence of chronic pain)^{2,14,23}. Our findings suggest we should focus our efforts on identifying biomarkers specific to the medical conditions associated with clinical pain. Chronic pain originates from various mechanisms—nociceptive, neuropathic, inflammatory, and nociceptive—each characterized by unique pathological features embedded in different bodily systems^{9,24,25}. Mixed pain states further complicate the landscape of chronic pain, as they involve combinations of these mechanisms, such as nociceptive and neuropathic pain in conditions like low back pain. Addressing these mixed states requires biomarkers that can accurately capture the interplay of these mechanisms. To enhance prediction accuracy for mixed pain conditions, composite biomarkers that integrate various mechanisms can be utilized. This approach enables the development of biomarkers that address not only specific conditions but also the multifaceted nature of mixed pain states. For instance, we have demonstrated the effectiveness of this approach with a composite blood signature that predicts several inflammatory, neuropathic, and nociceptive conditions (**Fig. 2**) and a functional connectivity signature for nociceptive pain (**Fig. 3**). These examples highlight that biomarkers reflecting shared underlying pathologies, rather than focusing on one specific condition, can provide robust predictive capabilities.

And page 23:

Importantly, this approach does not ignore the subjective experience of pain but rather acknowledges that understanding the biological underpinnings of specific conditions can lead to more precise management and treatment strategies of the pain itself. While our biomarkers are designed to target specific diagnoses, they also enhance our understanding of chronic pain by revealing the pathologies that underlie common pain symptoms. In this way, our findings support a model where biological pathology informs the risk of pain, while psychosocial factors shape its expression. Importantly, our findings demonstrate that these elements do not operate in isolation but interact synergistically, amplifying both the risk of developing pain pathology (as shown in **Fig. 5G**) and its actual expression (as shown in **Fig. 6**). This holistic approach allows us to address both the biological and experiential dimensions of pain, offering a more comprehensive understanding of its mechanisms.

Ref 2.3:

I appreciate that these are vague points to consider compared to specific reviewer comments, but the discussion especially feels a little confusing to read, with the narrative swapping between 'psychosocial factors are important in predicting pain, a holistic approach is important' to 'better prediction of specific conditions is important, we should sidestep subjective pain altogether'. Editing to bring these two viewpoints together, and/or use of subheadings if allowed would assist the reader.

I want to emphasize again that this work is excellent, and that these criticisms are relatively minor and conceptual.

Response: The discussion now begins with a discussion of the limitations in biomarkers for self-reported pain, followed by a discussion of the importance of focusing on pain condition specific biomarkers and how this can help us understand self-reported pain, and ending with a discussion

of the integration of biological and psychosocial elements to biomarker development. We added subheadings in this revised version, but we might have to remove them, given that we believe they are not allowed at this journal.

Ref 2.4:

minor comments/ questions:

~line 483 would benefit from discussion of mixed pain states - even focusing on singular pain related conditions does not guarantee a singular mechanistic pain descriptor / singular distinct pain mechanism

***Response:** We have included a discussion of mixed pain states in the discussion section on page 23, as mentioned in **Ref 2.2** above:*

Mixed pain states further complicate the landscape of chronic pain, as they involve combinations of these mechanisms, such as nociceptive and neuropathic pain in conditions like low back pain. Addressing these mixed states requires biomarkers that can accurately capture the interplay of these mechanisms. To enhance prediction accuracy for mixed pain conditions, composite biomarkers that integrate various mechanisms can be utilized. This approach enables the development of biomarkers that address not only specific conditions but also the multifaceted nature of mixed pain states. For instance, we have demonstrated the effectiveness of this approach with a composite blood signature that predicts several inflammatory, neuropathic, and nociceptive conditions (**Fig. 2**) and a functional connectivity signature for nociplastic pain (**Fig. 3**). These examples highlight that biomarkers reflecting shared underlying pathologies, rather than focusing on one specific condition, can provide robust predictive capabilities.

Ref 2.5:

Are the All of Us participants included in validation matched to UKB in terms of genetic ancestry, other characteristics?

***Response:** No, participants from the All of Us cohort were not matched to the UK Biobank in terms of genetic ancestry or other specific characteristics because no genetic analyses were conducted on the All of Us data. Instead, this cohort was employed solely to validate the blood immunoassay signature developed using the UK Biobank data, as shown in **Fig. 2**. We specifically chose the All of Us cohort for validation due to its ethnically diverse composition and broader age range compared to the UK Biobank. This choice allowed us to test whether the signature, originally derived from a predominantly older, white cohort, would also be applicable to a more varied and younger population (also see response to **Ref 4.2**). The results confirmed that the signature generalized well across these different demographics.*

A similar strategy was used for the brain imaging results using the OpenPain data repository. We did not validate other genetic or bone biomarkers because both modalities were not as effective for the prediction of either chronic pain conditions or self-reported pain.

Ref 2.6:

For PRS calculation, it is not clear if population structure was assessed in training vs testing sets, which may be present even when comparing same-ancestry UKB participant subgroups, or if genetic PCs were included in PRS analyses

Response: In our analysis, we accounted for potential population stratification by including the top 40 genetic principal components (PCs) as covariates in the PRS models. These PCs were included through the PRSice software's command line options --cov and --cov-col. Notably, only the top three were shown to sufficiently correct for population stratification, which aligns with recent research¹⁹. We opted for a more conservative approach given the diverse subgroups within the UK Biobank. Additionally, age and sex were included as covariates, which have been shown to significantly affect PRS modeling performance¹⁹. We also controlled for the genotyping platform and used dummy-coded recruitment sites as additional covariables to further minimize confounding.

These covariates were applied across both the training and testing sets. We have updated the Methods section of our manuscript to explicitly clarify the use of these covariates in our PRS analytical approach and addressed the limitations of including only participants of non-European ancestry in its calculation.

Page 33:

Genome Wide Association Studies (GWAS): For each training set across each pain phenotype and pain-associated diagnosis, we conducted genome-wide association studies (GWAS) using regenie²⁶. Regenie was selected for its ability to address cryptic relatedness among UK Biobank (UKB) participants and manage case/control imbalances. To control for population structure in GWAS estimation, covariates were applied across both training and testing sets, including sex, age, age squared, genotyping array, the first 40 genetic principal components, and dummy-coded recruitment sites. Sex was determined based on genetically ascertained sex (XY=man, XX=woman; field 22001). In regenie's step 1, 93K SNPs identified by the UKB for kinship estimation were analyzed. For step 2 in regenie, approximately one million SNPs were evaluated, selected from the European-specific subset of the pan-UK Biobank project. These SNPs are high-quality HapMap 3 variants that meet five criteria: located in autosomes, outside the MHC region, bi-allelic, with INFO > 0.9, and MAF > 1% in both UKB and gnomAD datasets. Individuals retained for analysis were of European ancestry (field 22006), excluding any with failed genotyping quality, unusual heterozygosity, sex chromosome anomalies, or those who had withdrawn from the study by December 2021 (according to the sample-QC file from resource 531). Lastly, narrow-sense heritability estimates for each pain phenotype and diagnosis were derived using LDSC from the GWAS summary statistics²⁷.

Response to Reviewer #3:

Ref 3.1:

Overall, the manuscript is well written and although the data analyses has been quite complex, it is structured well. Due to the complexity of biomarkers investigated the reviewer can only evaluate parts of the methods applied. The procedures performed seem to be well elaborated.

Response: We are thankful for the positive feedback on the manuscript's structure and clarity.

Ref 3.2:

Originality and significance: The major claims are well expressed and represent a significant advance in the field. Data material is very interesting and methods applied are straightforward.

Response: We thank the reviewer for their positive comments.

Ref 3.3:

As far as I could figure out the analysis plan had not been preregistered. Statistics have to be evaluated by a specialized reviewer.

Response: The reviewer is correct, and the analysis plan was not preregistered. While our study was not preregistered, all our models were cross validated, and the relevant identified biomarkers were assessed in cohorts outside the UKBiobank, which avoids overfitting and ensures the robustness of our findings.

Ref 3.4:

Conclusions seem to be robust and valid. References are well selected and Figures are complex but very good. The Abstract is clear and all parts of the manuscript appropriate.

Response: We thank the reviewer for their positive comments.

Ref 3.5:

Although the manuscript is long (especially the methods) I do not have a strategy to write it shorter. Overall, I applaud the authors for evaluating this huge amount of data and for identifying the major results from this enormous study.

Response: We thank the reviewer for their feedback and recognition of our efforts in analyzing these large datasets.

Response to Reviewer #4:**Ref 4.1:**

Their overall conclusions challenge the concept of achieving an 'objective' marker for multiple pain states, which is important and highlights the potential pitfalls of seeking to establish such markers for complex, incompletely-understood medical conditions.

Response: We are thankful to the reviewer for highlighting the importance of our work.

Ref 4.2:

Generalisability - this encompasses two factors. First, UK Biobank cohort is subject to selection bias, which may affect the applicability of findings therefrom to other contexts. Did the authors consider the use of weighting to mitigate this participation/'healthy volunteer' bias? (E.g. <https://www.nature.com/articles/s41562-023-01579-9>)

***Response:** The generalizability is an important point, and we are thankful to the reviewer for raising it. We acknowledge that the UK Biobank participants may not fully represent the broader population. To evaluate the generalizability of our biomarkers, we employed strategies that avoided the data distortion typically associated with pseudo-resampling methods. Our rationale is that weighting 'healthy volunteers' to match a target population (in the case of the referenced article, this would be the five Health Survey of England (HSE) cohorts) inherently changes the structure of the data based on arbitrary criteria such as the selection of the reference cohort. While matching the UK Biobank to the HSE cohorts may provide a more accurate representation of the UK population, this approach may not generalize well to populations outside the UK, such as those in the United States. Moreover, the selection of health variables used to match the population (in this case alcohol intake, tobacco, income, household size, BMI, amongst other), over alternative variables that could be just as justified (e.g. quality of sleep, psychiatric conditions, physical exercise, history of trauma), involves a certain degree of flexibility and decision making that may influence the outcomes. Thus, choosing amongst different matching solutions could lead to different pseudo samples that impact the results. These problems can be solved by directly testing the generalizability of our biomarkers in different contexts, outside of the UKBiobank.*

*In our study, we emphasized the generalizability of our markers by leveraging multiple cohorts, ranging across a wide spectrum of demographics and health statuses. For instance, we applied our blood-based biomarkers in tens of thousands of participants from the All of Us cohort (**Fig. 2G**), which is a diverse and representative cohort from the United States showing very different demographics compared to the UKBiobank (see figures below). The same was done for the brain-based biomarkers, which were tested in patients from the Open Pain repository, containing MRI from chronic pain patients recruited in Chicago, the UK, and in Japan. To overcome the generalization problem of biomarkers, as stated by the reviewer, the state-of-the-art approach in machine learning is normally to directly apply the identified biomarkers in a different cohort of patients rather than creating counterfactual data based on either simulations or a weighting system that is inherently dependent on the variables used to weight the participants and the selected target population to which they are matched.*

We would like to emphasize that we took the generalizability problem very seriously. Below, we detail the demographic disparities across the UK Biobank, All of Us, and OpenPain cohorts, demonstrating how these differences support the applicability of our biomarkers across various global contexts and populations.

Color legend

Sex

Race/Ethnicity

Age

Based on the suggested paper from Reviewer 4, we have now also examined sociodemographic and health differences between the UK Biobank and All of Us cohort's using the metrics mentioned in the paper provided by the reviewer, such as smoking status, income, employment, general health, and alcohol consumption. The rationale is to demonstrate that despite differences in demographics and general health, our biomarkers generalized across populations, indicating that they were not the result of selection bias or other potential unidentified biases. Notably, the All of Us cohort had a higher prevalence of smokers (37.4% vs. 10.6% in UKB) and alcohol consumption (96.3% vs. 91.9% in UKB), and lower employment rates (40.8% vs. 57.1% in UKB).

The successful validation of our biomarkers in these external cohorts, which vary significantly in sociodemographic and health-related variables, reinforces the applicability of our findings beyond the UK Biobank. We believe these measures effectively mitigate concerns about selection bias and confirm the broader applicability of our conclusions.

Finally, we would like to emphasize the robustness of our blood-based biomarkers, which is evident from our longitudinal analyses. Despite the 'healthy' selection bias, we successfully predicted the development of new chronic pain conditions in healthy individuals. To our knowledge, this is the first large-scale study to identify biomarkers for chronic pain conditions that generalize across different cohorts and predict the onset of pain conditions over time.

These analyses have now been included in **Extended Fig. 8D** and added to the discussion:

Page 26: Additionally, the UK Biobank may have a 'healthy volunteer' selection bias given the low participation of 5.45%³⁴. We, however, generalized our blood and brain-based biomarkers in independent cohorts with different demographic and general health characteristics (**Extended Fig. 8D**), indicating that our biomarkers were not the result of selection bias or other potential unidentified biases.

Ref 4.2:

Second, the use of polygenic risk information was only obtained/evaluated in those with European ancestry. This significantly limits the implications of their findings to those from ethnic minority backgrounds. Did the authors consider using other approaches to maximise the involvement of people of non-European ancestry in the analyses in general? It is not clear from the Methods/main paper how this was sought. Do each of the analyses comprise different sub-cohorts?

Response: Our study utilized polygenic risk scores (PRS) based on genome-wide association studies (GWAS) from populations of European ancestry. We acknowledge the growing evidence that supports the development of inclusive PRS that better represent global genetic diversity¹⁴. However, our current study was constrained by the genetic makeup of the UK Biobank cohort, which is predominantly composed of individuals of European descent (>80%, data field 22006). The cohort's composition posed significant challenges for extending our analysis to non-European ancestries due to technical issues such as differences in allele frequencies, linkage disequilibrium structures, heterozygosity rates across populations, as well as differing admixture dosages from divergent ancestries¹⁵. Furthermore, the accuracy and validity of PRS are known to decrease as the genetic distance from the GWAS training population increases¹⁶. For example, studies have shown that PRS derived from a predominantly African American sample may perform poorly in individuals with significant European ancestry admixture¹⁷. Conversely, causal variants identified in European GWAS are not always applicable to other ancestries, though some research suggests a degree of shared causal variants across different groups¹⁸.

We would also like to emphasize that the genetic analyses did not provide very good predictions for most chronic pain conditions we evaluated, even when considering only the European ancestry group. Therefore, we avoided making any strong claims about genetic predisposition for either self-reported pain or chronic pain conditions throughout the manuscript. For other types of biological measures used to train our models, such as those involving blood biochemistry and brain imaging, we did not apply any ethnicity-based exclusion criteria. The differences based on sex and ethnicity were initially presented in **Extended Fig. 3**, for all biomarkers and conditions, where the results show minimal differences between groups. These were then validated in external datasets including more ethnically diverse populations. These validations suggest that the findings from our biochemical and brain-based models are applicable and generalizable across different ethnic backgrounds.

We have now included a limitations section in our manuscript's Discussion that focuses on how European-derived PRS restricts the applicability of our findings to other ancestries, among other concerns raised by reviewers.

Page 26: Our study has important limitations. First, the UK Biobank lacks diversity, being a predominantly white population of middle-aged and older individuals. This may restrict the applicability of our models, as studies have demonstrated that algorithms trained on mostly white participants can be mischaracterized in non-white participants³⁴. This is particularly relevant for our PRS analyses, where individuals of non-European ancestry were excluded from GWAS estimation, a decision informed by the existing literature and genetic makeup of the UK Biobank cohort, which is composed of greater than 80% European descent. However, other models in our study, such as those involving blood biochemistry and brain imaging, were validated using external datasets like All of Us (Fig. 2G) and OpenPain (Fig. 3E), which include more ethnically diverse populations. In most cases our biomarkers show minimal differences in performance based on sex and ethnicities (Extended Fig. 3). These validations suggest that the findings from our blood-based and brain-based markers are applicable and generalizable across different ethnic backgrounds.

Ref 4.3:

Missing data handling - median imputation was used to handle missing values in biochemical data. Why was this approach used rather than a multiple imputation, or a more robust form of deterministic imputation that integrates relationships between covariates? Median imputation will artificially manipulate/restrict the distribution of predictors.

*Response: We are thankful to the reviewer for their suggestion. Our decision to use median imputation was informed by previous work conducted by our group within the UK Biobank, in which we had compared multiple Bayesian imputation and simple median imputation for psychosocial variables. This comparison revealed minimal differences in downstream machine learning predictive performance when predicting chronic pain²¹. However, we recognize that biochemical measures differ in their characteristics and may indeed benefit from a more sophisticated imputation approach that considers relationships between covariates. We have since implemented multiple Bayesian imputation using scikit-learn for the biochemical data and re-analyzed its impact in our models predicting pain diagnoses. The results, now detailed in **Supplementary Fig. 11A and 11C** (see figure below), show negligible differences in both model performance and the correlation of structure coefficients between the two imputation methods; the average difference in AUC scores was only 0.0025, and the correlation between structure coefficients ranged between 0.95 and 0.99. This high level of concordance suggests that both imputation methods yield similar outcomes for our specific dataset and analysis.*

The similarity in our results across different imputation methods can be attributed to several key factors related to our dataset and the nature of the imputation techniques:

- 1. **Low Overall Missingness:** The biochemistry data exhibits relatively low missingness, ranging from 1% to 13%, with an average of only 4.5%. Such minimal missing data likely limits the impact of the imputation method on the overall data distribution.*
- 2. **Missingness in Non-Critical Predictors:** The assays with the highest missingness, such as Testosterone (13%), Phosphate (12.4%), and Glucose (12%), are weaker predictors in our*

models. In contrast, stronger predictors like C-reactive protein, Leukocyte count, and Lymphocyte percentage have much lower missing rates of 4.3%, 1.8%, and 1.6%, respectively. This may suggest that imputation does not significantly influence the more impactful predictors in our analyses.

- Data Structure and Distribution:** The distribution of our biochemical data may inherently align well with median values and represent the central tendency within our data.
- Covariate Relationships:** Although multiple Bayesian imputation accounts for relationships between covariates, the high multicollinearity among our biochemical predictors suggests that simpler imputation methods might suffice without significant information loss. This is supported by the strong consistency in structure coefficients observed across different imputation methods (Supplementary Figure C below).

A

B

Differences in model performance metrics between imputation methods (Median imputation score - Multiple imputation score)

	Log accuracy	Brier score	Pseudo R2
Mean difference	-0.0002	-0.0006	0.0007
Max difference	0.0477	0.0051	0.0211
Min difference	-0.0252	-0.0098	-0.0488

C

Based on these findings, we have decided to retain the median imputation approach in our manuscript to maintain consistency with our established methodology. We have updated the manuscript to include these results and a discussion on the rationale behind our choice of imputation methods.

Ref 4.4:

Clarity on definition of outcome - the authors discuss the pain phenotypes in the 'Online Methods' section. The timing of ascertaining prediction 'target' is not clear - I assume that these were based on baseline session at the assessment centre? It is useful to be completely clear about the temporality and other facets of defining the 'endpoint'. The title refers to 'future', but pain-associated diagnoses appear to be able to be present on baseline assessment, or on NHS records (during follow-up?). Were incident cases and prevalent cases used, or only one type? The relationship between the timing of the predictor measurements and the outcomes they seek to predict is not clear but impacts the interpretation of results and steers the appropriateness of analyses.

***Response:** We acknowledge the need for clearer articulation of the temporality and definition of our outcomes in the manuscript. In our study, we distinguish between two types of targets for our machine learning models: "Pain Phenotypes" and "Pain-Associated Diagnoses."*

- 1. **Pain Phenotypes collected on each visit:** These outcomes are based on self-reported pain experiences captured during each study session (Baseline and about 9-year follow-up). Each report is concurrent with the collection of specific biomarkers—biochemical assays at baseline and brain imaging measures at the 9-year follow-up. This ensures that our pain phenotypic predictive models use contemporaneous predictor measurements and outcomes.*
- 2. **Pain-Associated Diagnoses:** These diagnoses are sourced from two places: participant self-reports during study sessions and NHS records (including general practitioner and primary care data). In our analysis, a diagnosis is considered existing if it was reported during a study session coinciding with predictor measurements (e.g., biochemical assays at baseline) or if the diagnosis from NHS records predates the session. This method captures both incident and prevalent cases, allowing for cross-sectional (e.g., **Fig. 1C**) and longitudinal analyses (e.g., **Fig. 5G**) of pain-related conditions.*

*Acknowledging the reviewer's point about the necessity for precise clarity on the timing of predictor measurements and outcomes, we have made the following revisions. First, **Extended Fig. 1A** has been revised to more clearly illustrate the timeline of data acquisition across the three UK Biobank study visits, detailing which predictor variables were collected at each session.*

A | Timeline and candidate modalities

B | Machine learning pipeline for classifying pain endpoints

Extended Data Fig. 1 | Candidate modalities and machine learning pipeline. **A**, Top: A timeline depicts the collection of biological and psychosocial modalities across 3 different UK Biobank study sessions, with the number of samples available for analysis after data cleaning indicated above each modality icon. Below: Each modality is broken down into subcategories where applicable, with a description of example measures within these subcategories. Modalities are highlighted in large, bold font; subcategories in smaller bold font; and example measures are listed in bulleted regular font. **B**, A schematic outlines the machine learning pipeline used to evaluate the ability of models trained on candidate modalities to classify pain endpoints. A nested Cross-Validation (CV) approach with 5-fold inner and 5-fold outer CV was used to optimize model performance without data leakage. The inner loop optimizes performance by training a model on each training fold and tuning hyperparameters on the validation fold to maximize the score. In the outer loop, the model's generalizability is gauged by averaging the scores across left-out test folds. Three machine learning algorithms we're assessed: gradient boosting trees, logistic regression, and linear support vector machines. This process was iterated five times for each modality, with participant order randomized in each iteration to prevent model performance bias based on train/test participant arrangement. GBT, Gradient boosting trees; LR, Logistic regression; SVM, Support vector machine; ROC-AUC, Receiver operating characteristic area under the curve.

The term "future" in our initial title and analyses primarily refers to the predictive capacity of the identified biomarkers (from cross sectional analyses described above) in longitudinal datasets, as detailed in Fig. 5 and 6. These models aim to predict the onset of chronic pain conditions in participants who did not have such conditions at the time their initial biomarker or psychosocial factor measurements were taken. For example, our synergistic model (Fig. 5G,H) uses both biomarker and psychosocial data, trained on baseline data, to forecast the development of chronic

*pain conditions in healthy individuals. Additionally, the structural equation model in **Fig. 6** shows how conditions like rheumatoid arthritis are predicted using both biological and psychosocial factors, while the associated pain symptomatology is predicted exclusively through psychosocial factors.*

The online methods now include a more detailed description of the timing of predictor variables and target outcomes.

Page 38:

Deriving 14 target pain phenotypes: From these data, we derived 14 pain phenotypes, including a general chronic pain phenotype representing pain at any of the body sites, and a general acute pain phenotype. Additionally, we identified seven chronic pain site phenotypes, each representing pain experienced at one of the specified body sites, one phenotype for pain reported all over the body, and four phenotypes that quantified the number of chronic pain sites reported, categorizing the extent of chronic pain spread ranging from 1 to 4 or more distinct body sites. Importantly, pain-phenotypic predictive models were trained using data from the same sessions in which pain phenotypes were reported (see **Fig. 4**). For instance, models using blood immunoassays were designed to predict pain based on immunoassays conducted during the baseline session. Similarly, models based on brain imaging collected at the 9-year follow up visit aimed to predict pain reports from the 9-year follow-up, each time using data collected at that same follow-up visit. Thus, these models aim to predict the concurrent, or cross-sectional, presence of pain phenotypes.

Pain-Associated Diagnoses: For our analysis, we selected 35 pain-associated diagnoses based on their significant pain prevalence and ample sample size. Criteria included having over 45% pain prevalence (acute or chronic) and an occurrence exceeding 100 participants at the 9-year follow-up. These diagnoses were then aligned with the first occurrences database, ICD-10 codes and primary care data, collating all participants with a record for each specific diagnosis. For instance, the sciatica group amalgamated both self-reported and healthcare recorded (ICD codes: M543 & M544) instances of sciatica. A detailed list of codes for the 35 diagnoses is available in **Supplementary Table 7**. To ascertain that an illness's onset or diagnosis predated a given study visit, we contrasted the earliest recorded illness date with the participant's respective visit date at each visit (e.g. baseline, 4-year, and 9-year follow-up). This method allowed us to conduct both cross-sectional and longitudinal analyses, examining the presence or onset of diagnoses either concurrent with or following the measurement of predictor variables, as in **Fig. 2B**. Health records also facilitated time-to-event analyses by providing precise diagnosis dates (used in **Fig. 5G**).

Online UK Biobank Pain Questionnaire: From 2019 to 2020, the UK Biobank (UKB) conducted follow-up assessments through online questionnaires in a subset of participants originally recruited at baseline. These "Experience of Pain" questionnaires were administered approximately 8–13 years after the baseline visit (median: 10 years) to improve the phenotyping of individuals with chronic pain. Out of the 332,587 participants who received invitations, 167,255 completed the questionnaire. Subsections were used to examine the associations between biological and psychosocial variables recorded at baseline and longitudinal pain outcomes assessed at the online pain questionnaire. These pain outcomes measured several dimensions of chronic pain, such as its interference with daily activities measured via the Brief Pain Inventory, the number of self-reported chronic pain sites to assess pain spread, and the severity of the worst pain experienced in

the last 24 hours to gauge pain intensity. Detailed information about the questionnaire is available in the UKB documentation https://biobank.ndph.ox.ac.uk/showcase/ukb/docs/pain_questionnaire.pdf

Furthermore, to eliminate any potential confusion, we have revised the manuscript title excluding the term "future". The new title is now "A Holistic Framework for Biomarkers of Pain Conditions and Self-Reported Pain".

Ref 4.5:

Use of logistic regression - an example of why definition of the endpoint is crucial to understand. If there is a time-to-event component, i.e. individuals may not have had pain at baseline, but developed pain/received a pain state diagnosis years after covariate collection, then logistic regression is not an appropriate modelling choice (far better choices could be a flexible parametric survival model, or penalised Cox models that handle the time-to-event nature of these data).

Response: Thank you for this feedback. We appreciate the opportunity to clarify the specific contexts in which different models were employed.

In our study, logistic regression was applied exclusively in cross-sectional analyses. In these instances, the target diagnosis or pain phenotype had already been established prior to the measurement of predictor variables at each study session. For this specific scenario, where the goal was to differentiate between individuals with an existing diagnosis or pain phenotype and those without, logistic regression was used.

*For longitudinal analyses, where our objective was to explore the development of pain diagnoses or phenotypes subsequent to the measurement of predictor variables, we recognized the necessity for models that appropriately handle the time-to-event nature of the data. Accordingly, we utilized Cox proportional hazards models (see **Fig. 5G**). These models are better suited for assessing the impact of predictors over time, allowing us to estimate the hazard of developing the target conditions during the follow-up period.*

*We acknowledge that our initial manuscript may not have made these distinctions clear and could have led to confusion regarding our choice of analytical methods. We have made amendments in the manuscript to better delineate the timing of predictor variables and outcomes, specifically addressing the concerns raised in **Ref. 4.4**.*

Page 54:

For cross-sectional analyses involving established diagnoses or pain phenotypes prior to predictor measurements, we applied L2 (ridge) logistic regression. Longitudinal analyses, where pain diagnoses or phenotypes developed subsequent to predictor measurements, employed Cox proportional hazards models.

Ref 4.6:

Usefulness of AUC as a comparison between markers/models - there has been some criticism of comparing models using a single discrimination metric. Did the authors consider using a broader set of techniques to formally compare the predictive capabilities of models using different markers/different sets? (<https://www.fharrell.com/post/addvalue/>)

***Response:** We appreciate the reviewer's insight regarding the use of AUC scores. The main reason why we use AUC is because it is a widely accepted measure of model performance, particularly in terms of discrimination ability. Alternative metrics such as Brier Score are less commonly used and less understood by the machine learning community. Based on the reviewer's concern, we computed alternative metrics to rank our best blood-based biomarkers and the order of performance remained largely the same (see figure below).*

Our rationale to only present AUC in our manuscript is motivated by the advantages points that other metrics don't have:

- 1. **Robustness to Class Imbalance:** AUC is less sensitive to class imbalance, which is particularly relevant in our dataset where some pain diagnoses are less common (e.g., fibromyalgia, multiple sclerosis, ankylosing spondylitis)⁹. Other metrics such as accuracy can be sensitive to class imbalance and may be misleading in our case.*
- 2. **Threshold Independence:** AUC evaluates the model's performance across all possible classification thresholds, providing a comprehensive measure of the model's discriminative power, particularly if no established risk cutoff exists⁴.*
- 3. **Goals of the study:** AUC best aligns with the goals of discrimination in machine learning predictive paradigm, as outlined by³:*
 - **Discriminative Power:** AUC measures the ability of the model to rank positive cases higher than negative cases, which is highly relevant for predictive tasks. This focus on ranking and discrimination has been used in many biomedical machine learning applications^{7,8,10,11}.*
 - **Predictive Performance:** In the context of predicting pain diagnoses, the primary goal is to develop models that generalize well to new data and accurately distinguish between the case and control condition. AUC provides a robust measure of this discriminative ability, making it a suitable choice for evaluating our models^{5,6}.*

*We recognize that metrics such as the Brier score, log loss, and pseudo- R^2 offer valuable insights for statistical inference by evaluating the **differences between predicted probabilities and actual outcomes**. However, in our machine learning-focused study, the AUC is particularly crucial as it highlights the **discriminative ability of the biomarkers to classify cases from controls**. As an example, the figure below shows a broader set of performance metrics, as suggested in the referenced blog post. Specifically, we have incorporated the Brier score and Pseudo R^2 , along with a commonly used prediction-oriented metric, Balanced Accuracy. Our findings indicate that discrimination-oriented metrics such as the AUC score and Accuracy consistently demonstrate moderate to strong performance across models, with AUC scores ranging from 0.70 to 0.83 and Accuracy between 0.65 and 0.75. Conversely, model fit indices like Pseudo R^2 and Brier score show a broader range of effectiveness, from weak to strong, with Pseudo R^2 values between 0.13 and 0.33 and Brier scores from 0.17 to 0.22. Thus, the choice of performance metrics to represent model effectiveness remains debatable and surrounded by significant discourse^{3,11,12,13}. For instance, Assel et al. (2017) caution against the use of the Brier score in diagnostic tests or clinical*

prediction models, citing that the Brier score's dependency on prevalence can lead to misleading outcomes where clinical implications do not align with the prevalence, making it unsuitable for determining the clinical utility of a test or model or for choosing between two competing models. Yet, in the example below representing our best blood-based biomarkers, the various metrics were relatively concordant; the one with the best AUC also provided the best accuracy, the highest PseudoR2, and the lowest Brier Score.

*Providing multiple metrics of a given model and discussing their nuances and implications may be advised for the assessment of a single model. However, given the extensive number of models trained in our study—392 models covering 35 medical conditions, 14 self-report pain phenotypes; each of them trained using 16 predictor classes (8 biological and 8 psychosocial)—reporting multiple metrics and nuancing discrepancies between them may be overwhelming for readers. Here, we believe that focusing on AUC and **demonstrating that our best biomarkers generalized to other cohorts beyond the UK Biobank** should be sufficient to mitigate concerns about potential biases or uncertainty in predictive capacity. In the context of our manuscript, where the primary objective is to distinguish between cases and controls, discrimination measures such as AUC are particularly relevant and widely used (and understood) by the community. We therefore chose to keep the AUC score throughout our manuscript, as previously done by us²¹ and others^{1,11} working on predictive modeling in the UKBiobank.*

References:

1. Tian, Y. E. *et al.* Heterogeneous aging across multiple organ systems and prediction of chronic disease and mortality. *Nat Med* **29**, 1221–1231 (2023).
2. Dhindsa, R. S. *et al.* Rare variant associations with plasma protein levels in the UK Biobank. *Nature* **622**, 339–347 (2023).
3. Poldrack, R. A., Huckins, G. & Varoquaux, G. Establishment of Best Practices for Evidence for Prediction: A Review. *JAMA Psychiatry* **77**, 534 (2020).
4. Pencina, M. J., D'Agostino, R. B., D'Agostino, R. B. & Vasan, R. S. Evaluating the added predictive ability of a new marker: From area under the ROC curve to reclassification and beyond. *Statistics in Medicine* **27**, 157–172 (2008).
5. Steyerberg, E. W. *et al.* Assessing the Performance of Prediction Models: A Framework for Traditional and Novel Measures. *Epidemiology* **21**, 128–138 (2010).
6. Zou, K. H., O'Malley, A. J. & Mauri, L. Receiver-Operating Characteristic Analysis for Evaluating Diagnostic Tests and Predictive Models. *Circulation* **115**, 654–657 (2007).
7. Vamathevan, J. *et al.* Applications of machine learning in drug discovery and development. *Nat Rev Drug Discov* **18**, 463–477 (2019).
8. Buergel, T. *et al.* Metabolomic profiles predict individual multidisease outcomes. *Nat Med* **28**, 2309–2320 (2022).
9. Aurelio, Y. S., De Almeida, G. M., De Castro, C. L. & Braga, A. P. Learning from Imbalanced Data Sets with Weighted Cross-Entropy Function. *Neural Process Lett* **50**, 1937–1949 (2019).
10. Reddy, Y. N. V. *et al.* An evidence-based screening tool for heart failure with preserved ejection fraction: the HFpEF-ABA score. *Nat Med* (2024) doi:[10.1038/s41591-024-03140-1](https://doi.org/10.1038/s41591-024-03140-1).

11. Carrasco-Zanini, J. *et al.* Proteomic signatures improve risk prediction for common and rare diseases. *Nat Med* (2024) doi:[10.1038/s41591-024-03142-z](https://doi.org/10.1038/s41591-024-03142-z).
12. Assel, M., Sjoberg, D.D. & Vickers, A.J. The Brier score does not evaluate the clinical utility of diagnostic tests or prediction models. *Diagn Progn Res* **1**, 19 (2017). <https://doi.org/10.1186/s41512-017-0020-3>
13. Wu, Y.-C. & Lee, W.-C. Alternative Performance Measures for Prediction Models. *PLoS ONE* **9**, e91249 (2014).
14. Ben-Eghan, C. *et al.* Don't ignore genetic data from minority populations. *Nature* **585**, 184–186 (2020).
15. Atkinson, E. G. *et al.* Tractor uses local ancestry to enable the inclusion of admixed individuals in GWAS and to boost power. *Nat Genet* **53**, 195–204 (2021).
16. Ding, Y. *et al.* Polygenic scoring accuracy varies across the genetic ancestry continuum. *Nature* **618**, 774–781 (2023).
17. Bitarello, B. D. & Mathieson, I. Polygenic Scores for Height in Admixed Populations. *G3 (Bethesda)* **10**, 4027–4036 (2020).
18. Wang, Y. *et al.* Theoretical and empirical quantification of the accuracy of polygenic scores in ancestry divergent populations. *Nat Commun* **11**, 3865 (2020).
19. Lin, B. D. *et al.* Adjusting for population stratification in polygenic risk score analyses: a guide for model specifications in the UK Biobank. *J Hum Genet* **68**, 653–656 (2023).
20. van der Miesen, M. M., Lindquist, M. A. & Wager, T. D. Neuroimaging-based biomarkers for pain: state of the field and current directions. *Pain Rep* **4**, e751 (2019).
21. Tanguay-Sabourin, C., Fillingim, M., Guglietti, G.V. *et al.* A prognostic risk score for development and spread of chronic pain. *Nat Med* **29**, 1821–1831 (2023). <https://doi.org/10.1038/s41591-023-02430-4>